# Coherent Soft Imitation Learning

**Joe Watson**[*†‡]

**Sandy H. Huang**[§]

**Nicolas Heess**[§]

[§] **Google DeepMind**
London, United Kingdom
{shhuang,heess}@google.com

[†]**TU Darmstadt**
Darmstadt, Germany
joe@robot-learning.de

[‡]**Systems AI for Robot Learning**
German Research Center for AI
dfki.de

## Abstract

Imitation learning methods seek to learn from an expert either through behavioral cloning (BC) for the policy or inverse reinforcement learning (IRL) for the reward. Such methods enable agents to learn complex tasks from humans that are difficult to capture with hand-designed reward functions. Choosing between BC or IRL for imitation depends on the quality and state-action coverage of the demonstrations, as well as additional access to the Markov decision process. Hybrid strategies that combine BC and IRL are rare, as initial policy optimization against inaccurate rewards diminishes the benefit of pretraining the policy with BC. This work derives an imitation method that captures the strengths of both BC and IRL. In the entropy-regularized ('soft') reinforcement learning setting, we show that the behavioral-cloned policy can be used as both a shaped reward and a critic hypothesis space by inverting the regularized policy update. This *coherency* facilitates fine-tuning cloned policies using the reward estimate and additional interactions with the environment. Our approach conveniently achieves imitation learning through initial behavioral cloning and subsequent refinement via RL with online or offline data sources. The simplicity of the approach enables graceful scaling to high-dimensional and vision-based tasks, with stable learning and minimal hyperparameter tuning, in contrast to adversarial approaches. For the open-source implementation and simulation results, see joemwatson.github.io/csil.

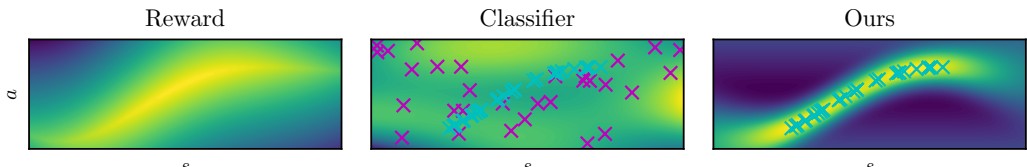

Figure 1: By using regression rather than classification, our approach infers a shaped reward from expert data for this contextual bandit problem, whereas classifiers require additional non-expert data and may struggle to resolve the difference between expert (✗) and non-expert (✗) samples.

## 1 Introduction

Imitation learning (IL) methods [1] provide algorithms for learning skills from expert demonstrations in domains such as robotics [2], either through direct mimicry (behavioral cloning, BC [3]) or inferring the latent reward (inverse reinforcement learning, IRL [4–6]) assuming a Markov decision process (MDP) [7, 8]. While BC is straightforward to implement, it is susceptible to covariate shift during evaluation [9]. IRL overcomes this problem but tackles the more complex task of jointly learning

---

[*]Corresponding author. Work done during an internship at Google DeepMind.

37th Conference on Neural Information Processing Systems (NeurIPS 2023).

a reward and policy from interactions. So far, these two approaches to imitation have been largely separated [10], as optimizing the policy with an evolving reward estimate counteracts the benefit of pre-training the policy [11]. Combining the objectives would require careful weighting during learning and lacks convergence guarantees [12]. The benefit of combining BC and IRL is sample-efficient learning that enables any BC policy to be further improved using additional experience.

**Contribution.** Our IL method naturally combines BC and IRL by using the perspectives of entropy-regularized reinforcement learning (RL) [13–19] and gradient-based IRL [20, 21]. Our approach uses a BC policy to define an estimate of a shaped reward for which it is optimal, which can then be used to finetune the policy with additional knowledge, such as online interactions, offline data, or a dynamics model. Using the cloned policy to specify the reward avoids the need for careful regularization and hyperparameter tuning associated with adversarial imitation learning [11, 22], and we estimate a shaped variant of the *true* reward, rather than use a heuristic proxy, e.g., [23, 24]. In summary,

1. We use entropy-regularized RL to obtain a reward for which a behavioral-cloned policy is optimal, which we use to improve the policy and overcome the covariate shift problem.
2. We introduce approximate stationary stochastic process policies to implement this approach for continuous control by constructing specialized neural network architectures.
3. We show strong performance on online and offline imitation for high-dimensional and image-based continuous control tasks compared to current state-of-the-art methods.

## 2 Background & related work

A Markov decision process is a tuple $\langle \mathcal{S}, \mathcal{A}, \mathcal{P}, r, \gamma, \mu_0 \rangle$, where $\mathcal{S}$ is the state space, $\mathcal{A}$ is the action space, $\mathcal{P} : \mathcal{S} \times \mathcal{A} \times \mathcal{S} \to \mathbb{R}^+$ is the transition model, $r : \mathcal{S} \times \mathcal{A} \to \mathbb{R}$ is the reward function, $\gamma$ is the discount factor, and $\mu_0$ is the initial state distribution $\boldsymbol{s}_0 \sim \mu_0(\cdot)$. When evaluating a policy $\pi \in \Pi$, we use the occupancy measure $\rho_\pi(\boldsymbol{s}, \boldsymbol{a}) \triangleq \pi(\boldsymbol{a} \mid \boldsymbol{s}) \, \mu_\pi(\boldsymbol{s})$, $\mu_\pi(\boldsymbol{s}) \triangleq \sum_{t=0}^\infty \gamma^t \mu_t^\pi(\boldsymbol{s})$ where $\mu_{t+1}^\pi(\boldsymbol{s}') = \int \mathcal{P}(\boldsymbol{s}' \mid \boldsymbol{s}, \boldsymbol{a}) \, \pi(\boldsymbol{a} \mid \boldsymbol{s}) \, \mu_t^\pi(\boldsymbol{s}) \, \mathrm{d}\boldsymbol{s} \, \mathrm{d}\boldsymbol{a}$. This measure is used to compute infinite-horizon discounted expectations, $\mathbb{E}_{\boldsymbol{s}, \boldsymbol{a} \sim \rho_\pi}[f(\boldsymbol{s}, \boldsymbol{a})] = \mathbb{E}_{\boldsymbol{s}_{t+1} \sim \mathcal{P}(\cdot \mid \boldsymbol{s}_t, \boldsymbol{a}_t), \, \boldsymbol{a}_t \sim \pi(\cdot \mid \boldsymbol{s}_t)}[\sum_{t=0}^\infty \gamma^t f(\boldsymbol{s}_t, \boldsymbol{a}_t)]$. Densities $d_\pi(\boldsymbol{s}, \boldsymbol{a}) = (1 - \gamma) \, \rho_\pi(\boldsymbol{s}, \boldsymbol{a})$ and $\nu_\pi(\boldsymbol{s}) = (1 - \gamma) \, \mu_\pi(\boldsymbol{s})$ are normalized measures. When the reward is unknown, imitation can be performed from a dataset $\mathcal{D}$ of transitions $(\boldsymbol{s}, \boldsymbol{a}, \boldsymbol{s}')$ as demonstrations, obtained from the discounted occupancy measure of the expert policy $\pi_* = \arg\max_\pi \mathbb{E}_{\boldsymbol{s}, \boldsymbol{a} \sim \rho_\pi}[r(\boldsymbol{s}, \boldsymbol{a})]$. The policy could be inferred directly or by inferring a reward and policy jointly, referred to as behavioral cloning and inverse reinforcement learning respectively.

**Behavioral cloning.** The simplest approach to imitation is to directly mimic the behavior using regression [3], $\min_{\pi \in \Pi} l(\pi, \mathcal{D}) + \lambda \Psi(\pi)$, with some loss function $l$, hypothesis space $\Pi$ and regularizer $\Psi$. This method is effective with sufficient data coverage and an appropriate hypothesis space but suffers from compounding errors and cannot improve unless querying the expert [9]. The MIMIC-EXP algorithm of Rajaraman et al. [25] shows that the optimal no-interaction policy matches BC.

**Inverse reinforcement learning.** Rather than just inferring the policy, IRL infers the reward and policy jointly and relies on access to the underlying MDP [26, 5]. IRL iteratively finds a reward for which the expert policy is optimal compared to the set of possible policies while also finding the policy that maximizes this reward function. The learned policy seeks to match or even improve upon the expert using additional data beyond the demonstrations, such as environment interactions [6]. To avoid repeatedly solving the inner RL problem, one can consider the game-theoretic approach [27], where the optimization problem is a two-player zero-sum game where the policy and reward converge to the saddle point minimizing the 'apprenticeship error' $\min_{\pi \in \Pi} \max_{r \in \mathcal{R}} \mathbb{E}_\mathcal{D}[r] - \mathbb{E}_{\rho_\pi}[r]$. This approach is attractive as it learns the policy and reward concurrently [28]. However, in practice, saddle-point optimization can be challenging when scaling to high dimensions and can exhibit instabilities and hyperparameter sensitivity [11]. Compared to BC, IRL methods in theory scale to longer task horizons by overcoming compounding errors through environment interactions [29].

**Maximum entropy inverse reinforcement learning & soft policy iteration.** IRL is an under-specified problem [26], in particular in regions outside of the demonstration distribution. An effective way to address this is *causal entropy* regularization [30], by applying the principle of maximum entropy [31] to IRL (ME-IRL). Moreover, the entropy-regularized formulation has an elegant closed-form solution [30]. The approach can be expressed as an equivalent constrained minimum relative entropy problem for policy $q(\boldsymbol{a} \mid \boldsymbol{s})$ against a uniform prior $p(\boldsymbol{a} \mid \boldsymbol{s}) = \mathcal{U}_\mathcal{A}(\boldsymbol{a})$ using the Kullback-

Leibler (KL) divergence and a constraint matching expected features $\phi$, with Lagrangian

$$\min_{q,\boldsymbol{w}} \int_{\mathcal{S}} \nu_q(\boldsymbol{s}) \, \mathbb{D}_{\mathrm{KL}}[q(\boldsymbol{a} \mid \boldsymbol{s}) \mid\mid p(\boldsymbol{a} \mid \boldsymbol{s})] \, \mathrm{d}\boldsymbol{s} + \boldsymbol{w}^\top (\mathbb{E}_{\boldsymbol{s},\boldsymbol{a} \sim \mathcal{D}}[\boldsymbol{\phi}(\boldsymbol{s},\boldsymbol{a})] - \mathbb{E}_{\boldsymbol{s},\boldsymbol{a} \sim \rho_q}[\boldsymbol{\phi}(\boldsymbol{s},\boldsymbol{a})]). \quad (1)$$

The constraint term can be interpreted as the apprenticeship error, where $\boldsymbol{w}^\top \boldsymbol{\phi}(\boldsymbol{s},\boldsymbol{a}) = r_{\boldsymbol{w}}(\boldsymbol{s},\boldsymbol{a})$. Solving Equation 1 using dynamic programming yields the 'soft' Bellman equation [30],

$$\mathcal{Q}(\boldsymbol{s},\boldsymbol{a}) = r_{\boldsymbol{w}}(\boldsymbol{s},\boldsymbol{a}) + \gamma \mathbb{E}_{\boldsymbol{s}' \sim \mathcal{P}(\cdot \mid \boldsymbol{s},\boldsymbol{a})}[\mathcal{V}_\alpha(\boldsymbol{s}')], \; \mathcal{V}_\alpha(\boldsymbol{s}) = \alpha \log \int_{\mathcal{A}} \exp\left(\frac{1}{\alpha}\mathcal{Q}(\boldsymbol{s},\boldsymbol{a})\right) p(\boldsymbol{a} \mid \boldsymbol{s}) \, \mathrm{d}\boldsymbol{a}, \quad (2)$$

for temperature $\alpha = 1$. Using Jensen's inequality and importance sampling, this target is typically replaced with a lower-bound [32], which has the same optimum and samples from the optimized policy rather than the initial policy like many practical deep RL algorithms [17]:

$$\mathcal{Q}(\boldsymbol{s},\boldsymbol{a}) = r_{\boldsymbol{w}}(\boldsymbol{s},\boldsymbol{a}) + \gamma \, \mathbb{E}_{\boldsymbol{a}' \sim q(\cdot \mid \boldsymbol{s}'), \, \boldsymbol{s}' \sim \mathcal{P}(\cdot \mid \boldsymbol{s},\boldsymbol{a})}[\mathcal{Q}(\boldsymbol{s}',\boldsymbol{a}') - \alpha \, (\log q(\boldsymbol{a}' \mid \boldsymbol{s}') - \log p(\boldsymbol{a}' \mid \boldsymbol{s}'))]. \quad (3)$$

The policy update blends the exponentiated advantage function 'pseudo-likelihood' with the prior, as a form of regularized Boltzmann policy [17] that resembles a Bayes posterior [33],

$$q_\alpha(\boldsymbol{a} \mid \boldsymbol{s}) \propto \exp(\alpha^{-1}(\mathcal{Q}(\boldsymbol{s},\boldsymbol{a}) - \mathcal{V}_\alpha(\boldsymbol{s})) \, p(\boldsymbol{a} \mid \boldsymbol{s}). \quad (4)$$

These regularized updates can also be used for RL, where it is the solution to a KL-regularized RL objective, $\max_q \mathbb{E}_{\boldsymbol{s},\boldsymbol{a} \sim \rho_q}[\mathcal{Q}(\boldsymbol{s},\boldsymbol{a})] - \alpha \, \mathbb{D}_{\mathrm{KL}}[q(\boldsymbol{a} \mid \boldsymbol{s}) \mid\mid p(\boldsymbol{a} \mid \boldsymbol{s})]$, where the temperature $\alpha$ now controls the strength of the regularization. This regularized policy update is known as soft- [14, 17] or posterior policy iteration [34, 35] (SPI, PPI), as it resembles a Bayesian update. In the function approximation setting, the update is performed in a variational fashion by minimizing the reverse KL divergence between the parametric policy $q_{\boldsymbol{\theta}}$ and the critic-derived update at sampled states $\boldsymbol{s}$ [17],

$$\boldsymbol{\theta}_* = \arg\min_{\boldsymbol{\theta}} \mathbb{E}_{\boldsymbol{s} \sim \mathcal{B}}[\mathbb{D}_{\mathrm{KL}}[q_{\boldsymbol{\theta}}(\boldsymbol{a} \mid \boldsymbol{s}) \mid\mid q_\alpha(\boldsymbol{a} \mid \boldsymbol{s})]] = \arg\max_{\boldsymbol{\theta}} \mathcal{J}_\pi(\boldsymbol{\theta}),$$
$$\mathcal{J}_\pi(\boldsymbol{\theta}) = \mathbb{E}_{\boldsymbol{a} \sim q_{\boldsymbol{\theta}}(\cdot \mid \boldsymbol{s}), \, \boldsymbol{s} \sim \mathcal{B}}[\mathcal{Q}(\boldsymbol{s},\boldsymbol{a}) - \alpha \, (\log q_{\boldsymbol{\theta}}(\boldsymbol{a} \mid \boldsymbol{s}) - \log p(\boldsymbol{a} \mid \boldsymbol{s}))]. \quad (5)$$

The above objective $\mathcal{J}_\pi$ can be maximized using reparameterized gradients and minibatches from replay buffer $\mathcal{B}$ [17]. A complete derivation of SPI and ME-IRL is provided in Appendix K.1.

**Gradient-based inverse reinforcement learning.** An alternative IRL strategy is a gradient-based approach (GIRL) that avoids saddle-point optimization by learning a reward function such that the BC policy's policy gradient is zero [20], which satisfies first-order optimality. However, this approach does not remedy the ill-posed nature of IRL. Moreover, the Hessian is required for the sufficient condition of optimality [36], which is undesirable for policies with many parameters.

**Related work.** Prior state-of-the-art methods have combined the game-theoretic IRL objective [27] with entropy-regularized RL. These methods can be viewed as minimizing a divergence between the expert and policy, and include GAIL [37], AIRL [38], $f$-MAX [39], DAC [40], ValueDICE [41], IQLearn [42] and proximal point imitation learning (PPIL, [43]). These methods differ through their choice of on-policy policy gradient (GAIL) or off-policy actor-critic (DAC, IQLearn, PPIL), and also how the minimax optimization is implemented, e.g., using a classifier (GAIL, DAC, AIRL), implicit reward functions (ValueDICE, IQLearn) or Lagrangian dual objective (PPIL). Alternative approaches to entropy-regularized IL use the Wasserstein metric (PWIL) [44], labelling of sparse proxy rewards (SQIL) [23], feature matching [30, 45–47], maximum likelihood [48] and matching state marginals [49] to specify the reward. Prior works at the intersection of IRL and BC include policy matching [50], policy-based GAIL classifiers [51], annealing between a BC and GAIL policy [12] and discriminator-weighted BC [52]. Our approach is inspired by gradient-based IRL [20, 36, 21], which avoids the game-theoretic objective and instead estimates the reward by analyzing the policy update steps with respect to the BC policy. An entropy-regularized GIRL setting was investigated in the context of learning from policy updates [53]. For further discussion on related work, see Appendix A.

## 3 Coherent imitation learning

For efficient and effective imitation learning, we would like to combine the simplicity of behavioral cloning with the structure of the MDP, as this would provide a means to initialize and refine the imitating policy through interactions with the environment. In this section, we propose such a method using a GIRL-inspired connection between BC and IRL in the entropy-regularized RL setting. By inverting the entropy-regularized policy update in Equation 4, we derive a reward for which the behavioral-cloning policy is optimal, a quality we refer to as *coherence*.

**Inverting soft policy iteration and reward shaping.** The $\alpha$-regularized closed-form policy based on Equation 4 can be inverted to provide expressions for the critic and reward (Theorem 1).

**Theorem 1.** *(KL-regularized policy improvement inversion). Let $p$ and $q_\alpha$ be the prior and pseudo-posterior policy given by posterior policy iteration (Equation 4). The critic can be expressed as*

$$\mathcal{Q}(\boldsymbol{s}, \boldsymbol{a}) = \alpha \log \frac{q_\alpha(\boldsymbol{a} \mid \boldsymbol{s})}{p(\boldsymbol{a} \mid \boldsymbol{s})} + \mathcal{V}_\alpha(\boldsymbol{s}), \quad \mathcal{V}_\alpha(\boldsymbol{s}) = \alpha \log \int_{\mathcal{A}} \exp\left(\frac{1}{\alpha} \mathcal{Q}(\boldsymbol{s}, \boldsymbol{a})\right) p(\boldsymbol{a}|\boldsymbol{s})\, \mathrm{d}\boldsymbol{a}. \quad (6)$$

*Substituting into the KL-regularized Bellman equation lower-bound from Equation 2,*

$$r(\boldsymbol{s}, \boldsymbol{a}) = \alpha \log \frac{q_\alpha(\boldsymbol{a} \mid \boldsymbol{s})}{p(\boldsymbol{a} \mid \boldsymbol{s})} + \mathcal{V}_\alpha(\boldsymbol{s}) - \gamma\, \mathbb{E}_{\boldsymbol{s}' \sim \mathcal{P}(\cdot|\boldsymbol{s}, \boldsymbol{a})}\left[\mathcal{V}_\alpha(\boldsymbol{s}')\right]. \quad (7)$$

*The $\mathcal{V}_\alpha(\boldsymbol{s})$ term is the 'soft' value function. We assume $q_\alpha(\boldsymbol{a} \mid \boldsymbol{s}) = 0$ whenever $p(\boldsymbol{a} \mid \boldsymbol{s}) = 0$.*

For the proof, see Appendix K.2. Jacq et al. [53] and Cao et al. [54] derived a similar inversion but for a maximum entropy policy iteration between two policies, rather than PPI between the prior and posterior. These expressions for the reward and critic are useful as they can be viewed as a valid form of shaping (Lemma 1) [54, 53], where $\alpha(\log q(\boldsymbol{a} \mid \boldsymbol{s}) - \log p(\boldsymbol{a} \mid \boldsymbol{s}))$ is a shaped 'coherent' reward.

**Lemma 1.** *(Reward shaping, Ng et al. [55]). For a reward function $r : \mathcal{S} \times \mathcal{A} \to \mathbb{R}$ with optimal policy $\pi \in \Pi$, for a bounded state-based potential function $\Psi : \mathcal{S} \to \mathbb{R}$, a reward function $\tilde{r}$ shaped by the potential function satisfying $r(\boldsymbol{s}, \boldsymbol{a}) = \tilde{r}(\boldsymbol{s}, \boldsymbol{a}) + \Psi(\boldsymbol{s}) - \gamma\, \mathbb{E}_{\boldsymbol{s}' \sim \mathcal{P}(\cdot|\boldsymbol{s}, \boldsymbol{a})}[\Psi(\boldsymbol{s}')]$ has a shaped critic $\tilde{\mathcal{Q}}$ satisfying $\mathcal{Q}(\boldsymbol{s}, \boldsymbol{a}) = \tilde{\mathcal{Q}}(\boldsymbol{s}, \boldsymbol{a}) + \Psi(\boldsymbol{s})$ and the same optimal policy as the original reward.*

**Definition 1.** *The shaped 'coherent' reward and critic are derived from the log policy ratio by combining Lemma 1 and Theorem 1, with value function $\mathcal{V}_\alpha(\boldsymbol{s})$ as the potential $\Psi(\boldsymbol{s})$. When policy $q_\alpha(\boldsymbol{a}|\boldsymbol{s})$ models the data $\mathcal{D}$ while matching its prior $p$ otherwise, the density ratio should exhibit the following shaping*

$$\tilde{r}(\boldsymbol{s}, \boldsymbol{a}) = \alpha \log \frac{q_\alpha(\boldsymbol{a} \mid \boldsymbol{s})}{p(\boldsymbol{a} \mid \boldsymbol{s})} \begin{cases} \geq 0 \text{ if } \boldsymbol{s}, \boldsymbol{a} \in \mathcal{D}, \\ < 0 \text{ if } \boldsymbol{s} \in \mathcal{D}, \boldsymbol{a} \notin \mathcal{D}, \\ = 0 \text{ if } \boldsymbol{s} \notin \mathcal{D}, \forall \boldsymbol{a} \in \mathcal{A}. \end{cases}$$

*In a continuous setting, the learned policy should capture this shaping approximately (Figure 2).*

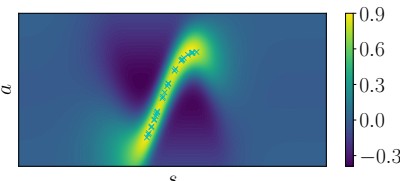

Figure 2: The coherent reward from Figure 1, made using a stationary Gaussian process policy, extended out-of-distribution. The reward approximates the shaping described in Definition 1.

Coherent soft imitation learning (CSIL, Algorithm 1) uses the BC policy to initialize the coherent reward and uses this reward to improve the policy further with additional interactions outside of $\mathcal{D}$. However, for the coherent reward to match Definition 1, the policy $q(\boldsymbol{a} \mid \boldsymbol{s})$ needs to match the policy prior $p(\boldsymbol{a} \mid \boldsymbol{s})$ outside of the data distribution. To achieve this, it is useful to recognize the policy $q(\boldsymbol{a} \mid \boldsymbol{s})$ as a 'pseudo-posterior' of prior $p(\boldsymbol{a} \mid \boldsymbol{s})$ and incorporate this Bayesian view into the BC step.

**Imitation learning with pseudo-posteriors.** The 'pseudo-posteriors' formalism [35, 56–59] is a generalized way of viewing the policy derived for ME-IRL and KL-regularized RL in Equation 4.

**Definition 2.** *Pseudo-posteriors are solutions to KL-constrained or minimum divergence problems with an additional scalar objective or vector-valued constraint term and Lagrange multipliers $\lambda$,*

$$\max_q \quad \mathbb{E}_{\boldsymbol{x} \sim q(\cdot)}[f(\boldsymbol{x})] - \lambda\left(\mathbb{D}_{\mathrm{KL}}[q(\boldsymbol{x}) \mid\mid p(\boldsymbol{x})] - \epsilon\right) \quad \to \quad q_\lambda(\boldsymbol{x}) \propto \exp(\lambda^{-1} f(\boldsymbol{x}))\, p(\boldsymbol{x}).$$

$$\min_q \quad \mathbb{D}_{\mathrm{KL}}[q(\boldsymbol{x}) \mid\mid p(\boldsymbol{x})] - \boldsymbol{\lambda}^\top\left(\mathbb{E}_{\boldsymbol{x} \sim q(\cdot)}[\boldsymbol{f}(\boldsymbol{x})] - \boldsymbol{f}^*\right) \quad \to \quad q_{\boldsymbol{\lambda}}(\boldsymbol{x}) \propto \exp(\boldsymbol{\lambda}^\top \boldsymbol{f}(\boldsymbol{x}))\, p(\boldsymbol{x}).$$

*The $\exp(\boldsymbol{\lambda}^\top \boldsymbol{f}(\boldsymbol{x}))$ term is an unnormalized 'pseudo-likelihood', as it facilities Bayesian inference to solve a regularized optimization problem specified by $\boldsymbol{f}$.*

Optimizing the policy distribution, the top objective captures KL-regularized RL, where $f$ is the return, and the regularization is implemented as a constraint with bound $\epsilon$ or soft penalty with constant $\lambda$. The bottom objective is the form seen in ME-IRL (Equation 1), where $\boldsymbol{f}$ is the feature space that defines the reward model. Contrasting these pseudo-posterior policies against BC with Gaussian processes, e.g., [60, 61], we can compare the Bayesian likelihood used for regression with the critic-based pseudo-likelihood obtained from imitation learning. The pseudo-likelihoods in entropy-regularized IL methods produce an effective imitating policy by incorporating the MDP and trajectory distribution because $f$ captures the cumulative reward, compared to just the action prediction error typically captured by BC regression likelihoods. This point is expanded in Appendix B.

---

**Algorithm 1:** Coherent soft imitation learning (CSIL)

**Data:** Expert demonstrations $\mathcal{D}$, initial temperature $\alpha$, refinement temperature, $\beta$,
parametric policy class $q_{\boldsymbol{\theta}}(\boldsymbol{a} \mid \boldsymbol{s})$, prior policy $p(\boldsymbol{a} \mid \boldsymbol{s})$, regression regularizer $\Psi$, iterations $N$
**Result:** $q_{\boldsymbol{\theta}_N}(\boldsymbol{a} \mid \boldsymbol{s})$, matching or improving the initial policy $q_{\boldsymbol{\theta}_1}(\boldsymbol{a} \mid \boldsymbol{s})$
Train initial policy from demonstrations, $\boldsymbol{\theta}_1 = \arg\max_{\boldsymbol{\theta}} \mathbb{E}_{\boldsymbol{s},\boldsymbol{a}\sim\mathcal{D}}[\log q_{\boldsymbol{\theta}}(\boldsymbol{a} \mid \boldsymbol{s}) - \Psi(\boldsymbol{\theta})]$;
Define fixed shaped coherent reward, $\tilde{r}_{\boldsymbol{\theta}_1}(\boldsymbol{s}, \boldsymbol{a}) = \alpha(\log q_{\boldsymbol{\theta}_1}(\boldsymbol{a} \mid \boldsymbol{s}) - \log p(\boldsymbol{a} \mid \boldsymbol{s}))$;
Choose reference policy $\pi(\boldsymbol{a} \mid \boldsymbol{s})$ to be $p(\boldsymbol{a} \mid \boldsymbol{s})$ or $q_{\boldsymbol{\theta}_1}(\boldsymbol{a} \mid \boldsymbol{s})$ and initialize the shaped critic,
$\quad \tilde{\mathcal{Q}}_1(\boldsymbol{s}, \boldsymbol{a}) = \tilde{r}_{\boldsymbol{\theta}_1}(\boldsymbol{s}, \boldsymbol{a}) + \gamma\,(\tilde{\mathcal{Q}}_1(\boldsymbol{s}', \boldsymbol{a}') - \alpha\,(\log q_{\boldsymbol{\theta}_1}(\boldsymbol{a}' \mid \boldsymbol{s}') - \log \pi(\boldsymbol{a}' \mid \boldsymbol{s}'))$, $\boldsymbol{s}, \boldsymbol{a}, \boldsymbol{s}', \boldsymbol{a}' \sim \mathcal{D}$;
**for** $n = 2 \to N$ **do**         `// finetune policy with reinforcement learning`
$\quad$ Compute $\tilde{\mathcal{Q}}_n$ and $q_{\boldsymbol{\theta}_n}$ using soft policy iteration (Section 2), e.g. SAC, with temperature $\beta$.
**end**

---

**Behavior cloning with pseudo-posteriors.**     Viewing the policy $q(\boldsymbol{a} \mid \boldsymbol{s})$ as a pseudo-posterior has three main implications when performing behavior cloning and designing the policy and prior:

1. We perform conditional density estimation to maximize the posterior predictive likelihood, rather than using a data likelihood, which would require assuming additive noise [62, 33].
2. We require a hypothesis space $p(\boldsymbol{a} \mid \boldsymbol{s}, \boldsymbol{w})$, e.g., a tabular policy or function approximator, that is used to define both the prior and posterior through weights $\boldsymbol{w}$.
3. We need a fixed weight prior $p(\boldsymbol{w})$ that results in a predictive distribution that matches the desired policy prior $p(\boldsymbol{a} \mid \boldsymbol{s}) = \int p(\boldsymbol{a} \mid \boldsymbol{s}, \boldsymbol{w})p(\boldsymbol{w})\,\mathrm{d}\boldsymbol{w}\ \forall \boldsymbol{s} \in \mathcal{S}$.

These desiderata are straightforward to achieve for maximum entropy priors in the tabular setting, where count-based conditional density estimation is combined with the prior, e.g., $p(\boldsymbol{a} \mid \boldsymbol{s}) = \mathcal{U}_{\mathcal{A}}(\boldsymbol{a})$. Unfortunately, point 3 is challenging in the continuous setting when using function approximation, as shown in Figure 3. However, we can adopt ideas from Gaussian process theory to approximate such policies, which is discussed further in Section 4. The learning objective combines maximizing the likelihood of the demonstrations w.r.t. the predictive distribution, as well as KL regularization against the prior weight distribution, performing regularized heteroscedastic regression [63],

$$\max_{\boldsymbol{\theta}} \mathbb{E}_{\boldsymbol{a},\boldsymbol{s}\sim\mathcal{D}}[\log q_{\boldsymbol{\theta}}(\boldsymbol{a} \mid \boldsymbol{s})] - \lambda\,\mathbb{D}_{\mathrm{KL}}[q_{\boldsymbol{\theta}}(\boldsymbol{w}) \,\|\, p(\boldsymbol{w})], \quad q_{\boldsymbol{\theta}}(\boldsymbol{a} \mid \boldsymbol{s}) = \int p(\boldsymbol{a} \mid \boldsymbol{s}, \boldsymbol{w})\, q_{\boldsymbol{\theta}}(\boldsymbol{w})\,\mathrm{d}\boldsymbol{w}. \quad (8)$$

This objective has been used to fit parametric Gaussian processes with a similar motivation [64].

We now describe how this BC approach and the coherent reward are used for imitation learning (Algorithm 1) when combined with soft policy iteration algorithms, such as SAC.

On a high level, Algorithm 1 can be summarized by the following steps,

1. Perform regularized BC on the demonstrations with a parametric stochastic policy $q_{\boldsymbol{\theta}}(\boldsymbol{a} \mid \boldsymbol{s})$.
2. Define the coherent reward, $r_{\boldsymbol{\theta}}(\boldsymbol{s}, \boldsymbol{a}) = \alpha(\log q_{\boldsymbol{\theta}}(\boldsymbol{a} \mid \boldsymbol{s}) - \log p(\boldsymbol{a} \mid \boldsymbol{s}))$, with temperature $\alpha$.
3. With temperature $\beta < \alpha$, perform SPI to improve on the BC policy with the coherent reward.

The reduced temperature $\beta$ is required to improve the policy after inverting it with temperature $\alpha$.

This coherent approach contrasts combining BC and prior actor-critic IRL methods, where learning the reward and critic from scratch can lead to 'unlearning' the initial BC policy [11]. Moreover, while our method does not seek to learn the true underlying reward, but instead opts for one shaped by the environment used to generate demonstrations, we believe this is a reasonable compromise in practice. Firstly, CSIL uses IRL to tackle the covariate shift problem in BC, as the reward shaping (Definition 1) encourages returning to the demonstration distribution when outside it, which is also the goal of prior imitation methods, e.g., [23, 24]. Secondly, in the entropy-regularized setting, an IRL method has the drawback of requiring demonstrations generated from two different environments or discount factors to accurately infer the true reward [54], which is often not readily available in practice, e.g., when teaching a single robot a task. Finally, additional MLPs could be used to estimate the true (unshaped) reward from data [65, 53] if desired. We also provide a divergence minimization perspective of CSIL in Section J of the Appendix. Another quality of CSIL is that it is conceptually simpler than alternative IRL approaches such as AIRL. As we simply use a shaped estimate of the true reward with entropy-regularized RL, CSIL inherits the theoretical properties of these regularized algorithms [17, 66] regarding performance. As a consequence, this means that analysis of the imitation quality requires analyzing the initial behavioral cloning procedure.

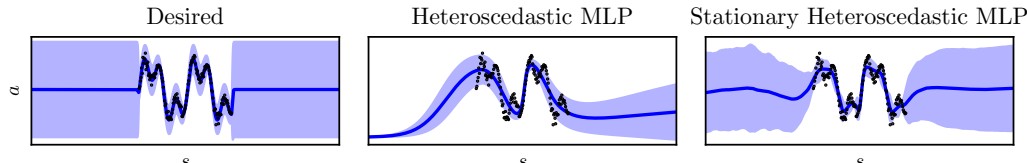

| Desired | Heteroscedastic MLP | Stationary Heteroscedastic MLP |

Figure 3: For the coherent reward to be effective, the stochastic policy should return to the prior distribution outside of the data distribution (left). The typical heteroscedastic parametric policies have undefined out-of-distribution behavior and typically collapse to the action limits due to the network extrapolation and tanh transformation (middle). By approximating stationary Gaussian processes, we can design policies that exhibit the desired behavior with minimal network modifications (right).

**Partial expert coverage, minimax optimization, and regularized regression.** The reward shaping theory from Lemma 1 shows that an advantage function can be used as a shaped reward function when the state potential is a value function. By inverting the soft policy update, the log policy ratio acts as a form of its advantage function. However, in practice, we do not have access to the complete advantage function but rather an estimate due to finite samples and partial state coverage. Suppose the expert demonstrations provide only partial coverage of the state-action space. In this case, the role of an inverse reinforcement learning algorithm is to use additional knowledge of the MDP, e.g., online samples, a dynamics model, or an offline dataset, to improve the reward estimate. As CSIL uses the log policy ratio learned only from demonstration data, how can it be a useful shaped reward estimate? We show in Theorem 2 that using entropy regularization in the initial behavioral cloning step plays a similar role to the saddle-point optimization in game-theoretic IRL.

**Theorem 2.** *(Coherent inverse reinforcement learning as KL-regularized behavioral cloning). A KL-regularized game-theoretic IRL objective, with policy $q_{\boldsymbol{\theta}}(\boldsymbol{a} \mid \boldsymbol{s})$ and coherent reward parameterization $r_{\boldsymbol{\theta}}(\boldsymbol{s}, \boldsymbol{a}) = \alpha(\log q_{\boldsymbol{\theta}}(\boldsymbol{a} \mid \boldsymbol{s}) - \log p(\boldsymbol{a} \mid \boldsymbol{s}))$ where $\alpha \geq 0$, is lower bounded by a scaled KL-regularized behavioral cloning objective and a constant term when the optimal policy, posterior, and prior share a hypothesis space $p(\boldsymbol{a} \mid \boldsymbol{s}, \boldsymbol{w})$ and finite parameters $\boldsymbol{w}$,*

$$\mathbb{E}_{\boldsymbol{s}, \boldsymbol{a} \sim \mathcal{D}}\left[r_{\boldsymbol{\theta}}(\boldsymbol{s}, \boldsymbol{a})\right] - \mathbb{E}_{\boldsymbol{a} \sim q_{\boldsymbol{\theta}}(\cdot \mid \boldsymbol{s}), \, \boldsymbol{s} \sim \mu_{q_{\boldsymbol{\theta}}}(\cdot)}\left[r_{\boldsymbol{\theta}}(\boldsymbol{s}, \boldsymbol{a}) - \beta(\log q_{\boldsymbol{\theta}}(\boldsymbol{a} \mid \boldsymbol{s}) - \log p(\boldsymbol{a} \mid \boldsymbol{s}))\right] \geq$$

$$\alpha\left(\mathbb{E}_{\boldsymbol{s}, \boldsymbol{a} \sim \mathcal{D}}\left[\log q_{\boldsymbol{\theta}}(\boldsymbol{a} \mid \boldsymbol{s})\right] - \lambda \, \mathbb{D}_{\mathrm{KL}}[q_{\boldsymbol{\theta}}(\boldsymbol{w}) \mid\mid p(\boldsymbol{w})] + \mathbb{E}_{\boldsymbol{s}, \boldsymbol{a} \sim \mathcal{D}}\left[\log p(\boldsymbol{a} \mid \boldsymbol{s})\right]\right).$$

*where $\lambda = (\alpha - \beta)/\alpha$ and $\mathbb{E}_{\boldsymbol{s}, \boldsymbol{a} \sim \mathcal{D}}\left[\log p(\boldsymbol{a} \mid \boldsymbol{s})\right]$ is constant. The regression objective bounds the IRL objective for a worst-case on-policy state distribution $\mu_q(\boldsymbol{s})$, which motivates its scaling through $\beta$. If $\mathcal{D}$ has sufficient coverage, no RL finetuning or KL regularization is required, so $\beta = \alpha$ and $\lambda = 0$. If $\mathcal{D}$ does not have sufficient coverage, then let $\beta < \alpha$ so $\lambda > 0$ to regularize the BC fit and finetune the policy with RL accordingly with additional soft policy iteration steps.*

The proof is provided in Appendix K.3. In the tabular setting, it is straightforward for the cloned policy to reflect the prior distribution outside of the expert's data distribution. However, this behavior is harder to capture in the continuous setting where function approximation is typically adopted.

## 4 Stationary processes for continuous control policies

Maximum entropy methods used in reinforcement learning can be recast as a minimum relative entropy problem against a regularizing prior policy with a uniform action distribution, i.e., a prior $p(\boldsymbol{a} \mid \boldsymbol{s}) = \mathcal{U}_{\mathcal{A}}(\boldsymbol{a}) \, \forall \, \boldsymbol{s} \in \mathcal{S}$, where $\mathcal{U}_{\mathcal{A}}$ is the uniform distribution over $\mathcal{A}$. Achieving such a policy in the tabular setting is straightforward, as the policy can be updated independently for each state. However, such a policy construction is far more difficult for continuous states due to function approximation capturing correlations between states. To achieve the desired behavior, we can use stationary process theory (Definition 3) to construct an appropriate function space (Figure 3).

**Definition 3.** *(Stationary process, Cox and Miller [67]). A process $f : \mathcal{X} \to \mathcal{Y}$ is stationary if its joint distribution in $\boldsymbol{y} \in \mathcal{Y}$, $p(\boldsymbol{y}_1, \ldots, \boldsymbol{y}_n)$, is constant w.r.t. $\boldsymbol{x}_1, \ldots, \boldsymbol{x}_n \in \mathcal{X}$ and all $n \in \mathbb{N}_{>0}$.*

To approximate a stationary policy using function approximation, Gaussian process (GP) theory provides a means using features in a Gaussian linear model that defines a stationary process [62]. To approximate such feature spaces using neural networks, this can be achieved through a relatively wide final layer with a periodic activation function ($f_{\mathrm{per}}$), which can be shown to satisfy the stationarity property [68]. Refer to Appendix C for technical details. To reconcile this approach with prior work such as SAC, we use the predictive distribution of our Gaussian process in lieu of a network directly

predicting Gaussian moments. The policy is defined as $\boldsymbol{a} = \tanh(\boldsymbol{z}(\boldsymbol{s}))$, where $\boldsymbol{z}(\boldsymbol{s}) = \boldsymbol{W}\boldsymbol{\phi}(\boldsymbol{s})$. The weights are factorized row-wise $\boldsymbol{W} = [\boldsymbol{w}_1, \ldots, \boldsymbol{w}_{d_a}]^\top$, $\boldsymbol{w}_i = \mathcal{N}(\boldsymbol{\mu}_i, \boldsymbol{\Sigma}_i)$ to define a GP with independent actions. Using change-of-variables like SAC [17], the policy is expressed per-action as

$$q(a_i \mid \boldsymbol{s}) = \mathcal{N}\left(z_i; \boldsymbol{\mu}_i^\top \boldsymbol{\phi}(\boldsymbol{s}), \boldsymbol{\phi}(\boldsymbol{s})^\top \boldsymbol{\Sigma}_i \boldsymbol{\phi}(\boldsymbol{s})\right) \cdot \left| \det\left(\frac{\mathrm{d}a_i}{\mathrm{d}z_i}\right) \right|^{-1}, \quad \boldsymbol{\phi}(\boldsymbol{s}) = f_{\mathrm{per}}(\tilde{\boldsymbol{W}}\boldsymbol{\phi}_{\mathrm{mlp}}(\boldsymbol{s})).$$

$\tilde{\boldsymbol{W}}$ are weights drawn from a distribution, e.g., a Gaussian, that also characterizes the stochastic process and $\phi_{\mathrm{mlp}}$ is an arbitrary MLP. We refer to this heteroscedastic stationary model as HETSTAT.

Function approximation necessitates several additional practical implementation details of CSIL.

**Approximating and regularizing the critic.** Theorem 1 and Lemma 1 show that the log policy ratio is also a shaped critic. However, we found this model was not expressive enough for further policy evaluation. Instead, the critic is approximated using as a feedforward network and pre-trained after the policy using SARSA [8] and the squared Bellman error. For coherency, a useful inductive bias is to minimize the critic Jacobian w.r.t. the expert actions to approximate first-order optimality, i.e., $\min_\phi \mathbb{E}_{\boldsymbol{s},\boldsymbol{a}\sim\mathcal{D}}[\nabla_{\boldsymbol{a}}\mathcal{Q}_\phi(\boldsymbol{s},\boldsymbol{a})]$, as an auxiliary critic loss. For further discussion, see Section H in the Appendix. Pre-training and regularization are ablated in Figures 44 and 46 in the Appendix.

**Using the cloned policy as prior.** While a uniform prior is used in the CSIL reward throughout, in practice, we found that finetuning the cloned policy with this prior in the soft Bellman equation (Equation 3) leads to divergent policies. Replacing the maximum entropy regularization with KL regularization against the cloned policy, i.e., $\mathbb{D}_{\mathrm{KL}}[q_{\boldsymbol{\theta}}(\boldsymbol{a} \mid \boldsymbol{s}) \,||\, q_{\boldsymbol{\theta}_1}(\boldsymbol{a} \mid \boldsymbol{s})]$, mitigated divergent behavior. This regularization still retains a maximum entropy effect if the BC policy behaves like a stationary process. This approach matches prior work on KL-regularized RL from demonstrations, e.g., [69, 61].

**'Faithful' heteroscedastic regression loss.** To fit a parametric pseudo-posterior to the expert dataset, we use the predictive distribution for conditional density estimation [64] using heteroscedastic regression [63]. A practical issue with heteroscedastic regression with function approximators is the incorrect modeling of data as noise [70]. This can be overcome with a 'faithful' loss function and modelling construction [71], which achieves the desired minimization of the squared error in the mean and fits the predictive variance to model the residual errors. For more details, see Appendix D.

**Refining the coherent reward.** The HETSTAT network we use still approximates a stationary process, as shown in Figure 3. To ensure spurious reward values from approximation errors are not exploited during learning, it can be beneficial to refine, in a minimax fashion, the coherent reward with the additional data seen during learning. This minimax refinement both reduces the stationary approximation errors of the policy, while also improving the reward from the game-theoretic IRL perspective. For further details, intuition and ablations, see Appendix E, G and N.6 respectively.

Algorithm 1 with these additions can be found summarized in Algorithm 2 in Appendix I. An extensive ablation study of these adjustments can be found in Appendix N.

## 5 Experimental results

We evaluate CSIL against baseline methods on tabular and continuous state-action environments. The baselines are popular entropy-regularized imitation learning methods discussed in Section 2. Moreover, ablation studies are provided in Appendix N for the experiments in Section 5.2 and 5.3

| MDP | Variant | Expert | BC | Classifier | ME-IRL | GAIL | IQLearn | PPIL | CSIL |
|-----|---------|--------|------|-----------|--------|-------|---------|-------|-------|
| Dense | Nominal | 0.266 | 0.200 | 0.249 | 0.251 | 0.253 | 0.244 | 0.229 | **0.257** |
|  | Windy | 0.123 | 0.086 | 0.103 | **0.111** | 0.105 | 0.104 | 0.104 | 0.107 |
| Sparse | Nominal | 1.237 | 1.237 | **1.237** | 1.131 | **1.237** | **1.237** | **1.237** | **1.237** |
|  | Windy | 0.052 | 0.002 | **0.044** | 0.036 | 0.043 | 0.002 | 0.002 | **0.044** |

Table 1: Inverse optimal control, combining SPI and known dynamics, in tabular MDPs. The 'dense' MDP has a uniform initial state distribution and four goal states. The 'sparse' MDP has one initial state, one goal state and one forbidden state. The agents are trained on the nominal MDP. The 'windy' MDP has a random disturbance across all states to evaluate the robustness of the policy. CSIL performs well across all settings despite being the simpler algorithm relative to the IRL baselines.

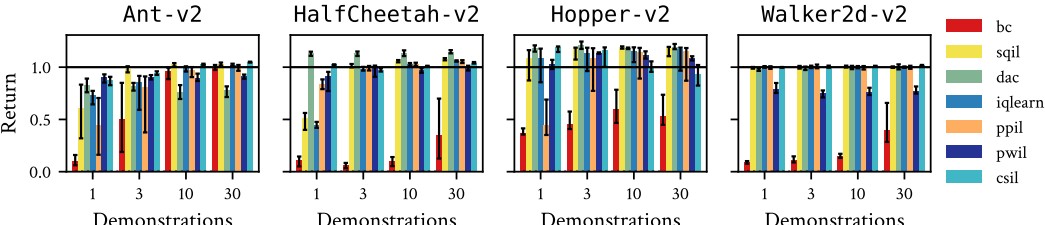

Figure 4: Normalized performance of CSIL against baselines for online imitation learning for MuJoCo Gym tasks. Uncertainty intervals depict quartiles over ten seeds. To assess convergence across seeds, performance is chosen using the highest 25th percentile of the episode return during learning.

## 5.1 Tabular inverse optimal control

We consider inverse optimal control in two tabular environments to examine the performance of CSIL without approximations. One ('dense') has a uniform initial distribution across the state space and several goal states, while the other ('sparse') has one initial state, one goal state, and one forbidden state with a large negative reward. Table 1 and Appendix M.1 shows the results for a nominal and a 'windy' environment, which has a fixed dynamics disturbance to test the policy robustness. The results show that CSIL is effective across all settings, especially the sparse environment where many baselines produce policies that are brittle to disturbances and, therefore, covariate shift. Moreover, CSIL produces value functions similar to GAIL (Figures 23 and 27) while being a simpler algorithm.

## 5.2 Continuous control from agent demonstrations

A standard benchmark of deep imitation learning is learning MuJoCo [72] Gym [73] and Adroit [74] tasks from agent demonstrations. We evaluate online and offline learning, where a fixed dataset is used in lieu of environment interactions [75]. Acme [76] was used to implement CSIL and baselines, and expert data was obtained using rlds [77] from existing sources [11, 74, 75]. Returns are normalized with respect to the reported expert and random performance [11] (Table 2).

**Online imitation learning.** In this setting, we used DAC (actor-critic GAIL), IQLearn, PPIL, SQIL, and PWIL as entropy-regularized imitation learning baselines. We evaluate on the standard benchmark of locomotion-based Gym tasks, using the SAC expert data generated by Orsini et al. [11]. In this setting, Figures 4 and 28 show CSIL closely matches the best baseline performance across environments and dataset sizes. We also evaluate on the Adroit environments, which involve manipulation tasks with a complex 27-dimensional robot hand [74]. In this setting, Figures 5 and 29 show that saddle-point methods struggle due to the instability of the optimization in high dimensions without careful regularization and hyperparameter selection [11]. In contrast, CSIL is very effective, matching or surpassing BC, highlighting the benefit of coherency for both policy initialization and improvement. In the Appendix, Figure 29 includes SAC from demonstrations (SACfD) [78, 74] as an oracle baseline with access to the true reward. CSIL exhibits greater sample efficiency than SACfD, presumably due to the BC initialization and the shaped reward, and often matches final performance. Figures 38, 39 and 40 in the Appendix show an ablation of IQLearn and PPIL with BC pre-training. We observe a fast unlearning of the BC policy due to the randomly initialized rewards, which was also observed by Orsini et al. [11], so any initial performance improvement is negligible or temporary.

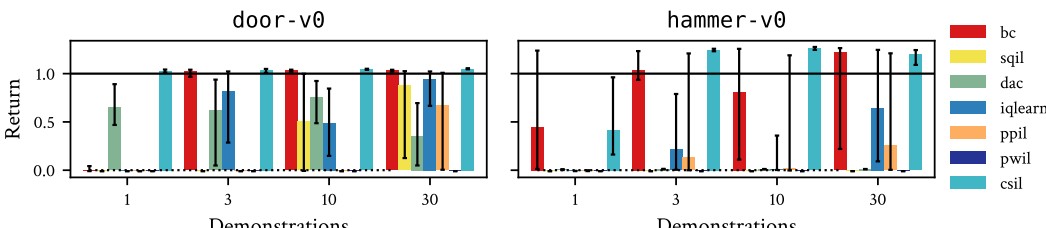

Figure 5: Normalized performance of CSIL against baselines for online imitation learning for Adroit tasks. Uncertainty intervals depict quartiles over ten seeds. To assess convergence across seeds, performance is chosen using the highest filtered 25th percentile of the return during learning.

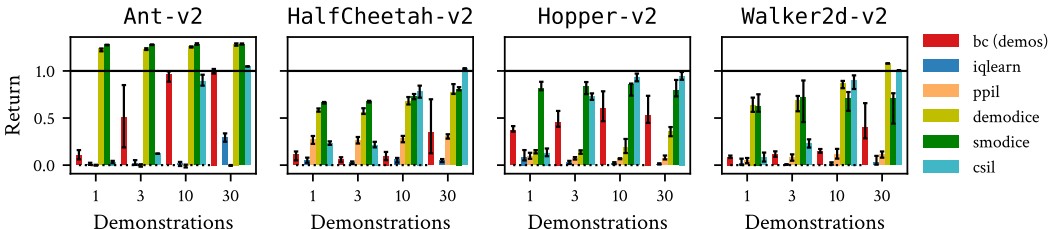

Figure 6: Normalized performance of CSIL against baselines for offline imitation learning for `Gym` tasks. Uncertainty intervals depict quartiles over ten seeds. To assess convergence across seeds, performance is chosen using the highest filtered 25th percentile of the episode return during learning.

**Offline imitation learning.** Another challenging setting for deep RL is *offline* learning, where the online interactions are replaced with a static 'supplementary' dataset. Note that prior works have used 'offline' learning to describe the setting where only the demonstrations are used for learning [41, 79, 42, 43], like in BC, such that the IL method becomes a form of regularized BC [80]. Standard offline learning is challenging primarily because the critic approximation can favor unseen actions, motivating appropriate regularization (e.g., [81]). We use the `full-replay` datasets from the `d4rl` benchmark [75] of the `Gym` locomotion tasks for the offline dataset, so the offline and demonstration data is not the same. We also evaluate SMODICE [82] and DemoDICE [83], two offline imitation learning methods which use state-based value functions, weighted BC policy updates, and a discriminator-based reward. We opt not to reproduce them in `acme` and use the original implementations. As a consequence, the demonstrations are from a different dataset, but are normalized appropriately. For further details on SMODICE and DemoDICE, see Appendix A and L.

Figures 6 and 30 demonstrate that offline learning is significantly harder than online learning, with no method solving the tasks with few demonstrations. This can be attributed to the lack of overlap between the offline and demonstration data manifesting as a sparse reward signal. However, with more demonstrations, CSIL can achieve reasonable performance. This performance is partly due to the strength of the initial BC policy, but CSIL can still demonstrate policy improvement in some cases (Figure 40). Figure 30 includes CQL [81] as an oracle baseline, which has access to the true reward function. For some environments, CSIL outperforms CQL with enough demonstrations. The DemoDICE and SMODICE baselines are both strong, especially for few demonstrations. However, performance does not increase so much with more demonstrations. Since CSIL could also be implemented with a state-based value function and a weighed BC policy update (e.g., [16]), further work is needed to determine which components matter most for effective offline imitation learning.

## 5.3 Continuous control from human demonstrations from states and images

As a more realistic evaluation, we consider learning robot manipulation tasks such as picking (`Lift`), pick-and-place (`PickPlaceCan`), and insertion (`NutAssemblySquare`) from random initializations and mixed-quality human demonstrations using the `robomimic` datasets [84], which also include image observations. The investigation of Mandlekar et al. [84] considered offline RL and various forms of BC with extensive model selection. Instead, we investigate the applicability of imitation learning, using online learning with CSIL as an alternative to BC model selection. One aspect of these tasks is that they have a sparse reward based on success, which can be used to define an absorbing

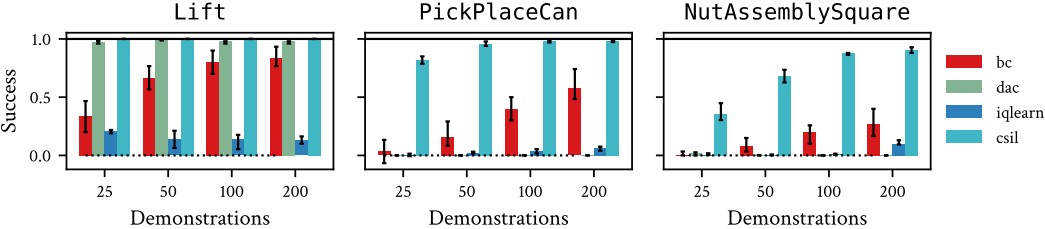

Figure 7: Average success rate over 50 evaluations for online imitation learning for `robomimic` tasks. Uncertainty intervals depict quartiles over ten seeds. To assess convergence across seeds, performance is chosen using the highest 25th percentile of the averaged success during learning.

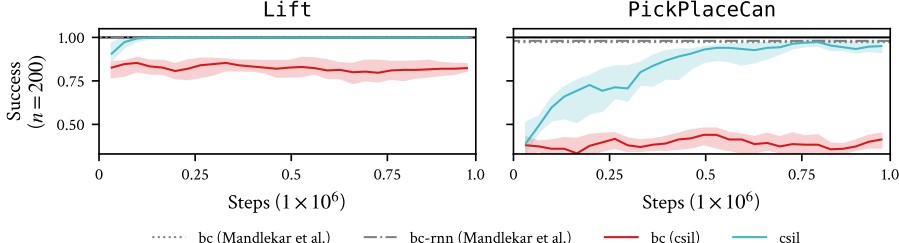

Figure 8: Average success rate over 50 evaluations for image-based online imitation learning for `robomimic` tasks. Uncertainty intervals depict quartiles over five seeds. Baseline results obtained from Mandlekar et al. [84]. BC (CSIL) denotes the performance of CSIL's initial policy for comparison.

state. As observed in prior work [40], in practice there is a synergy between absorbing states and the sign of the reward, where positive rewards encourage survival and negative rewards encourage minimum-time strategies. We found that CSIL was initially ill-suited to these goal-oriented tasks as the rewards are typically positive. In the same way that the AIRL reward can be designed for a given sign [11], we can design negative CSIL rewards by deriving an upper-bound of the log policy ratio and subtracting it in the reward. For further details, see Appendix F. An ablation study in Appendix N.9 shows that the negative CSIL reward matches or outperforms a constant negative reward, especially when given fewer demonstrations. These tasks are also more difficult due to the variance in the initial state, requiring much larger function approximators than in Section 5.2.

Figures 7 and 36 show the performance of CSIL and baselines, where up to 200 demonstrations are required to sufficiently solve the tasks. CSIL achieves effective performance and reasonable demonstration sample efficiency across all environments, while baselines such as DAC struggle to solve the harder tasks. Moreover, Figure 36 shows CSIL is again more sample efficient than SACfD.

Figure 8 shows the performance on CSIL when using image-based observations. A shared CNN torso is used between the policy, reward and critic, and the convolutional layers are frozen after the BC stage. CSIL was able to solve the simpler tasks in this setting, matching the model selection strategy of Mandlekar et al. [84], demonstrating its scalability. In the offline setting, Figure 37 shows state-based results with sub-optimal ('mixed human') demonstrations as supplementary dataset. While some improvement could be made on simpler tasks, on the whole it appears much harder to learn from suboptimal human demonstrations offline. This supports previous observations that human behaviour can be significantly different to that of RL agents, such that imitation performance is affected [11, 84].

## 6 Discussion

We have shown that 'coherency' is an effective approach to IL, combining BC with IRL-based finetuning by using a shaped learned reward for which the BC policy is optimal. We have demonstrated the effectiveness of CSIL empirically across a range of settings, particularly for high-dimensional tasks and offline learning, due to CSIL leveraging BC for initializing the policy and reward.

In Figure 35 of the Appendix, we investigate why baselines IQLearn and PPIL, both similar to CSIL, struggle in the high-dimensional and offline settings. Firstly, the sensitivity of the saddle-point optimization can be observed in the stability of the expert reward and critic values during learning. Secondly, we observe that the policy optimization does not effectively minimize the BC objective by fitting the expert actions. This suggests that these methods do not necessarily converge to a BC solution and lack the 'coherence' quality that CSIL has that allows it to refine BC policies.

A current practical limitation of CSIL is understanding when the initial BC policy is viable as a coherent reward. Our empirical results show that CSIL can solve complex control tasks using only one demonstration, where its BC policy is ineffective. However, if a demonstration-rich BC policy cannot solve the task (e.g., image-based `NutAssemblySquare`), CSIL also appears to be unable to solve the task. This suggests there is scope to improve the BC models and learning to make performance more consistent across more complex environments. An avenue for future work is to investigate the performance of CSIL with richer policy classes beyond MLPs, such as recurrent architectures and multi-modal action distributions, to assess the implications of the richer coherent reward and better model sub-optimal human demonstrations.

## Acknowledgments and Disclosure of Funding

We wish to thank Jost Tobias Springenberg, Markus Wulfmeier, Todor Davchev, Ksenia Konyushkova, Abbas Abdolmaleki and Matthew Hoffman for helpful discussions during the project. We would also like to thank Gabriel Dulac-Arnold and Sabela Ramos for help with setting up datasets and experiments, Luca Viano for help reproducing PPIL, and Martin Riedmiller, Markus Wulfmeier, Oleg Arenz, Davide Tateo, Boris Belousov, Michael Lutter and Gokul Swarmy for feedback on previous drafts. We also thank the wider Google DeepMind research and engineering teams for the technical and intellectual infrastructure upon which this work is built, in particular the developers of `acme`. Some baseline experiments were run on the computer cluster of the Intelligence Autonomous Systems group at TU Darmstadt, which is maintained by Daniel Palenicek and Tim Schneider. Joe Watson acknowledges the grant "Einrichtung eines Labors des Deutschen Forschungszentrum für Künstliche Intelligenz (DFKI) an der Technischen Universität Darmstadt" of the Hessian Ministry of Science and Art.

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

# A Extended related work

This section discusses the relevant prior literature in more breadth and depth.

Generative adversarial imitation learning (GAIL) [37] combines the game-theoretic IRL objective with causal entropy regularization and convex regularization of the reward. This primal optimization problem can be transformed into a dual objective, which has the interpretation of minimizing a divergence between the expert and imitator's stationary distributions, where the divergence corresponds to the choice of convex reward regularization. Choosing regularization that reduced the reward to a classification problem corresponds to the Jensen-Shannon (JS) divergence. While GAIL used on-policy RL, DAC [40] used off-policy actor-critic methods (e.g., SAC) in an ad hoc fashion for greater sample efficiency. ORIL [85] adopts a classification-based reward in the offline setting. Kostrikov et al. [40] also note that the GAIL classifier can be used in different forms to produce rewards with different forms. AIRL [65] highlight that the GAIL reward is shaped by the dynamics if the MDP following the shaping theory of Ng and Russell [5], and use additional approximators that disentangle the shaping terms. NAIL [79] replaces the minimax nature of adversarial learning with a more stable max-max approach using expectation maximization-like optimization on a lower-bound of the KL divergence between policies. Its reward function is defined, as with AIRL, as the log density ratio of the stationary distributions of the expert and policy, regularized by the causal entropy. Therefore, NAIL and CSIL have similar reward structures; however, while CSIL's reward can be obtained from BC and prior design, NAIL requires iterative density ratio estimation during learning. Ghasemipour et al. [39] investigate the performance of AIRL with different $f$-divergences and corresponding reward regularization beyond JS such as the forward and reverse KL divergence. Ghasemipour et al. [39] discuss the connection between the log density ratio, AIRL, GAIL and their proposed methods. Ghasemipour et al. [39] also connect BC to IRL, but from the divergence perspective rather than the likelihood perspective, as BC can be cast as the conditional KL divergence between the expert and agent policies.

A separate line of study uses the convex duality in a slightly different fashion for 'discounted distribution correction estimation' (DICE) methods. ValueDICE uses the Donsker-Varadhan representation to transform the KL minimization problem into an alternative form and a change of variables to represent the log ratio of the stationary distributions as with an implicit representation using the Bellman operator. However, this alternative form is still a saddle-point problem like GAIL, but the final objective is more amenable to off-policy optimization. DemoDICE [83] considers the offline setting and incorporates KL-regularization into the DICE framework, simplifying optimization. Due to the offline setting, the reward is obtained using a pre-trained classifier. Policy improvement reduces to weighted BC, like in many KL-regularized RL methods [86, 16]. SmoDICE [82] is closely related to DemoDICE, but focuses on matching the state distribution and, therefore, does not rely on expert actions, so the discriminator-based reward depends only on the state.

Inverse soft $Q$-learning (IQLearn) [42] combines the game-theoretic IRL objective, entropy-regularized actor-critic (SAC), convex reward regularization and implicit rewards to perform IRL using only the critic by maximizing the expert's implicit reward. While Garg et al. [42] claim their method is non-adversarial because it does not use a classifier, it is still derived from the game-theoretic objective. For continuous control, the IQLearn implementation requires shaping the implicit reward using online samples, making learning not so different to GAIL-based algorithms. Moreover, the implementation replaces the minimization of the initial value with the value evaluated for the expert samples. For SAC, in which the value function is computed by evaluating the critic with the current policy, this shapes the critic to have a local maximimum around the expert samples. We found this implementation detail crucial to reproduce the results for a few demonstrations and discuss further in Appendix H.

Proximal point imitation learning (PPIL) [43] combines the game-theoretic and linear programming forms of IRL and entropy-regularized actor-critic RL to derive an effective method with convergence guarantees. Its construction yields a single objective (the dual) for optimizing the reward and critic jointly in a more stable fashion, rather than alternating updates like in DAC, and unlike IQLearn, it has an explicit reward model. For continuous control, its implementation builds on SAC and IQLearn, including the aforementioned tricks, but with two key differences: an explicit reward function and minimization of the 'logistic', rather than squared, Bellman error [19].

Compared to the works discussed above, CSIL has two main distinctions. One is the reward hypothesis space in the form of the log policy ratio, derived from KL-regularized RL, as opposed to a classifier-

based reward or critic-as-classifier implicit reward. Secondly, the consequence of this coherent reward is that it can be pre-trained using BC, and the BC policy can be used as a reference policy. While BC pre-training could be incorporated into prior methods in an ad-hoc fashion, their optimization objectives do not guarantee that coherency is maintained. This is demonstrated in Figure 35, where IQLearn and PPIL combined with BC pre-training and KL regularization do not converge like CSIL.

Concurrent work, least-squares inverse $Q$-learning (LS-IQ) [87], addresses the undocumented regularization of IQLearn by minimizing a mixture divergence, which yields the desired regularization terms and better treatment of absorbing states. The treatment moves away from the entropy-regularized RL view and instead opts for explicit regularization critics.

Szot et al. [88], in concurrent work, use the bi-level optimization view of IRL to optimize the maximum entropy reward through an upper-level behavioral cloning loss using 'meta' gradients. This approach achieves coherency through the top-level loss while regularizing the learned reward. However, this approach is restricted to on-policy RL algorithms that directly use the reward rather than actor-critics methods that use a $Q$ function, making it less sample efficient.

Brantley et al. [24] propose disagreement-regularized imitation learning, which trains an ensemble of policies via BC and uses their predictive variance to define a proxy reward. This work shares the same motivation as CSIL in achieving coherency and tackling the covariate shift problem. Moreover, using ensembles is similar to CSIL's stationary policies, as both models are inspired by Bayesian modeling and provide uncertainty quantification. Unlike CSIL, the method is not motivated by reward shaping, and the reward is obtained by thresholding the predictive variance to $\pm 1$ rather than assume that the likelihood values are meaningful.

Taranovic et al. [89] derive an AIRL formalation from the KL divergence between expert and policy that results in a reward consisting of log policy ratios. However, they use a classifier to estimate these quantities and ignore some policy terms in the implementation.

Swamy et al. [90] use the BC policy for replay estimation [25] and a membership classifier to augment the expert dataset used during IRL.

'Noisy' BC [91] is tangentially related to CSIL, as it uses the BC log-likelihood as a proxy reward to finetune the BC policy using policy gradient updates when learning from demonstrations that are sub-optimal due to additive noise.

Soft $Q$-imitation learning (SQIL) [23] uses binary rewards for the demonstration and interaction transitions, bypassing the complexity of standard IRL. The benefit of this approach comes from the critic learning to overcome the covariate shift problem through the credit assignment. This approach has no theoretical guarantees, and the simplistic nature of the reward does not encourage stable convergence in practice.

Barde et al. [51] proposed adversarial soft advantage fitting (ASAF), an adversarial approach that constructs the classifier using two policies over trajectories or transitions. This approach is attractive as the classification step performs the policy update, so no policy evaluation step is required. While this approach has clear connections to regularized behavioral cloning, it's uncertain how the policy learns to overcome covariate shift without any credit assignment. Moreover, this method has no convergence guarantees.

CSIL adopts a Bayesian-style approach to its initial BC policy by specifying a prior policy but uses the predictive distribution rather than a likelihood to fit the demonstration data. This is in contrast to Bayesian IRL [92], where the prior is placed over the reward function as a means of reasoning about its ambiguity. As the reward is an unobserved abstract quantity, the exponentiated critic is adopted as a pseudo-likelihood, and approximate inference is required to estimate the posterior reward. Using a pseudo-likelihood means that Bayesian IRL and ME-IRL are not so different in practice, especially when using point estimates of the posterior [93].

## B Likelihoods and priors in imitation learning

The pseudo-posterior perspective in this work was inspired by van Campenhout and Cover [56], who identify that effective likelihoods arise from relevant functions that describe a distribution. With this in mind, we believe it is useful to see IRL as extending BC from a myopic, state-independent likelihood to one that encodes the causal, state-dependent nature of the problem as done ME-IRL, essentially incorporating the structure of the MDP into the regression problem [50]. This perspective has two key consequences. One is the importance of the prior, which provides important regularization. For example, Ziniu et al. [80] have previously shown that the ValueDICE does not necessarily outperform BC as previously reported but matches BC with appropriate regularization. Moreover, the MIMIC-EXP formulation of BC of Rajaraman et al. [25] proposes a BC policy with a stationary prior,

$$\pi(\boldsymbol{a} \mid \boldsymbol{s}) = \begin{cases} p_{\mathcal{D}}(\boldsymbol{a} \mid \boldsymbol{s}) \text{ if } \boldsymbol{s} \in \mathcal{D}, \text{ where } p_{\mathcal{D}} \text{ denotes conditional density estimation of } \mathcal{D}, \\ \mathcal{U}_{\mathcal{A}}(\boldsymbol{a}) \text{ otherwise.} \end{cases}$$

This work shows how policies like these can be implemented for continuous control using stationary architectures.

The second consequence is the open question of how to define effective likelihoods for imitation learning. ME-IRL and other prior works use feature matching, which, while an appropriate constraint, raises the additional question of which features are needed. Using a classifier in AIRL has proved highly effective, but it is not always straightforward to implement in practice without regularization and hyperparameter tuning. Much of IQLearn's empirical success for continuous tasks arises from shaping the likelihood via the critic through the expert and on-policy regularization terms.

This work addresses the likelihood design problem through coherence and the inversion analysis in Theorem 1, which informs the function-approximated critic regularization described in Section 4. An open question is how these design choices could be further improved, as the ablation study in Figure 46 shows that the benefit of the critic regularization is not evident for all tasks.

## C Parametric stationary processes

The effectiveness of CSIL is down to its stationary policy design. This section briefly summarizes the definition of stationary processes and how they can be approximated with MLPs following the work of Meronen et al. [68].

While typically discussed for temporal stochastic processes whose input is time, stationary processes (Definition 3) are those whose statistics do not vary with their input. In machine learning, these processes are useful as they allow us to design priors that are consistent per input but capture correlations (e.g., smoothness) *across* inputs, which can be used for regularization in regression and classification [62]. In the continuous setting, stationarity can be defined for linear models, i.e., $y = \boldsymbol{w}^{\top}\boldsymbol{\phi}(\boldsymbol{x})$, $\boldsymbol{w} \sim \mathcal{N}(\boldsymbol{\mu}, \boldsymbol{\Sigma})$, through their kernel function $\mathcal{K}(\boldsymbol{x}, \boldsymbol{x}') = \boldsymbol{\phi}(\boldsymbol{x})^{\top}\boldsymbol{\phi}(\boldsymbol{x}')$, the feature inner product. For stationary kernels, the kernel depends only on a relative shift $\boldsymbol{r} \in \mathbb{R}^d$ between inputs, i.e. $\mathcal{K}(\boldsymbol{r}) = \boldsymbol{\phi}(\boldsymbol{x})^{\top}\boldsymbol{\phi}(\boldsymbol{x} + \boldsymbol{r})$. By combining the kernel with the weight prior of the linear model, we can define the covariance function, e.g. $\mathcal{C}(\boldsymbol{r}) = \boldsymbol{\phi}(\boldsymbol{x})^{\top}\boldsymbol{\Sigma}\,\boldsymbol{\phi}(\boldsymbol{x} + \boldsymbol{r})$. Boschner's theorem [94] states that the covariance function of a stationary process can be represented as the Fourier transform of a positive finite measure. If the measure has a density, it is known as the spectral density $S(\boldsymbol{\omega})$, and it is a Fourier dual of the covariance function if the necessary conditions apply, known as the Wiener-Khinchin theorem [95],

$$\mathcal{C}(\boldsymbol{r}) = \frac{1}{(2\pi)^d} \int_{\mathbb{R}^d} S(\boldsymbol{\omega}) \exp(i\boldsymbol{\omega}^{\top}\boldsymbol{r}) \,\mathrm{d}\boldsymbol{r}, \quad S(\boldsymbol{\omega}) = \int_{\mathbb{R}^d} \mathcal{C}(\boldsymbol{r}) \exp(-i\boldsymbol{\omega}^{\top}\boldsymbol{r}) \,\mathrm{d}\boldsymbol{\omega}. \qquad (9)$$

To use this theory to design stationary function approximators, we can use the Wiener-Khintchin theorem to design function approximators that produce a Monte Carlo approximation of stationary kernels. Firstly, we consider only the last layer of the MLP, such that it can be viewed as a linear model. Secondly, Meronen et al. [68] describe several periodic activation functions that produce products of the form $\exp(i\omega^{\top}\boldsymbol{r})$, such that the inner product of the whole feature space performs a Monte Carlo approximation of the integral. We implemented sinusoidal, triangular, and periodic ReLU activations (see Figure 3 of Meronen et al. [68]) as a policy architecture hyperparameter. Thirdly, the input into the stationary kernel approximation should be small (e.g., $< 20$), as the stationary behavior

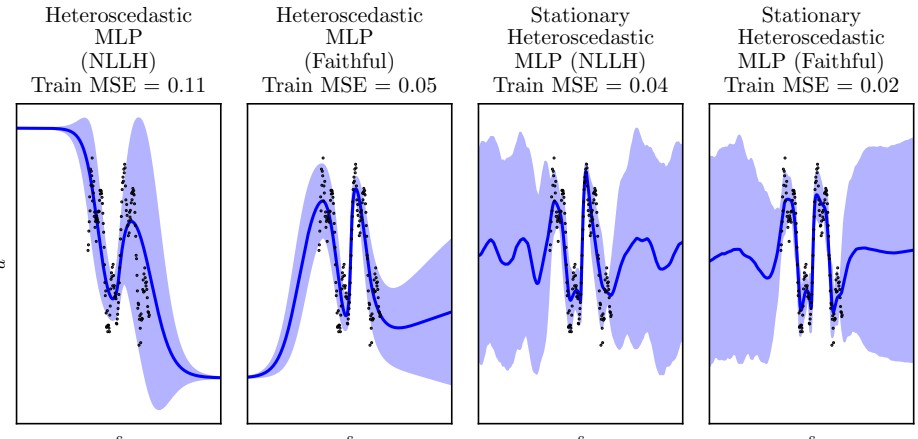

Figure 9: We repeat the regression problem from Figure 3 to highlight the effect of 'faithful' heteroscedastic regression. The NLLH objective has a tendency to model 'signal' as noise, which is most clearly seen in the heteroscedastic MLP. The faithful heteroscedastic loss reduces the mean prediction error significantly without affecting uncertainty quantification for both models.

is dictated by $r$, and this distance becomes less meaningful in higher dimensions. This means the MLP feature space that feeds into the last layer should be small or have a 'bottleneck' architecture to compress the internal representation. Finally, we require a density to represent the spectral density, which also defines the nature of the stationary process. We use a Gaussian, but other distributions such as a Student-$t$ or Cauchy could be used if 'rougher' processes are desired. In summary, the last layer activations take the form $f_p(\boldsymbol{W}\boldsymbol{\phi}(\boldsymbol{x}))$, where $f_p$ is the periodic activation function of choice, $\boldsymbol{W}$ are weights sampled from the distribution of choice and $\boldsymbol{\phi}(\boldsymbol{x})$ is an arbitrary (but bottlenecked) MLP feature space of choice.

Related work combining stationary kernel approximations with MLPs have constrained the feature space with a residual architecture and spectral normalization to bound the upper and lower Lipschitz constant [96], to prevent these features overfitting. We found we did not need this to achieve approximate stationarity (e.g., Figure 3), which avoided the need for input-dependent policy architectures which may lead to a risk of underfitting.

A crucial practical detail is that the sampled weights $\boldsymbol{W}$ should be trained and not kept constant, as the theory suggests. In practice, we did not observe the statistics of these weights changing significantly during training, while training this weight layer reduced underfitting enough for substantially more effective BC performance.

## D    Faithful heteroscedastic regression

This section summarizes the approach of Stirn et al. [71]. The faithful objective combines the mean squared error (MSE) and negative log-likelihood NLLH loss with a careful construction of the heteroscedastic model. The loss is

$$\mathcal{L}(\mathcal{D}, \boldsymbol{\theta}) = \mathbb{E}_{\boldsymbol{s},\boldsymbol{a}\sim\mathcal{D}} \left[ (\boldsymbol{a} - \boldsymbol{\mu_\theta}(\boldsymbol{s}))^2 - \log \tilde{q}(\boldsymbol{a} \mid \boldsymbol{s}) \right], \quad \tilde{q}(\boldsymbol{a} \mid \boldsymbol{s}) = \mathcal{N}(\mathrm{sg}(\boldsymbol{\mu_\theta}(\boldsymbol{s})), \boldsymbol{\Sigma_\theta}(\boldsymbol{s})), \quad (10)$$

where $\mathrm{sg}(\cdot)$ denotes the stop gradient operator. Moreover, for the models 'shared torso' (e.g., features), the gradient is stopped between the torso and the predictive variance so that the features are only trained to satisfy the MSE objective. Figure 9 illustrates the impact of the alternative loss function and model adjustments. Note that the faithful objective is not strictly necessary for BC or CSIL and is mitigating potential under-fitting previously observed empirically. The issue of modeling 'signal' as noise when using heteroscedastic function approximators is poorly understood and depends on the model architecture, initialization, and the dataset considered. While motivated by 1D toy examples, it was observed during development that the faithful objective is beneficial for CSIL's downstream performance.

# E  Reward refinement and function-space relative entropy estimation

In the function approximation setting, we refine the reward model with additional non-expert samples. Motivated by Theorem 2, we maximize the objective using samples from the replay buffers

$$\mathcal{J}_r(\boldsymbol{\theta}) = \mathbb{E}_{\boldsymbol{s},\boldsymbol{a}\sim\mathcal{D}}\left[\alpha \log \frac{q_{\boldsymbol{\theta}}(\boldsymbol{a}\mid\boldsymbol{s})}{p(\boldsymbol{a}\mid\boldsymbol{s})}\right] - \mathbb{E}_{\boldsymbol{s},\boldsymbol{a}\sim\rho_\pi}\left[\alpha \log \frac{q_{\boldsymbol{\theta}}(\boldsymbol{a}\mid\boldsymbol{s})}{p(\boldsymbol{a}\mid\boldsymbol{s})}\right]. \tag{11}$$

If the behavior policy $\pi$ is $q_{\boldsymbol{\theta}}(\boldsymbol{a}\mid\boldsymbol{s})$, we can view the second term as a Monte Carlo estimation of the relative KL divergence between $q$ and $p$. This estimator can suffer greatly from bias, as the KL should always be non-negative. To counteract this, we construct an unbiased, positively-constrained estimator following Schulman [97], using $\mathbb{E}_{\boldsymbol{x}\sim q(\cdot)}[R(\boldsymbol{x})^{-1}] = \int p(\boldsymbol{x})\,\mathrm{d}\boldsymbol{x} = 1$ and $\log(x) \leq x - 1$,

$$\mathbb{D}_{\mathrm{KL}}[q(\boldsymbol{x}) \mid\mid p(\boldsymbol{x})] = \mathbb{E}_{\boldsymbol{x}\sim q(\cdot)}[\log R(\boldsymbol{x})] = \mathbb{E}_{\boldsymbol{x}\sim q(\cdot)}[R(\boldsymbol{x})^{-1} - 1 + \log R(\boldsymbol{x})],$$

This estimator results in replacing the second term of Equation 11 with

$$\mathbb{E}_{\boldsymbol{s},\boldsymbol{a}\sim\rho_\pi}\left[r_{\boldsymbol{\theta}}(\boldsymbol{s},\boldsymbol{a})\right] \to \mathbb{E}_{\boldsymbol{s},\boldsymbol{a}\sim\rho_\pi}\left[r_{\boldsymbol{\theta}}(\boldsymbol{s},\boldsymbol{a}) - \alpha + \alpha \exp(-r_{\boldsymbol{\theta}}(\boldsymbol{s},\boldsymbol{a})/\alpha)\right].$$

This objective is maximized concurrently with policy evaluation, using samples from the replay buffer. This reward refinement is similar to behavioral cloning via function-space variational inference (e.g., [98, 99]), enforcing the prior using the prior predictive distribution rather than the prior weight distribution.

In practice, the samples are taken from a replay buffer of online and offline data, so the action samples are taken from a different distribution to $q_{\boldsymbol{\theta}}$. However, we still found the adjusted objective effective for reward regularization. Without it, the saddle-point refinement was less stable. Appendix G illustrates this refinement in a simple setting. Figure 45 ablates this component, where its effect is most beneficial when there are fewer demonstrations.

# F  Bounding the coherent reward

The coherent reward $\alpha(\log q(\boldsymbol{a}\mid\boldsymbol{s}) - \log p(\boldsymbol{a}\mid\boldsymbol{s}))$ can be upper bounded if $p(\boldsymbol{a}\mid\boldsymbol{s}) > 0$ whenever $q(\boldsymbol{a}\mid\boldsymbol{s}) > 0$. The bound is policy dependent. We use the bound in the continuous setting, where we use a tanh-transformed Gaussian policy. We add a small bias term $\sigma_{\min}^2$ to the predictive variance in order to define the Gaussian likelihood upper bound. Inconveniently, the tanh change of variables term in the log-likelihood $-\sum_{i=1}^{d_a} \log(1 - \tanh^2(u))$ [17] has no upper bound in theory, as the 'latent' action $u \in [-\infty, \infty]$, but in the Acme implementation this log-likelihood is bounded for numerical stability due to the inverse tanh operation. For action dimension $d_a$ and uniform prior across action range $[-1,1]^{d_a}$ and $\alpha = d_a^{-1}$, the upper bound is $r(\boldsymbol{s},\boldsymbol{a}) \leq \frac{1}{d_a}(-0.5 d_a \log 2\pi\sigma_{\min}^2 + c + d_a \log 2) = -0.5 \log \pi\sigma_{\min}^2/2 + \tilde{c}$, where $c, \tilde{c}$ depends on the tanh clipping term.

# G  Visualizing reward refinement

To provide intuition on Theorem 2 and connect it to reward refinement (Appendix E), we build on the regression example from Figure 3 by viewing it as a continuous contextual bandit problem, i.e., a single-state MDP with no dynamics. Outside of the demonstration state distribution, we desire that the reward is uniformly zero, as we have no data to inform action preference. Figure 10 shows the result of applying CSIL and reward refinement to this MDP, where the state distribution is uniform. The 'desired' BC fit exhibits a coherent reward with the desired qualities, and the HETSTAT policy approximates this desired policy. The heteroscedastic MLP exhibits undesirable out-of-distribution (OOD) behavior, where arbitrary regions have strongly positive or negative rewards. Reward refinement improves the coherent reward of the heteroscedastic MLP to be more uniform OOD, which in turn makes the regression fit look more like a stationary policy. This result reinforces Theorem 2, which shows that KL-regularized BC is connected to a lower bound on an entropy-regularized game-theoretic IRL objective that uses the coherent reward.

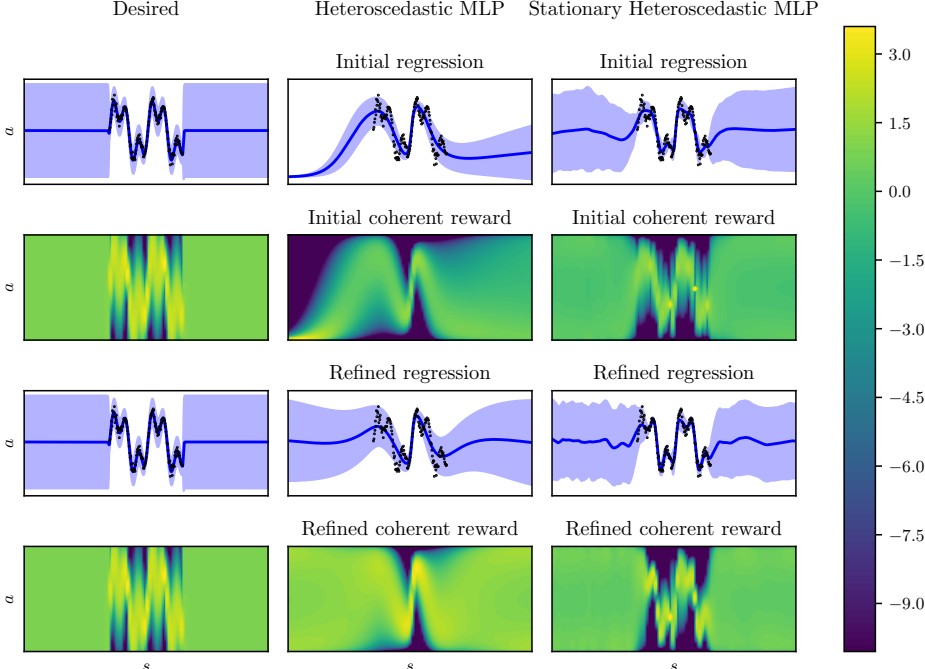

Figure 10: We extend the regression problem from Figure 3 to highlight the effect of reward refinement detailed in Section E from the BC and IRL perspective. Refining the coherent reward with a minimax objective results in a BC policy that appears more stationary, as seen in the heteroscedatic MLP. The stationary heteroscedastic policy is also refined, but to a lesser degree, as it's approximately stationary by construction. The colourmap is shared between contour plots, and rewards are clipped at -10 to improve the visual colour range. Regression uncertainty is one standard deviation.

## H  Critic regularization

For the function approximation setting, CSIL requires an auxiliary loss term during learning to incorporate coherency. The full critic optimization problem is $\min_\phi \mathcal{J}_\mathcal{Q}(\phi)$ for objective

$$\mathcal{J}_\mathcal{Q}(\phi) = \mathbb{E}_{s,a\sim\mathcal{B}}[(\mathcal{Q}_\phi(s,a) - \mathcal{Q}^*(s,a))^2] + \mathbb{E}_{s,a\sim\mathcal{D}}[|\nabla_a\mathcal{Q}_\phi(s,a)|^2], \qquad (12)$$

where $\mathcal{Q}^*$ denotes the target $Q$ values and $\mathcal{B}$ denotes the replay buffer, that combines demonstrations $\mathcal{D}$ and additional samples. The motivation is to shape the critic such that the demonstration actions are first-order optimal by minimizing the squared Frobenius norm of the action Jacobian.

IQLearn and PPIL adopt similar, but less explicit, regularization. Equation 10 of Garg et al. [42] is

$$\max_\phi \mathbb{E}_{s,a\sim\mathcal{B}}\left[f\left(\mathcal{Q}_\phi(s,a) - \gamma\,\mathbb{E}_{s'\sim\mathcal{P}(\cdot|s,a)}[\mathcal{V}_\phi^\pi(s')]\right)\right] - (1-\gamma)\mathbb{E}_{s\sim\mu_0}[\mathcal{V}_\phi^\pi(s)],$$

where $\mathcal{V}_\phi^\pi(s) = \mathcal{Q}(s,a')$, $a' \sim \pi(\cdot \mid s)$ and $f$ is the concave function defining the implicit reward regularization. In the implementation [100], the 'v0' objective variant replaces the initial distribution $\mu_0$ with the demonstration state distribution. This trick is deployed to improve regularization without affecting optimality (Section 5.3, Kostrikov et al. [41]). If the policy matches the expert actions with its mode, this term shapes a local maximum by minimizing the critic at action samples around the

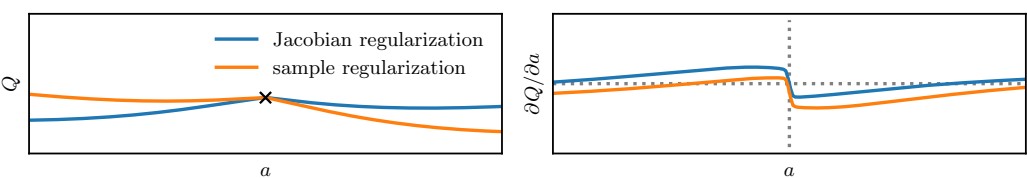

Figure 11: Illustrating the two forms of critic regularization on a toy 1D problem for critic $\mathcal{Q}(a)$ with regression target ✕. Jacobian regularization minimizes the squared gradient and corresponds to CSIL. Sample regularization minimizes sampled nearby actions and corresponds to IQLearn and PPIL. Both approaches shape the critic to have zero gradient around the target action, denoted by the dotted lines. A standard two-layer MLP with 256 units and ELU actions is used for approximation.

**Algorithm 2:** Off-policy coherent soft imitation learning with function approximation

---

**Data:** Expert demonstrations $\mathcal{D}$, initial temperature $\alpha$, refinement temperature, $\beta$,
parametric policy class $q_{\boldsymbol{\theta}}(\boldsymbol{a} \mid \boldsymbol{s})$, prior policy $p(\boldsymbol{a} \mid \boldsymbol{s})$, regression regularizer $\Psi$, total steps $T$
**Result:** $q_{\boldsymbol{\theta}_N}(\boldsymbol{a} \mid \boldsymbol{s})$, matching or improving the initial policy $q_{\boldsymbol{\theta}_1}(\boldsymbol{a} \mid \boldsymbol{s})$
             `// Pretrain coherent reward and critic`
Train initial policy from demonstrations, $\boldsymbol{\theta}_1 = \arg\max_{\boldsymbol{\theta}} \mathbb{E}_{\boldsymbol{s}, \boldsymbol{a} \sim \mathcal{D}}[\log q_{\boldsymbol{\theta}}(\boldsymbol{a} \mid \boldsymbol{s}) - \Psi(\boldsymbol{\theta})]$;
Define fixed shaped coherent reward, $\tilde{r}_{\boldsymbol{\theta}_1}(\boldsymbol{s}, \boldsymbol{a}) = \alpha(\log q_{\boldsymbol{\theta}_1}(\boldsymbol{a} \mid \boldsymbol{s}) - \log p(\boldsymbol{a} \mid \boldsymbol{s}))$;
Pretrain critic with SARSA on $\mathcal{D}$ using $\tilde{r}_{\boldsymbol{\theta}_1}$.
**for** $t = 1 \to T$ **do**       `// finetune policy with reinforcement learning`
 |  Interact with environment $\boldsymbol{s}_{t+1} = \text{Env}(\boldsymbol{s}_t, \boldsymbol{a}_t)$, $\boldsymbol{a}_t \sim q_{\boldsymbol{\theta}_t}(\cdot \mid \boldsymbol{s}_t)$, store in replay buffer $\mathcal{B}$;
 |  Optimize reward and critic using $\mathcal{J}_{\mathcal{Q}}(\phi)$ and $\mathcal{J}_r(\boldsymbol{\theta})$ (Equations 12, 11) on minibatch from $\mathcal{B}$;
 |  Update policy optimizing $\mathcal{J}_{\pi}(\boldsymbol{\theta})$ (Equation 13) on minibatch from $\mathcal{B}$;
**end**

---

expert. However, if the policy does not accurately match the expert actions, the effect of this term is harder to reason about. In Figure 11, we illustrate these two forms from critic regularization on a toy example, demonstrating that both can produce local maximums in the critic.

# I   Practical algorithm with function approximation

For deep imitation learning, CSIL adopts SAC for the SPI step in Algorithm 1, summarized in Algorithm 2. The critic objective for both pre-training and policy evaluation is the squared Bellman error. The policy objective $\max_{\boldsymbol{\theta}} \mathcal{J}_{\pi}(\boldsymbol{\theta})$ samples both the demonstrations and replay buffer,

$$\mathcal{J}_{\pi}(\boldsymbol{\theta}) = E_{\boldsymbol{a} \sim q_{\boldsymbol{\theta}}(\cdot \mid \boldsymbol{s}), \, \boldsymbol{s} \sim \mathcal{B} \cup \mathcal{D}} \left[ \mathcal{Q}(\boldsymbol{s}, \boldsymbol{a}) - \beta(\log q_{\boldsymbol{\theta}}(\cdot \mid \boldsymbol{s}) - \log q_{\boldsymbol{\theta}_1}(\cdot \mid \boldsymbol{s})) \right]. \tag{13}$$

The policy and demonstration data are combined equally. We did not experiment with tuning this ratio. Compared to SAC, we found we did not have to train multiple critics, but we did need target networks for the critics to stabilize learning. The critic and reward objectives were also evaluated on the policy and expert data each update. Moreover, like IQLearn and PPIL, we kept $\beta$ constant and did not use an adaptive strategy.

# J   A divergence minimization perspective

Many imitation learning algorithms, such as GAIL and IQLearn, are derived from minimizing a divergence between the agent's state-action distribution and the expert's. While CSIL is derived through policy inversion, it also has a divergence minimization perspective, albeit with two minimization steps rather than one.

The first step involves recognising that maximum likelihood BC fit minimizes the forward KL divergence between the demonstration distribution $p_{\mathcal{D}}(\boldsymbol{s}, \boldsymbol{a}) = p_{\mathcal{D}}(\boldsymbol{a} \mid \boldsymbol{s}) \, p_{\mathcal{D}}(\boldsymbol{s})$ and the policy,

$$\boldsymbol{\theta}_1 = \arg\min_{\boldsymbol{\theta}} \mathbb{E}_{\boldsymbol{s} \sim p_{\mathcal{D}}(\cdot)}[\mathbb{D}_{\text{KL}}[p_{\mathcal{D}}(\boldsymbol{a} \mid \boldsymbol{s}) \mid\mid q_{\boldsymbol{\theta}}(\boldsymbol{a} \mid \boldsymbol{s})]] = \arg\max_{\boldsymbol{\theta}} \mathbb{E}_{\boldsymbol{s}, \boldsymbol{a} \sim \mathcal{D}}[\log q_{\boldsymbol{\theta}}(\boldsymbol{a} \mid \boldsymbol{s})].$$

Using the forward KL means that the unknown log-likelihood of the data distribution is not required, and the policy can be trained using supervised learning and without environment interaction.

The second step involves rearranging the KL-regularized RL objective into two KL divergences,

$$\boldsymbol{\theta}_* = \arg\max_{\boldsymbol{\theta}} \mathbb{E}_{\boldsymbol{a} \sim q_{\boldsymbol{\theta}}(\cdot \mid \boldsymbol{s}), \, \boldsymbol{s} \sim \mu_{\pi_{\boldsymbol{\theta}}}(\cdot)} \left[ \alpha \log \frac{q_{\boldsymbol{\theta}_1}(\boldsymbol{a} \mid \boldsymbol{s})}{p(\boldsymbol{a} \mid \boldsymbol{s})} - \beta \log \frac{q_{\boldsymbol{\theta}}(\boldsymbol{a} \mid \boldsymbol{s})}{p(\boldsymbol{a} \mid \boldsymbol{s})} \right],$$

$$= \arg\min_{q_{\boldsymbol{\theta}}} \mathbb{E}_{\boldsymbol{s} \sim \mu_{q_{\boldsymbol{\theta}}}(\cdot)}[\alpha \, \mathbb{D}_{\text{KL}}[q_{\boldsymbol{\theta}}(\boldsymbol{a} \mid \boldsymbol{s}) \mid\mid q_{\boldsymbol{\theta}_1}(\boldsymbol{a} \mid \boldsymbol{s})] - (1 - \beta/\alpha) \, \mathbb{D}_{\text{KL}}[q_{\boldsymbol{\theta}}(\boldsymbol{a} \mid \boldsymbol{s}) \mid\mid p(\boldsymbol{a} \mid \boldsymbol{s})]].$$

Note that since $\beta \leq \alpha$, then $(1 - \beta/\alpha) \geq 0$. The left-hand term minimizes the reverse KL between the agent and the expert, using the BC policy as a proxy. This is beneficial because the reverse conditional KL requires evaluating policy $q_{\boldsymbol{\theta}}$ on the environment. Moreover, the reverse KL requires log-likelihoods of the target distribution, which is why the BC policy is used as a proxy of the demonstration distribution. The right-hand term incorporates the coherent reward and encourages the policy to stay in states where the policy differs from the prior. As $q_{\boldsymbol{\theta}}$ should be close to the BC policy due to the left-hand term, the coherent reward shaping from Definition 1 should hold.

In this formulation of the objective, it is also clear to see that if $\beta = \alpha$, no RL finetuning will occur, and $q_{\boldsymbol{\theta}}$ will simply mimic the BC policy $q_{\boldsymbol{\theta}_1}$ due to the left-hand term.

# K  Theoretical results

This section provides derivations and proofs for the theoretical results in the main text.

## K.1  Relative entropy regularization for reinforcement learning

This section re-derives KL-regularized RL and ME-IRL in a shared notation for the reader's reference.

**Reinforcement learning.**  We consider the discounted, infinite-horizon setting with a hard KL constraint on the policy update with bound $\epsilon$,

$$\max_q \mathbb{E}_{\boldsymbol{s}_{t+1} \sim \mathcal{P}(\cdot | \boldsymbol{s}_t, \boldsymbol{a}_t), \, \boldsymbol{a}_t \sim q(\cdot | \boldsymbol{s}_t), \, \boldsymbol{s}_0 \sim \mu_0(\cdot)}[\textstyle\sum_t \gamma^t r(\boldsymbol{s}_t, \boldsymbol{a}_t)] \quad \text{s.t.} \quad \mathbb{E}_{\boldsymbol{s} \sim \nu_q(\cdot)}[\mathbb{D}_{\mathrm{KL}}[q(\boldsymbol{a} \mid \boldsymbol{s}) \mid\mid p(\boldsymbol{a} \mid \boldsymbol{s})]] \leq \epsilon.$$

This constrained objective is transcribed into the following Lagrangian objective following Van Hoof et al. [15], using the discounted stationary joint distribution $d(\boldsymbol{s}, \boldsymbol{a}) = q(\boldsymbol{a} \mid \boldsymbol{s}) \nu(\boldsymbol{s}) = (1-\gamma)\rho(\boldsymbol{s}, \boldsymbol{a})$,

$$\mathcal{L}(d, \lambda_1, \lambda_2, \lambda_3) = (1 - \gamma)^{-1} \int_{\mathcal{S} \times \mathcal{A}} d(\boldsymbol{s}, \boldsymbol{a}) \, r(\boldsymbol{s}, \boldsymbol{a}) \, \mathrm{d}\boldsymbol{s} \, \mathrm{d}\boldsymbol{a} + \lambda_1 \left( 1 - \int_{\mathcal{S} \times \mathcal{A}} d(\boldsymbol{s}, \boldsymbol{a}) \, \mathrm{d}\boldsymbol{s} \, \mathrm{d}\boldsymbol{a} \right)$$

$$+ \int_{\mathcal{S}} \lambda_2(\boldsymbol{s}') \left( \int_{\mathcal{S} \times \mathcal{A}} \gamma \, \mathcal{P}(\boldsymbol{s}' \mid \boldsymbol{s}, \boldsymbol{a}) \, d(\boldsymbol{s}, \boldsymbol{a}) \, \mathrm{d}\boldsymbol{s} \, \mathrm{d}\boldsymbol{a} + (1 - \gamma)\mu_0(\boldsymbol{s}') - \int_{\mathcal{A}} d(\boldsymbol{s}', \boldsymbol{a}') \, \mathrm{d}\boldsymbol{a}' \right) \, \mathrm{d}\boldsymbol{s}'$$

$$+ \lambda_3 \left( \epsilon - \int_{\mathcal{S}} \nu(\boldsymbol{s}) \int_{\mathcal{A}} q(\boldsymbol{a} \mid \boldsymbol{s}) \log \frac{q(\boldsymbol{a} \mid \boldsymbol{s})}{p(\boldsymbol{a} \mid \boldsymbol{s})} \, \mathrm{d}\boldsymbol{a} \, \mathrm{d}\boldsymbol{s} \right),$$

where $\lambda_2$ is a function for an infinite-dimensional constraint. Solving for $\partial \mathcal{L}(q, \lambda_1, \lambda_2, \lambda_3)/\partial d = 0$,

$$q(\boldsymbol{a} \mid \boldsymbol{s}) \propto \exp(\alpha^{-1}(\underbrace{r(\boldsymbol{s}, \boldsymbol{a}) + \gamma \, \mathbb{E}_{\boldsymbol{s}' \sim \mathcal{P}(\cdot | \boldsymbol{s}, \boldsymbol{a})}[\mathcal{V}(\boldsymbol{s}')]}_{\mathcal{Q}(\boldsymbol{s}, \boldsymbol{a})} - \mathcal{V}(\boldsymbol{s}))) \, p(\boldsymbol{a} \mid \boldsymbol{s}),$$

with $\alpha = (1 - \gamma)\lambda_3$ and $\lambda_2(\boldsymbol{s}) = (1 - \gamma)\mathcal{V}(\boldsymbol{s})$, where $\mathcal{V}$ can be interpreted as the soft value function,

$$\mathcal{V}(\boldsymbol{s}) = \alpha \log \int_{\mathcal{A}} \exp \left( \frac{1}{\alpha} \, \mathcal{Q}(\boldsymbol{s}, \boldsymbol{a}) \right) \, p(\boldsymbol{a} \mid \boldsymbol{s}) \, \mathrm{d}\boldsymbol{a} \quad \text{as} \quad \int_{\mathcal{A}} q(\boldsymbol{a} \mid \boldsymbol{s}) \, \mathrm{d}\boldsymbol{a} = 1.$$

To obtain the more convenient lower bound of the soft Bellman equation, we combine importance sampling and Jensen's inequality to the soft Bellman equation

$$\mathcal{V}(\boldsymbol{s}) = \alpha \log \int_{\mathcal{A}} \exp \left( \frac{1}{\alpha} \, \mathcal{Q}(\boldsymbol{s}, \boldsymbol{a}) \right) \frac{p(\boldsymbol{a} \mid \boldsymbol{s})}{q(\boldsymbol{a} \mid \boldsymbol{s})} \, q(\boldsymbol{a} \mid \boldsymbol{s}) \, \mathrm{d}\boldsymbol{a},$$

$$\geq \mathbb{E}_{\boldsymbol{a} \sim q(\cdot | \boldsymbol{s})} \left[ \mathcal{Q}(\boldsymbol{s}, \boldsymbol{a}) - \alpha \log \frac{q(\boldsymbol{a} \mid \boldsymbol{s})}{p(\boldsymbol{a} \mid \boldsymbol{s})} \right].$$

This derivation is for the 'hard' KL constraint. In many implementations, including ours, the constraint is 'softened' where $\lambda_3$ and, therefore, $\alpha$ is a fixed hyperparameter.

**Inverse reinforcement learning.**  As mentioned in Section 2 and Definition 2, ME-IRL is closely related to KL-regularize RL, but with the objective and a constraint reversed,

$$\min_q \mathbb{E}_{\boldsymbol{s} \sim \nu_q(\cdot)}[\mathbb{D}_{\mathrm{KL}}[q(\boldsymbol{a} \mid \boldsymbol{s}) \mid\mid p(\boldsymbol{a} \mid \boldsymbol{s})]] \quad \text{s.t.} \quad \mathbb{E}_{\boldsymbol{s}, \boldsymbol{a} \sim \rho_q}[\boldsymbol{\phi}(\boldsymbol{s}, \boldsymbol{a})] = \mathbb{E}_{\boldsymbol{s}, \boldsymbol{a} \sim \mathcal{D}}[\boldsymbol{\phi}(\boldsymbol{s}, \boldsymbol{a})].$$

The constrained optimization is, therefore, transcribed into a similar Lagrangian,

$$\mathcal{L}(d, \lambda_1, \lambda_2, \boldsymbol{\lambda_3}) = \int_{\mathcal{S}} \nu(\boldsymbol{s}) \int q(\boldsymbol{a} \mid \boldsymbol{s}) \log \frac{q(\boldsymbol{a} \mid \boldsymbol{s})}{p(\boldsymbol{a} \mid \boldsymbol{s})} \, \mathrm{d}\boldsymbol{a} \, \mathrm{d}\boldsymbol{s} + \lambda_1 \left( 1 - \int_{\mathcal{S} \times \mathcal{A}} d(\boldsymbol{s}, \boldsymbol{a}) \, \mathrm{d}\boldsymbol{s} \, \mathrm{d}\boldsymbol{a} \right)$$

$$+ \int_{\mathcal{S}} \lambda_2(\boldsymbol{s}') \left( \int_{\mathcal{S} \times \mathcal{A}} \gamma \, \mathcal{P}(\boldsymbol{s}' \mid \boldsymbol{s}, \boldsymbol{a}) \, d(\boldsymbol{s}, \boldsymbol{a}) \, \mathrm{d}\boldsymbol{s} \, \mathrm{d}\boldsymbol{a} + (1 - \gamma)\mu_0(\boldsymbol{s}') - \int_{\mathcal{A}} d(\boldsymbol{s}', \boldsymbol{a}') \, \mathrm{d}\boldsymbol{a}' \right) \, \mathrm{d}\boldsymbol{s}'$$

$$+ \boldsymbol{\lambda_3}^{\top} \left( (1 - \gamma)^{-1} \int_{\mathcal{S} \times \mathcal{A}} d(\boldsymbol{s}, \boldsymbol{a}) \, \boldsymbol{\phi}(\boldsymbol{s}, \boldsymbol{a}) \, \mathrm{d}\boldsymbol{s} \, \mathrm{d}\boldsymbol{a} - \mathbb{E}_{\boldsymbol{s}, \boldsymbol{a} \sim \mathcal{D}}[\boldsymbol{\phi}(\boldsymbol{s}_t, \boldsymbol{a}_t)] \right).$$

The consequence of this change is that the likelihood temperature is now implicit in the reward model $r(\boldsymbol{s}, \boldsymbol{a}) = (1 - \gamma)^{-1} \boldsymbol{\lambda_3}^{\top} \boldsymbol{\phi}(\boldsymbol{s}, \boldsymbol{a})$ due to the Lagrange multiplier weights, this reward model is used in the soft Bellman equation instead of the true reward. The reward model is jointly optimized to minimize the apprenticeship error by matching the feature expectation.

## K.2 Proof for Theorem 1

**Theorem 1.** *(KL-regularized policy improvement inversion). Let $p$ and $q_\alpha$ be the prior and pseudo-posterior policy given by posterior policy iteration (Equation 4). The critic can be expressed as*

$$\mathcal{Q}(\boldsymbol{s}, \boldsymbol{a}) = \alpha \log \frac{q_\alpha(\boldsymbol{a} \mid \boldsymbol{s})}{p(\boldsymbol{a} \mid \boldsymbol{s})} + \mathcal{V}_\alpha(\boldsymbol{s}), \quad \mathcal{V}_\alpha(\boldsymbol{s}) = \alpha \log \int_\mathcal{A} \exp\left(\frac{1}{\alpha}\mathcal{Q}(\boldsymbol{s}, \boldsymbol{a})\right) p(\boldsymbol{a}|\boldsymbol{s}) \, \mathrm{d}\boldsymbol{a}. \quad (6)$$

*Substituting into the KL-regularized Bellman equation lower-bound from Equation 2,*

$$r(\boldsymbol{s}, \boldsymbol{a}) = \alpha \log \frac{q_\alpha(\boldsymbol{a} \mid \boldsymbol{s})}{p(\boldsymbol{a} \mid \boldsymbol{s})} + \mathcal{V}_\alpha(\boldsymbol{s}) - \gamma \, \mathbb{E}_{\boldsymbol{s}' \sim \mathcal{P}(\cdot|\boldsymbol{s}, \boldsymbol{a})}\left[\mathcal{V}_\alpha(\boldsymbol{s}')\right]. \quad (7)$$

*The $\mathcal{V}_\alpha(\boldsymbol{s})$ term is the 'soft' value function. We assume $q_\alpha(\boldsymbol{a} \mid \boldsymbol{s}) = 0$ whenever $p(\boldsymbol{a} \mid \boldsymbol{s}) = 0$.*

*Proof.* Equation 6 is derived by rearranging the posterior policy update in Equation 4. Equation 7 in derived by substituting the critic expression from Equation 6 into the KL-regularized Bellman equation (Equation 3),

$$\underbrace{\alpha \log \frac{q_\alpha(\boldsymbol{a} \mid \boldsymbol{s})}{p(\boldsymbol{a} \mid \boldsymbol{s})} + \mathcal{V}_\alpha(\boldsymbol{s})}_{\mathcal{Q}(\boldsymbol{s}, \boldsymbol{a})} =$$

$$r(\boldsymbol{s}, \boldsymbol{a}) + \gamma \, \mathbb{E}_{\boldsymbol{s}' \sim \mathcal{P}(\cdot|\boldsymbol{s}, \boldsymbol{a}), \, \boldsymbol{a}' \sim q_\alpha(\cdot|\boldsymbol{s}')} \left[ \underbrace{\alpha \log \frac{q_\alpha(\boldsymbol{a}' \mid \boldsymbol{s}')}{p(\boldsymbol{a}' \mid \boldsymbol{s}')} + \mathcal{V}_\alpha(\boldsymbol{s}')}_{\mathcal{Q}(\boldsymbol{s}', \boldsymbol{a}')} - \alpha \log \frac{q_\alpha(\boldsymbol{a}' \mid \boldsymbol{s}')}{p(\boldsymbol{a}' \mid \boldsymbol{s}')} \right].$$

$\square$

## K.3 Proof for Theorem 2

Theorem 2 relies on the data processing inequality (Lemma 3) to connect the function-space KL seen in CSIL with the weight-space KL used in regularized BC.

**Assumption 1.** *(Realizability) The optimal policy can be expressed by the parametric policy $q_{\boldsymbol{\theta}}(\boldsymbol{a}|\boldsymbol{s})$.*

**Lemma 2.** *(Coherent policy optimality). The optimal policy for a KL-regularized RL problem with coherent reward $\tilde{r}(\boldsymbol{s}, \boldsymbol{a}) = \alpha(\log q(\boldsymbol{a} \mid \boldsymbol{s}) - \log p(\boldsymbol{a} \mid \boldsymbol{s}))$ and temperature $\alpha$ is $q(\boldsymbol{a} \mid \boldsymbol{s})$.*

*Proof.* This result arises from the origin of the coherent reward from policy inversion and reward shaping (Theorem 1, Lemma 1). Moreover, considering the entropy-augmented reward $r_{\alpha, \pi}(\boldsymbol{s}, \boldsymbol{a}) = r(\boldsymbol{s}, \boldsymbol{a}) - \alpha(\log \pi(\boldsymbol{a} \mid \boldsymbol{s}) - \log p(\boldsymbol{a} \mid \boldsymbol{s}))$, when $r$ is the coherent reward $\tilde{r}$ and $\pi = q$, $\tilde{r}_{\alpha, q}(\boldsymbol{s}, \boldsymbol{a}) = 0 \, \forall \boldsymbol{s} \in \mathcal{S}, \, \boldsymbol{a} \in \mathcal{A}$ so $q(\boldsymbol{a} \mid \boldsymbol{s})$ is a solution to a soft policy iteration. $\square$

**Lemma 3.** *(Data processing inequality, Wu [101], Theorem 4.1). For two stochastic processes parameterized with finite random variables $\boldsymbol{w} \sim p(\cdot)$, $\boldsymbol{w} \in \mathcal{W}$ and shared hypothesis spaces $p(\boldsymbol{y} \mid \boldsymbol{x}, \boldsymbol{w})$, the conditional $f$-divergence in function space at points $\boldsymbol{X} \in \mathcal{X}^L, L \in \mathbb{Z}_+$. is upper bounded by the $f$-divergence in parameter space,*

$$\mathbb{D}_f[q(\boldsymbol{w}) \mid\mid p(\boldsymbol{w})] = \mathbb{D}_f[q(\boldsymbol{Y}, \boldsymbol{w} \mid \boldsymbol{X}) \mid\mid p(\boldsymbol{Y}, \boldsymbol{w} \mid \boldsymbol{X})] \geq \mathbb{D}_f[q(\boldsymbol{Y} \mid \boldsymbol{X}) \mid\mid p(\boldsymbol{Y} \mid \boldsymbol{X})].$$

**Theorem 2.** *(Coherent inverse reinforcement learning as KL-regularized behavioral cloning). A KL-regularized game-theoretic IRL objective, with policy $q_{\boldsymbol{\theta}}(\boldsymbol{a} \mid \boldsymbol{s})$ and coherent reward parameterization $r_{\boldsymbol{\theta}}(\boldsymbol{s}, \boldsymbol{a}) = \alpha(\log q_{\boldsymbol{\theta}}(\boldsymbol{a} \mid \boldsymbol{s}) - \log p(\boldsymbol{a} \mid \boldsymbol{s}))$ where $\alpha \geq 0$, is lower bounded by a scaled KL-regularized behavioral cloning objective and a constant term when the optimal policy, posterior, and prior share a hypothesis space $p(\boldsymbol{a} \mid \boldsymbol{s}, \boldsymbol{w})$ and finite parameters $\boldsymbol{w}$,*

$$\mathbb{E}_{\boldsymbol{s}, \boldsymbol{a} \sim \mathcal{D}}\left[r_{\boldsymbol{\theta}}(\boldsymbol{s}, \boldsymbol{a})\right] - \mathbb{E}_{\boldsymbol{a} \sim q_{\boldsymbol{\theta}}(\cdot|\boldsymbol{s}), \, \boldsymbol{s} \sim \mu_{q_{\boldsymbol{\theta}}}(\cdot)}\left[r_{\boldsymbol{\theta}}(\boldsymbol{s}, \boldsymbol{a}) - \beta(\log q_{\boldsymbol{\theta}}(\boldsymbol{a} \mid \boldsymbol{s}) - \log p(\boldsymbol{a} \mid \boldsymbol{s}))\right] \geq$$

$$\alpha\left(\mathbb{E}_{\boldsymbol{s}, \boldsymbol{a} \sim \mathcal{D}}\left[\log q_{\boldsymbol{\theta}}(\boldsymbol{a} \mid \boldsymbol{s})\right] - \lambda \, \mathbb{D}_{\mathrm{KL}}[q_{\boldsymbol{\theta}}(\boldsymbol{w}) \mid\mid p(\boldsymbol{w})] + \mathbb{E}_{\boldsymbol{s}, \boldsymbol{a} \sim \mathcal{D}}\left[\log p(\boldsymbol{a} \mid \boldsymbol{s})\right]\right).$$

*where $\lambda = (\alpha - \beta)/\alpha$ and $\mathbb{E}_{\boldsymbol{s}, \boldsymbol{a} \sim \mathcal{D}}\left[\log p(\boldsymbol{a} \mid \boldsymbol{s})\right]$ is constant. The regression objective bounds the IRL objective for a worst-case on-policy state distribution $\mu_q(\boldsymbol{s})$, which motivates its scaling through $\beta$. If $\mathcal{D}$ has sufficient coverage, no RL finetuning or KL regularization is required, so $\beta = \alpha$ and $\lambda = 0$. If $\mathcal{D}$ does not have sufficient coverage, then let $\beta < \alpha$ so $\lambda > 0$ to regularize the BC fit and finetune the policy with RL accordingly with additional soft policy iteration steps.*

| Task | Random return | Expert return |
|------|--------------|---------------|
| `HalfCheetah-v2` | -282 | 8770 |
| `Hopper-v2` | 18 | 2798 |
| `Walker2d-v2` | 1.6 | 4118 |
| `Ant-v2` | -59 | 5637 |
| `Humanoid-v2` | 123 | 9115 |
| `door-expert-v0` | -56 | 2882 |
| `hammer-expert-v0` | -274 | 12794 |

Table 2: Expert and random policy returns used to normalize the performance for `Gym` and `Adroit` tasks. Taken from Orsini et al. [11].

*Proof.* We begin with the KL-regularized game-theoretic IRL objective (c.f. Equation 1 [17]),

$$\mathcal{J}(r, \pi, \beta) = \mathbb{E}_{\boldsymbol{s}, \boldsymbol{a} \sim \mathcal{D}}\left[r(\boldsymbol{s}, \boldsymbol{a})\right] - \mathbb{E}_{\boldsymbol{s}, \boldsymbol{a} \sim \rho_\pi}\left[r(\boldsymbol{s}, \boldsymbol{a}) - \beta \log \frac{\pi(\boldsymbol{a} \mid \boldsymbol{s})}{p(\boldsymbol{a} \mid \boldsymbol{s})}\right].$$

Substituting the coherent reward from Theorem 1, the objective becomes

$$\mathcal{J}(\boldsymbol{\theta}, \pi, \beta) = \mathbb{E}_{\boldsymbol{s}, \boldsymbol{a} \sim \mathcal{D}}\left[\alpha \log \frac{q_{\boldsymbol{\theta}}(\boldsymbol{a} \mid \boldsymbol{s})}{p(\boldsymbol{a} \mid \boldsymbol{s})}\right] - \mathbb{E}_{\boldsymbol{s}, \boldsymbol{a} \sim \rho_\pi}\left[\alpha \log \frac{q_{\boldsymbol{\theta}}(\boldsymbol{a} \mid \boldsymbol{s})}{p(\boldsymbol{a} \mid \boldsymbol{s})} - \beta \log \frac{\pi(\boldsymbol{a} \mid \boldsymbol{s})}{p(\boldsymbol{a} \mid \boldsymbol{s})}\right].$$

Due to the realizablity assumption (Assumption 1), the optimal policy $\pi_{\boldsymbol{\theta}_*}$ can be obtained through maximum likelihood estimation $\boldsymbol{\theta}_* = \arg\max_{\boldsymbol{\theta}} \mathbb{E}_{\boldsymbol{s}, \boldsymbol{a} \sim \mathcal{D}}[\log q_{\boldsymbol{\theta}}(\boldsymbol{s}, \boldsymbol{a})]$. With this assumption, the left-hand term performs maximum likelihood estimation of the optimal policy. Therefore, combined with the coherent reward which is derived from policy inversion, we replace the inner policy optimization of $\pi$ with the maximum likelihood solution of $q_{\boldsymbol{\theta}}$, resulting in a simpler objective $\tilde{\mathcal{J}}$ in terms of only $\boldsymbol{\theta}$, which matches the objective $\mathcal{J}$ exactly when $\beta = \alpha$ and $\pi = q_{\boldsymbol{\theta}}$ (Lemma 2). We consider $\tilde{\mathcal{J}}$ with a small negative perturbation $\beta = \alpha - \delta\alpha$, $\delta\alpha \geq 0$ to examine the effect of the right-term term,

$$\tilde{\mathcal{J}}(\boldsymbol{\theta}) = \mathcal{J}(\boldsymbol{\theta}, q_{\boldsymbol{\theta}}, \alpha - \delta\alpha) = \mathbb{E}_{\boldsymbol{s}, \boldsymbol{a} \sim \mathcal{D}}\left[\alpha \log \frac{q_{\boldsymbol{\theta}}(\boldsymbol{a} \mid \boldsymbol{s})}{p(\boldsymbol{a} \mid \boldsymbol{s})}\right] - \mathbb{E}_{\boldsymbol{a} \sim q_{\boldsymbol{\theta}}(\cdot \mid \boldsymbol{s}), \boldsymbol{s} \sim \mu_{q_{\boldsymbol{\theta}}}}\left[\delta\alpha \log \frac{q_{\boldsymbol{\theta}}(\boldsymbol{a} \mid \boldsymbol{s})}{p(\boldsymbol{a} \mid \boldsymbol{s})}\right],$$

The right-hand term contains a conditional KL divergence, which we can replace with an upper-bound using the data processing inequality from Lemma 3 and the weight-space KL. This weight-space KL is constant w.r.t. the state distribution expectation and we introduce the scale factor $\lambda = \delta\alpha/\alpha$. The resulting lower-bound objective is proportional to the regularized behavioral cloning objective proposed in Equation 8,

$$\tilde{\mathcal{J}}(\boldsymbol{\theta}) \geq \alpha(\mathbb{E}_{\boldsymbol{s}, \boldsymbol{a} \sim \mathcal{D}}\left[\log q_{\boldsymbol{\theta}}(\boldsymbol{a} \mid \boldsymbol{s})\right] - \lambda \mathbb{D}_{\mathrm{KL}}[q_{\boldsymbol{\theta}}(\boldsymbol{w}) \mid\mid p(\boldsymbol{w})] + C), \quad C = \mathbb{E}_{\boldsymbol{s}, \boldsymbol{a} \sim \mathcal{D}}[-\log p(\boldsymbol{a} \mid \boldsymbol{s})].$$

$\square$

The intuition from Theorem 2 is that the KL regularization during BC achieves reward shaping similar to the IRL saddle-point solution when using the coherent reward. The lower bound's 'tightness' depends on the data processing inequality and is difficult to reason about. Therefore, this result demonstrates that KL-regularized BC provides useful reward shaping from the game-theoretic perspective, rather than provide a tight lower bound, To reinforce this result, we visualize the effect in Appendix G.

## L  Implementation and experimental details

Our codebase for this work is open-sourced and is available at github.com/google-deepmind/csil.

The tabular experiments are implemented in `jax` automatic differentiation and linear algebra library [102]. The continuous state-action experiments are implemented in `acme`, again using `jax` and its ecosystem [103]. One feature of `acme` is distributed training, where we use four concurrent actors to speed up the wall clock learning time. IQLearn and PPIL were re-implemented based on the open-source implementations as reference [100, 104]. CSIL, IQLearn, and PPIL shared a combined

implementation as 'soft' imitation learning algorithms due to their common structure and to facilitate ablation studies. Baselines DAC, SQIL, PWIL, SACfD, and CQL are implemented in `acme`. We used a stochastic actor for learning and a deterministic actor (i.e., evaluating the mean action) for evaluation.

The DEMODICE and SMODICE offline imitation learning baselines were implemented using the original implementations [105, 106]. This meant that the expert demonstrations were obtained from `d4rl` rather than Orsini et al. [11], but the returns were normalized to the `d4rl` returns for consistency. We note that there is a non-negligible discrepancy between the `d4rl` and Orsini et al. [11] return values for the `MuJoCo` tasks, but in the imitation setting, comparing normalized values should be a meaningful metric.

Due to the common use of SAC as the base RL algorithm, most implementations shared common standard hyperparameters. The policy and critic networks were comprised of two layers with 256 units and ELU activations. Learning rates were 3e-4, the batch size was 256, and the target network smoothing coefficient was 0.005. One difference to standard RL implementations is the use of `layernorm` in the critic, which is adopted in some algorithm implementations in `acme` and was found to improve performance. For the inverse temperature $\alpha$, for most methods, we used the adaptive strategy based on a lower bound on the policies entropy. IQLearn and PPIL both have a fixed temperature strategy that we tuned in the range $[1.0, 0.3, 0.1, 0.03, 0.01]$. CSIL also has a fixed temperature strategy but is applied to the KL penalty. In practice, they are very similar, so we swept the same range. We also investigated a hard KL bound and adaptive temperature strategy, but there was no performance improvement to warrant the extra complexity. CSIL's HETSTAT policies had an additional bottleneck layer of 12 units and a final layer of 256 units with stationary activations. For the `Gym` tasks, triangular activations were used. For `robomimic` tasks, periodic ReLU activations were used. IQLearn, PPIL, and CSIL all have single critics rather than dual critics, while the other baselines had dual critics to match the standard SAC implementation. We tried dual critics to stabilize IQLearn and PPIL, but this did not help. CSIL's reward finetuning learning rate was 1e-3. As the reward model was only being finetuned with non-expert samples, we did not find this hyperparameter to be very sensitive. For `robomimic`, larger models with two layers of 1024 units were required . The HETSTAT policies increased accordingly with a 1024 unit final layer and 48 units for the bottleneck layer.

For DAC, we used the hyperparameters and regularizers recommended by Orsini et al. [11], including spectral normalization for the single hidden layer discriminator with 64 units. IQLearn has a lower actor learning rate of 3e-5, and the open-source implementation has the option of different value function regularizers per task [100]. We used 'v0' regularization for all environments, which minimizes the averaged value function at the expert samples. In the original IQLearn experiments, this regularizer is tuned per environment [100]. PPIL shares the same smaller actor learning rate of 3e-5 and the open-source implementation also uses 20 critic updates for every actor update, which we maintained. The implementation of PPIL borrows the same value function regularization as IQLearn, so we used 'v0' regularization for all environments [104]. For SACfD, we swept the demo-to-replay ratio across $[0.1, 0.01]$. For `Adroit`, the ratio was 0.01, and for `robomimic` it was 0.1.

The major implementation details of CSIL are discussed in Sections 4, C, D and E. The pre-training hyperparameters are listed in Table 4. While the critic pre-training is not too important for performance (Figure 44), appropriate policy pre-training is crucial. Instead of training a regularization

| Experiment | Algorithm | | | | | |
|---|---|---|---|---|---|---|
| | DAC | PWIL | SQIL | IQLearn | PPIL | CSIL |
| Online `Gym` | $\mathbb{H}(\pi)$ | $\mathbb{H}(\pi)$ | $\mathbb{H}(\pi)$ | 0.01 | 0.01 | 0.01 |
| Offline `Gym` | $\mathbb{H}(\pi)$ | $\mathbb{H}(\pi)$ | $\mathbb{H}(\pi)$ | 0.03 | 0.03 | 0.1 |
| Online `Humanoid-v2` | $\mathbb{H}(\pi)$ | $\mathbb{H}(\pi)$ | $\mathbb{H}(\pi)$ | 0.01 | 0.01 | 0.01 |
| Online `Adroit` | $\mathbb{H}(\pi)$ | $\mathbb{H}(\pi)$ | $\mathbb{H}(\pi)$ | 0.03 | 0.03 | 0.1 |
| Online `robomimic` | $\mathbb{H}(\pi)$ | - | - | 0.1 | - | 0.1 |
| Offline `robomimic` | - | - | - | - | - | 1.0 |
| Image-based `robomimic` | - | - | - | - | - | 0.3 |

Table 3: The entropy regularization hyperparameter value used across methods and environments. Values refer to a fixed temperature while $\mathbb{H}(\pi)$ refers to the adaptive temperature scheme, which ensures the policy entropy is at least equal to $-d_a$, where $d_a$ is the dimension of the action space.

| Experiment | Pre-training hyperparameters | | | |
|---|---|---|---|---|
| | $n\,(\pi)$ | $\lambda\,(\pi)$ | $n\,(Q)$ | $\lambda\,(Q)$ |
| Online and offline `Gym` | 25000 | 1e-3 | 5000 | 1e-3 |
| Online `Humanoid-v2` | 500 | 1e-3 | 5000 | 1e-3 |
| Online `Adroit` | 25000 | 1e-4 | 5000 | 1e-3 |
| Online and offline `robomimic` | 50000 | 1e-4 | 2000 | 1e-3 |
| Image-based `robomimic` | 25000 | 1e-4 | 2000 | 1e-3 |

Table 4: CSIL's pre-training hyperparameters for the policy ($\pi$) and critic ($Q$), listing the learning rate ($\lambda$) and number of iterations ($n$).

hyperparameter, we opted for early stopping. As a result, there is a trade-off between BC accuracy and policy entropy. In the online setting, if the policy is trained too long, the entropy is reduced too far, and there is not enough exploration for RL. As a result, the policy pre-training hyperparameters need tuning when action dimension and dataset size change significantly to ensure the BC fit is sufficient for downstream performance. For `robomimic`, we used negative CSIL rewards scaled by 0.5 to ensure they lie around $[-1, 0]$. For vision-based `robomimic`, the CNN torso was an impala-style [107] `Acme ResNetTorso` down to a `LayerNorm` MLP with 48 output units, ELU activations, orthogonal weight initialization, and final tanh activation. Mandlekar et al. [84] used a larger `ResNet18`. As in Mandlekar et al. [84], image augmentation was achieved with random crops from 84 down to 76. Pixel values were normalized to [-0.5, 0.5]. For memory reasons, the batch size was reduced to 128. The implementations of IQLearn, PPIL and CSIL combined equally-sized mini-batches from the expert and non-expert replay buffers for learning the reward, critic and policy. We did not investigate tuning this ratio.

Regarding computational resources, the main experiments were run using `acme`, which implements distributed learning. As a result, our 'learner' (policy evaluation and improvement) runs on a single TPU v2. We ran four actors to interact with the environment. Depending on the algorithm, there were also one or more evaluators. For vision-based tasks, we used A100 GPUs for the vision-based policies.

# M Experimental results

This section includes additional experimental results from Section 5. Videos of the simulation studies are available at joemwatson.github.io/csil.

## M.1 Inverse optimal control

This section shows the converged value function, stationary distribution, and 'windy' stationary distribution for the tabular setting results from Table 1. 'Oracle classifier' constructs an offline classifier-based reward separating the states and actions with- and without demonstration data, in the spirit of methods such as SQIL and ORIL [85]. No optimization was conducted on the windy environment, just evaluation to assess the policy robustness to disturbance. We use the Viridis colour map, so yellow regions have a larger value than blue regions.

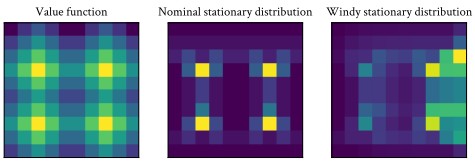

Figure 12: Expert on the dense MDP.

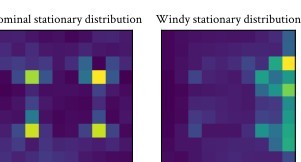

Figure 13: BC on the dense MDP.

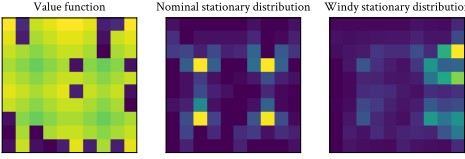

Figure 14: Oracle classifier on the dense MDP.

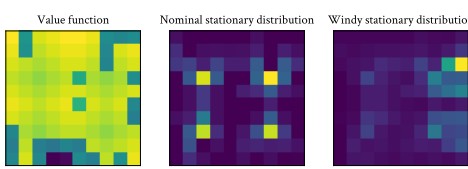

Figure 15: GAIL on the dense MDP.

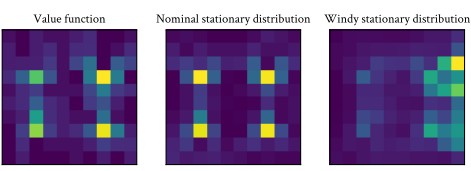

Figure 16: IQLearn on the dense MDP.

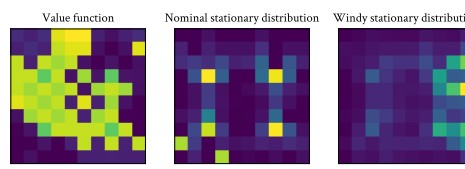

Figure 17: PPIL on the dense MDP.

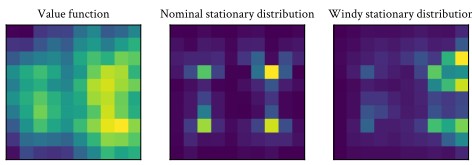

Figure 18: ME-IRL on the dense MDP.

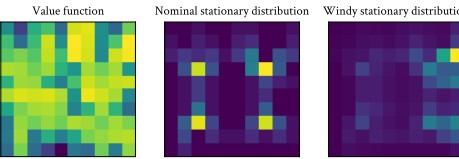

Figure 19: CSIL on the dense MDP.

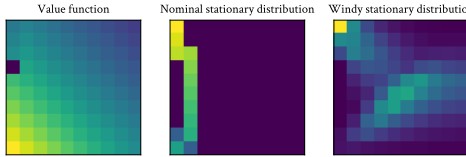

Figure 20: Expert on the sparse MDP.

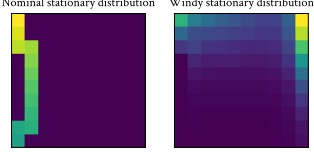

Figure 21: BC on the sparse MDP.

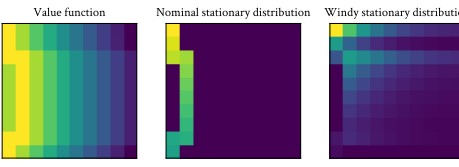

Figure 22: Oracle classifier on the sparse MDP.

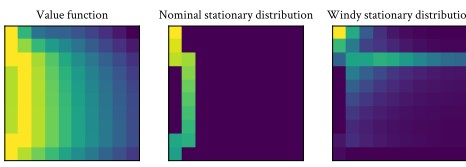

Figure 23: GAIL on the sparse MDP.

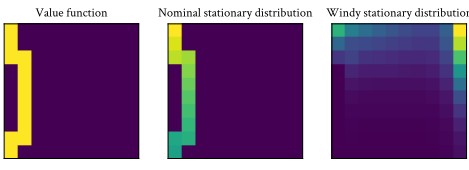

Figure 24: IQLearn on the sparse MDP.



Figure 25: PPIL on the sparse MDP.

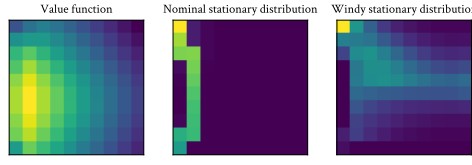

Figure 26: ME-IRL on the sparse MDP.

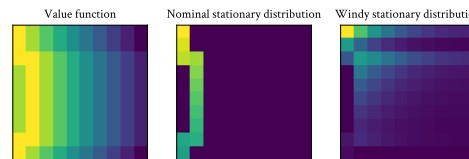

Figure 27: CSIL on the sparse MDP.

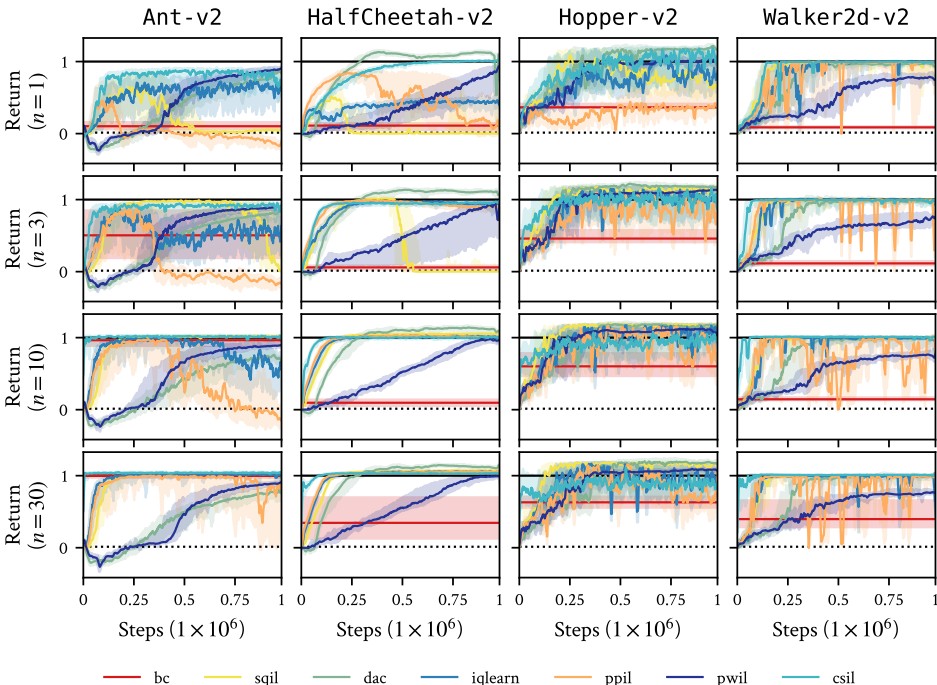

Figure 28: Normalized performance of CSIL again baselines for online imitation learning on MuJoCo Gym tasks. Uncertainty intervals depict quartiles over ten seeds. CSIL exhibits stable convergence with both sample and demonstration efficiency. BC here is applied to the demonstration dataset. $n$ refers to demonstration trajectories.

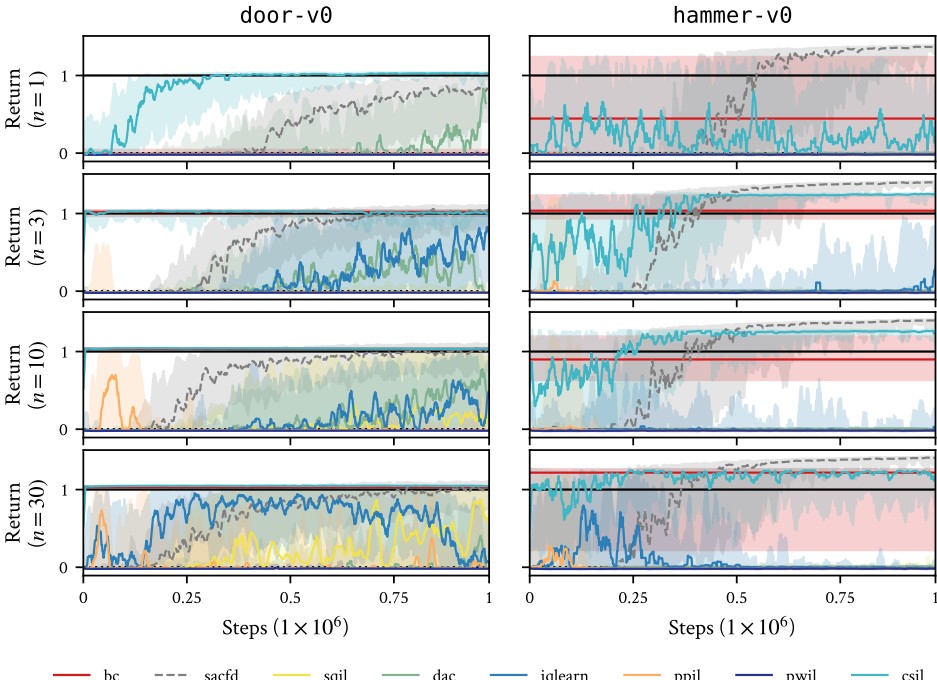

Figure 29: Normalized performance of CSIL against baselines for online imitation learning for Adroit tasks. Uncertainty intervals depict quartiles over ten seeds. CSIL exhibits stable convergence with both sample and demonstration efficiency. Many baselines cannot achieve stable convergence due to the high-dimensional action space. SACfd is an oracle baseline that combines the demonstrations with the true (shaped) reward. $n$ refers to demonstration trajectories.

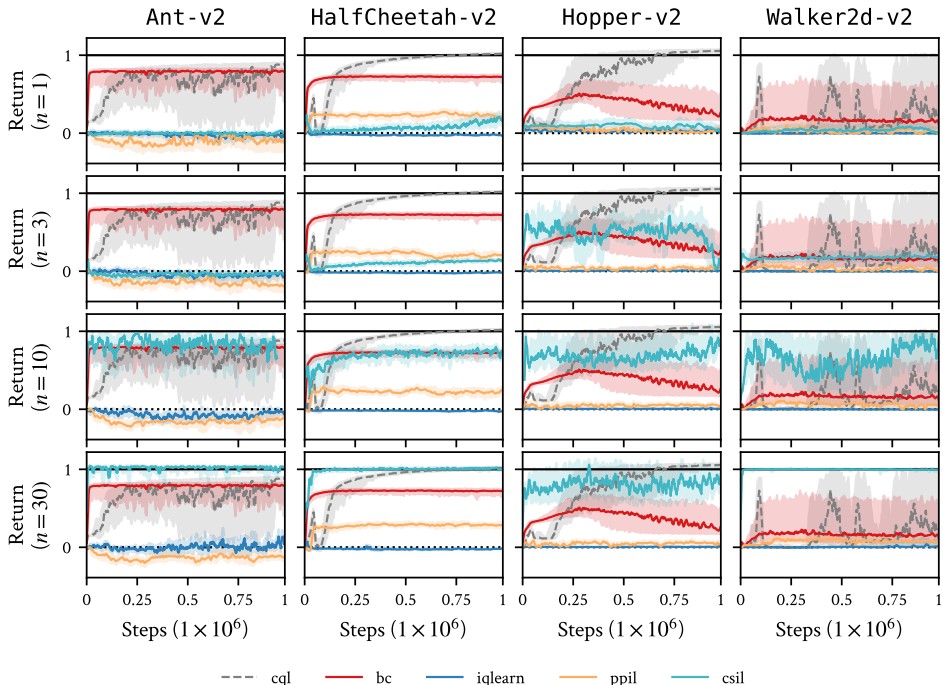

Figure 30: Normalized performance of CSIL against baselines for *offline* imitation learning for Gym tasks . Uncertainty intervals depict quartiles over ten seeds. While all methods struggle in this setting compared to online learning, CSIL manages convergence with enough demonstrations. CQL is an oracle baseline using the true rewards. The BC baseline is trained on the whole offline dataset, not just demonstrations. $n$ refers to demonstration trajectories.

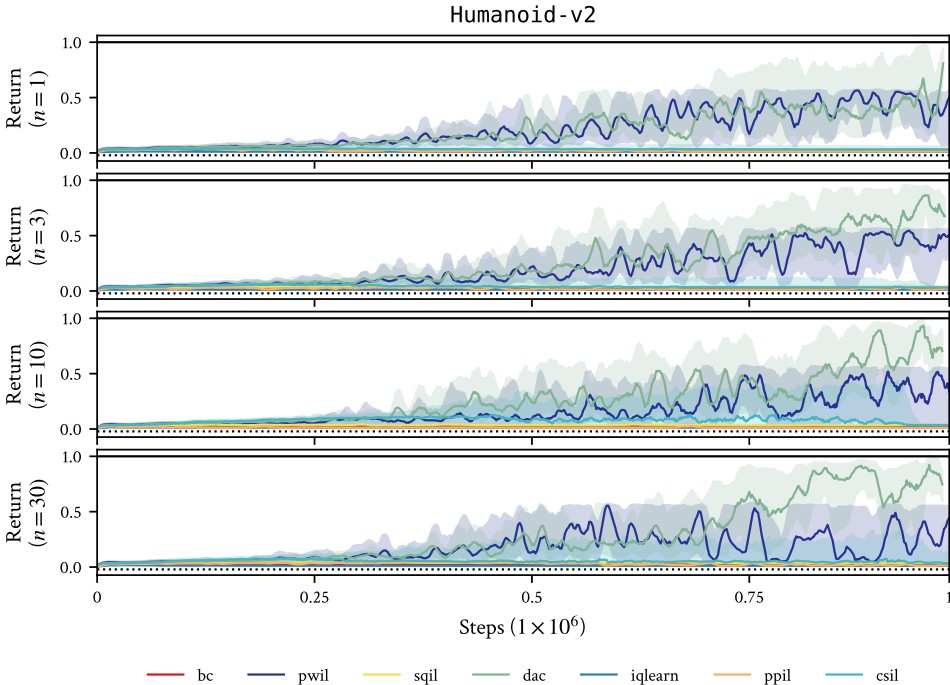

Figure 31: Normalized performance of CSIL against baselines for online imitation learning for Humanoid-v2. Uncertainty intervals depict quartiles over ten seeds. $n$ refers to demonstration trajectories.

## M.2 Continuous control from agent demonstrations

**Step-wise results.** Figure 28, 29 and 30 show step-wise performance curves, as opposed to the demonstration-wise results in the main paper. For online experiments, steps refers to the actor and learner, while for offline experiments, steps correspond to the learner.

**Humanoid-v2 results.** We evaluated CSIL and baselines on the high-dimensional `Humanoid-v2` locomotion task in Figure 31. We found that CSIL struggled to solve this task due to the difficulty for BC to perform well at this task, even when using all 200 of the available demonstrations in the dataset.

One aspect of `Humanoid-v2` to note is that a return of around 5000 (around 0.5 normalized) corresponds to the survival / standing bonus, rather than actual locomotion (around 9000). While the positive nature of the coherent reward enabled learning the stabilize, this was not consistent across seeds. We also found it difficult to reproduce baseline results due to IQLearn's instability, which we also reproduced on the author's implementation [100]. In theory, BC should be able to approximate the original optimal agent's policy with sufficient demonstrations. We believe the solution may be additional BC regularization, as the environment has a 376-dimensional state space.

**Performance improvement of coherent imitation learning** A desired quality of CSIL is its ability to match or exceed its initial BC policy. Figures 32, 33 and 34 show the performance of CSIL relative to its initial BC policy. Empirical, CSIL matches or exceeds the initial BC policy across environments.

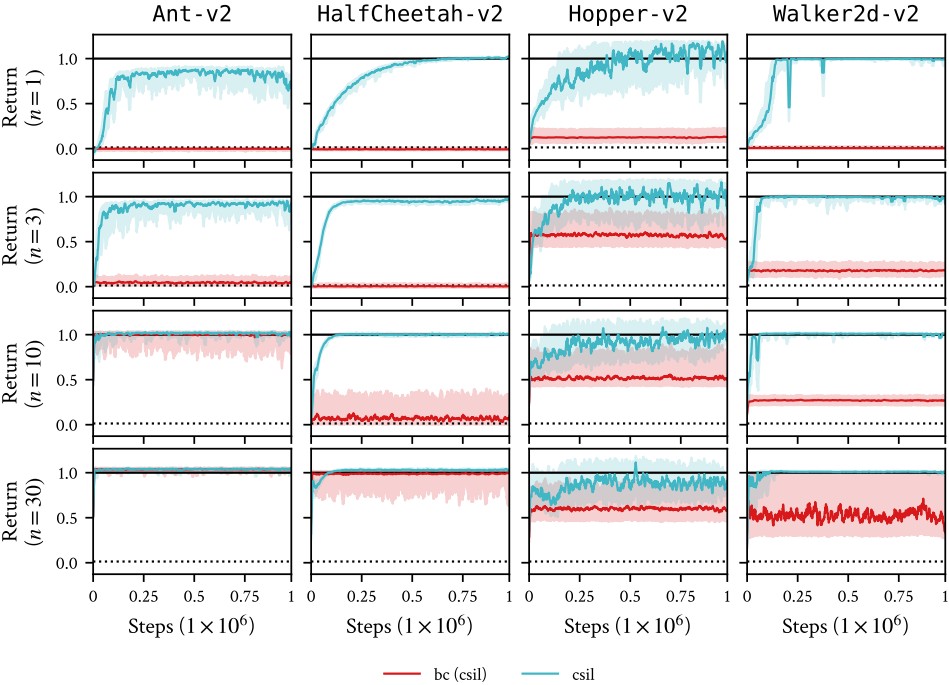

Figure 32: Normalized performance of CSIL for online `Gym` tasks against its BC initialization. Uncertainty intervals depict quartiles over ten seeds. $n$ refers to demonstration trajectories.

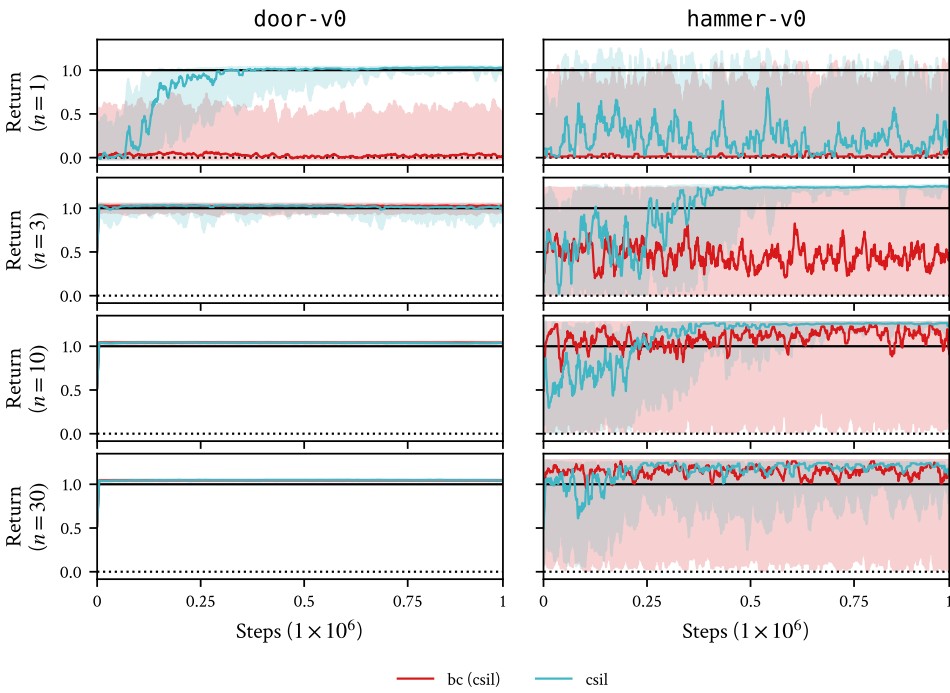

Figure 33: Normalized performance of CSIL for online `Adroit` tasks against its BC initialization. Uncertainty intervals depict quartiles over ten seeds. $n$ refers to demonstration trajectories.

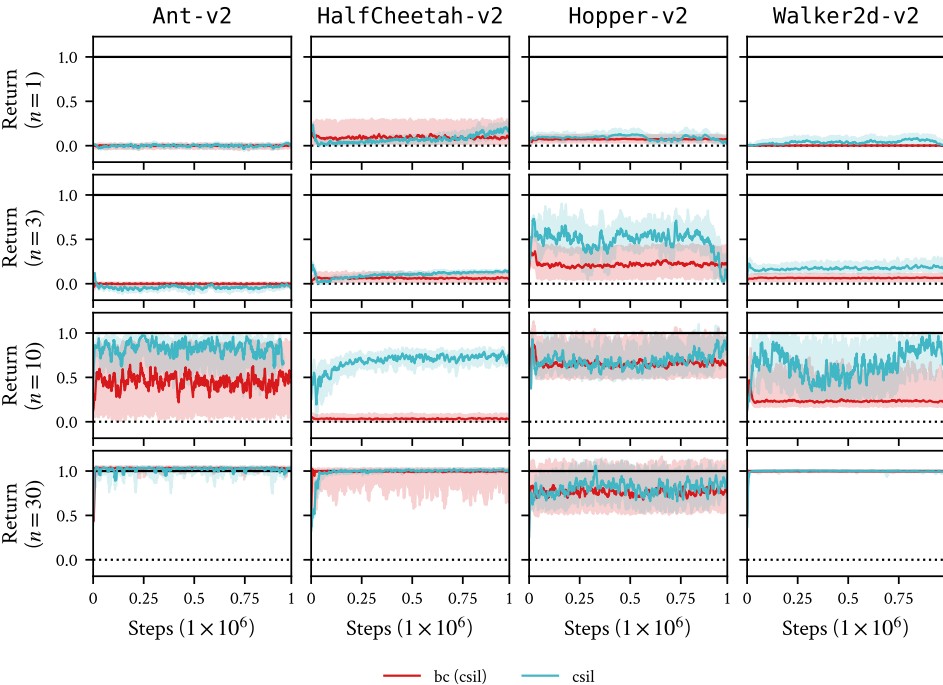

Figure 34: Normalized performance of CSIL for offline `Gym` tasks against its BC initialization. Uncertainty intervals depict quartiles over ten seeds. $n$ refers to demonstration trajectories.

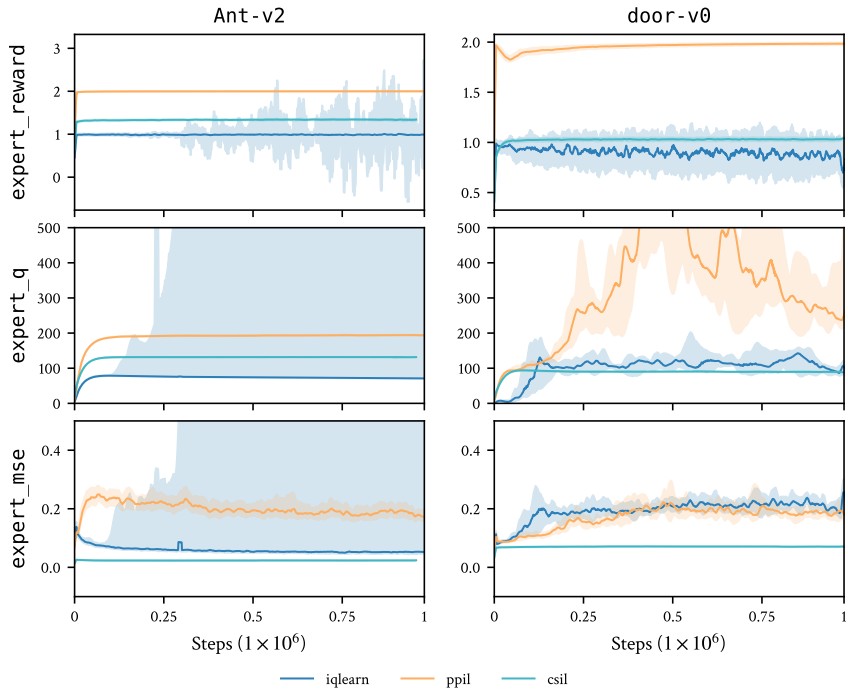

Figure 35: We investigate the weakness of IQLearn and PPIL for offline learning (Ant-v2) and high-dimensional action spaces (door-v0). Compared to CSIL, IQLearn and PPIL have less stable and effective $Q$ values, in particular for the expert demonstrations, which results in the policy not inaccurately predicting the demonstration actions (expert_mse).

**Divergence in minimax soft imitation learning** To investigate the performance issues in IQLearn and PPIL relative to CSIL, we monitor a few key diagnostics during training for online door-v0 and offline Ant-v2 in Figure 35. One key observation is the drift in the BC accuracy. We attribute this to the instability in the reward or critic due to the saddle-point optimization.

## M.3 Continuous control from human demonstrations

Figure 36 shows performance on the online setting w.r.t. learning steps, and Figure 37 shows the offline performance w.r.t. learning steps. For online learning, CSIL is consistently more sample efficient than the oracle baseline SACfD, presumably due to BC pre-training and CSIL's shaped reward w.r.t. robomimic's sparse reward on success. CSIL was also able to surpass Mandlekar et al. [84]'s success rate on NutAssemblySquare. For offline learning from suboptimal human demonstrations, it appears to be a much harder setting for CSIL to improve on the initial BC policy, especially for harder tasks.

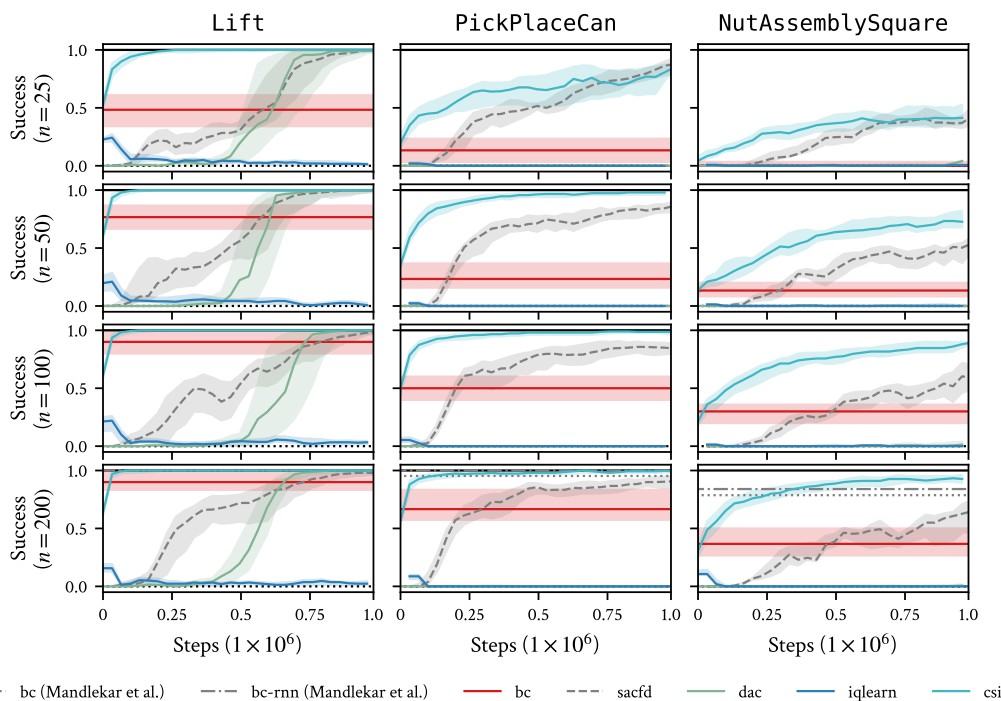

Figure 36: Average success rate over 50 evaluations for online imitation learning for `robomimic` tasks. Uncertainty intervals depict quartiles over ten seeds. The BC policy is trained on the demonstration data. $n$ refers to demonstration trajectories.

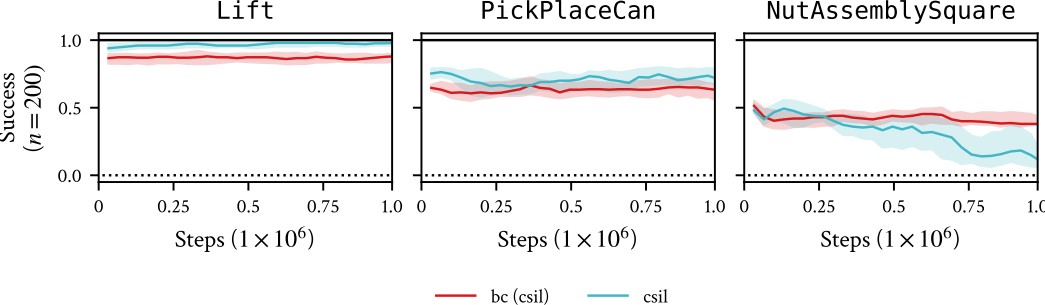

Figure 37: Average success rate over 50 evaluations for *offline* imitation learning for `robomimic` tasks. Uncertainty intervals depict quartiles over ten seeds. This experiment shows that learning from suboptimal offline demonstrations is much harder than on-policy data. It is also harder to use the KL regularization to ensure improvement over the initial policy. $n$ refers to demonstration trajectories.

# N   Ablation studies

This section includes ablation studies for the CSIL algorithm and baselines.

## N.1   Baselines with behavioral cloning pre-training

To assess the importance of BC pre-training, we evaluate the performance of IQLearn and PPIL with BC pre-training in Figure 38. While difficult to see across all experiments, there is a small drop in performance at the beginning of training as the policy 'unlearns' as the critic and reward are randomly initialized since there is no notion of coherency.

## N.2   Baselines with behavioral cloning pre-training and KL regularization

To bring IQLearn and PPIL closer to CSIL, we implement the baselines with BC pre-training and KL regularization, but with SAC-style policies rather than HETSTAT. We evaluate on the harder tasks, online Adroit, and offline Gym, to see if these features contribute sufficient stability to solve the task. Both Figure 39 and Figure 40 show an increase in initial performance, it is not enough to completely stabilize the algorithms.

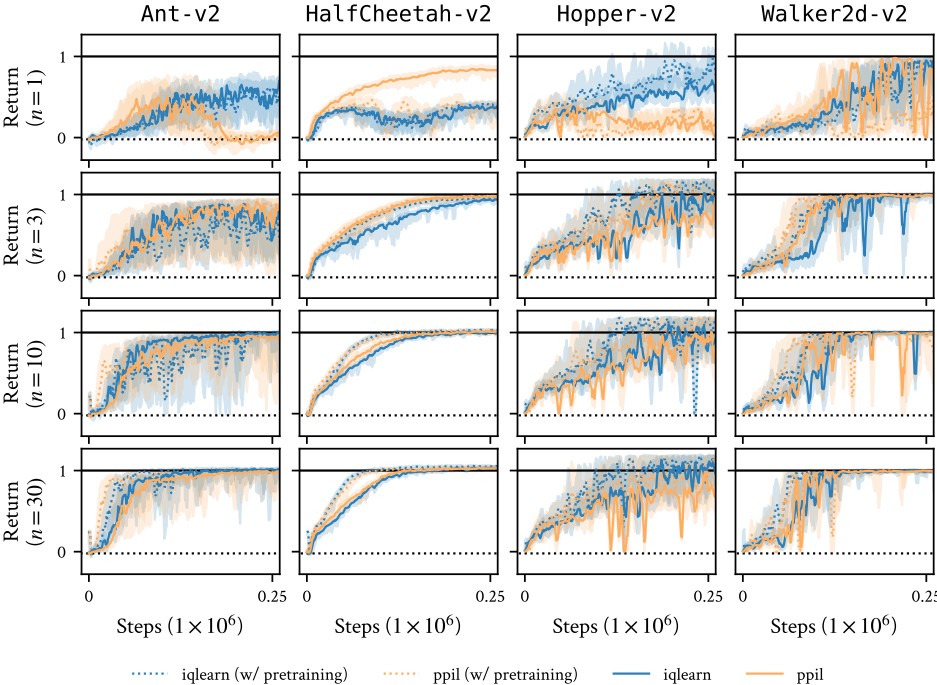

Figure 38: Normalized performance of IQLearn and PPIL for online Gym tasks with BC pre-training. Uncertainty intervals depict quartiles over ten seeds. $n$ refers to demonstration trajectories.

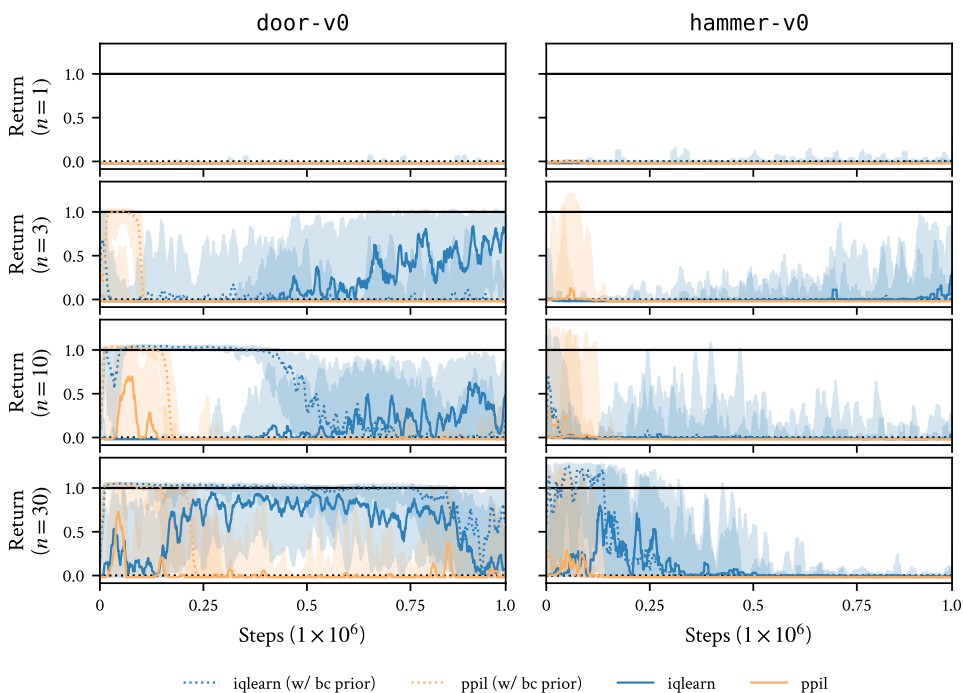

Figure 39: Normalized performance of IQLearn and PPIL for online `Adroit` tasks with BC pre-training. Uncertainty intervals depict quartiles over ten seeds. $n$ refers to demonstration trajectories.

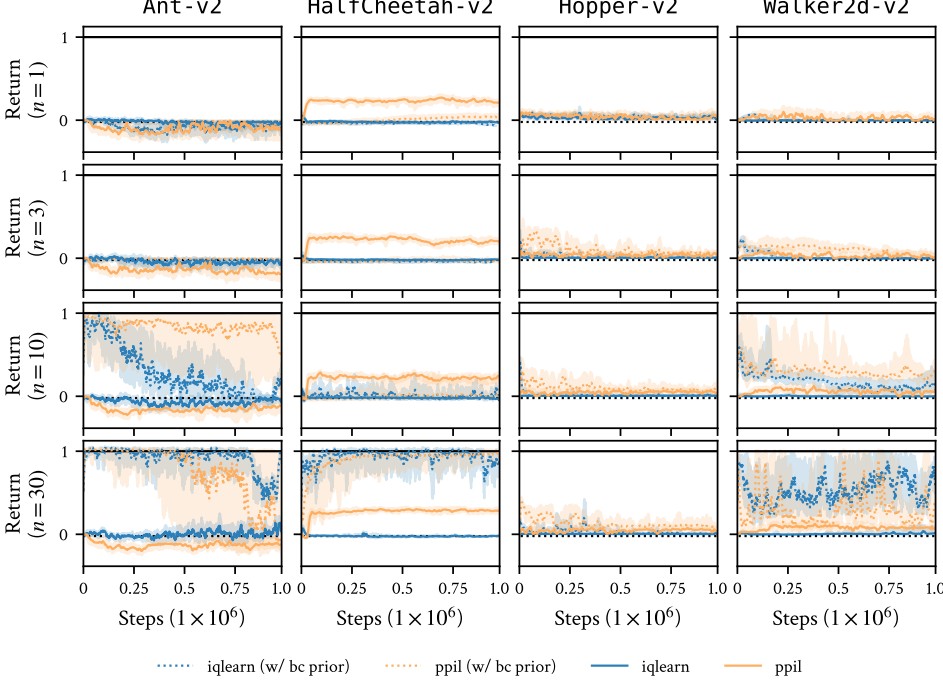

Figure 40: Normalized performance of IQLearn and PPIL for offline `Gym` tasks with BC pre-training. Uncertainty intervals depict quartiles over ten seeds. $n$ refers to demonstration trajectories.

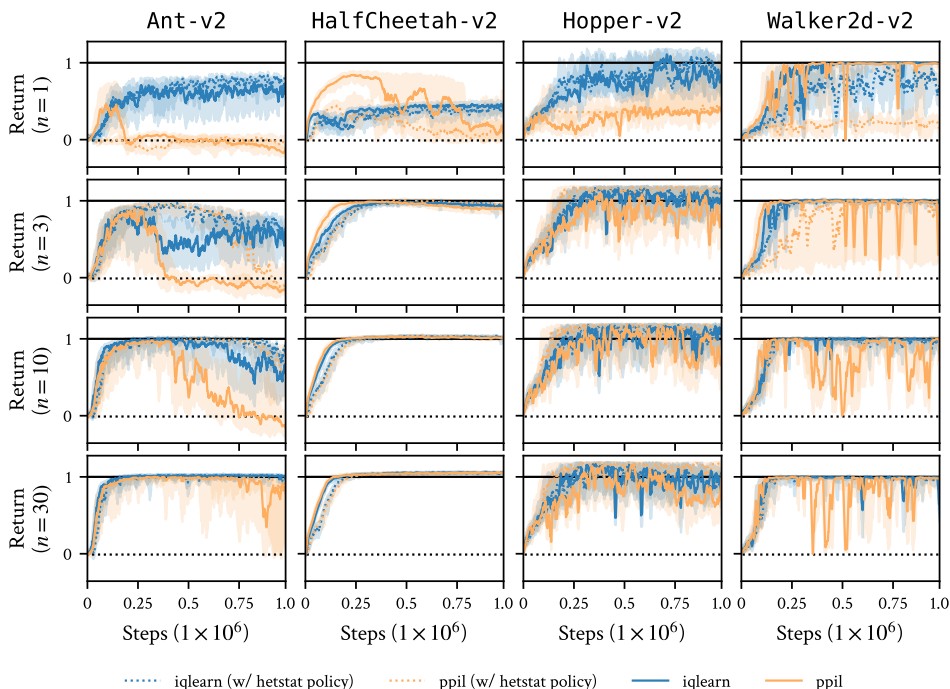

Figure 41: Normalized performance of IQLearn and PPIL for online Gym tasks with HETSTAT policies. Uncertainty intervals depict quartiles over ten seeds. $n$ refers to demonstration trajectories.

### N.3 Baselines with heteroscedastic stationary policies

To investigate the impact of HETSTAT policies alone, we evaluated PPIL and IQLearn with HETSTAT policies on the online Gym tasks in Figure 41. There is not a significant discrepancy between MLP and HETSTAT policies.

### N.4 Baselines with pre-trained heteroscedastic stationary policies as priors

To bring IQLearn and PPIL closer to CSIL, we implement the baselines with BC pre-training, HETSTAT policies and KL regularization with stationary policies. We evaluate on the harder tasks, online Adroit and offline Gym, to see if these features contribute sufficient stability to solve the task. Both Figure 42 and Figure 43 show an increase in performance, especially with increasing number of demonstrations, but it is ultimately not enough to completely stabilize the algorithms to the degree seen in CSIL.

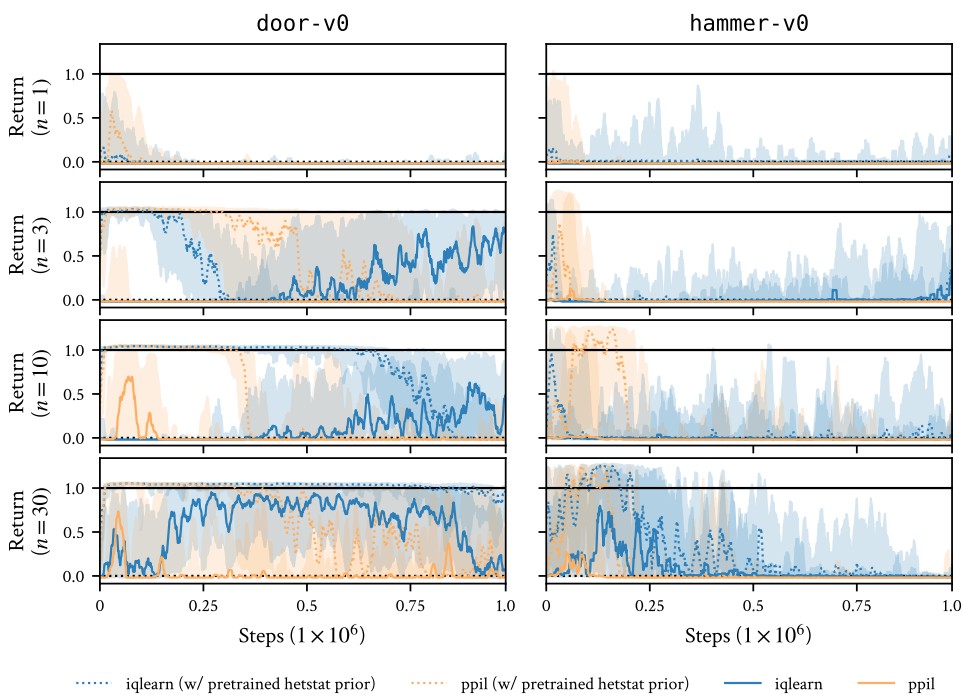

Figure 42: Normalized performance of IQLearn and PPIL for online `Adroit` tasks with pre-trained HETSTAT policies and KL-regularization. Uncertainty intervals depict quartiles over ten seeds. $n$ refers to demonstration trajectories.

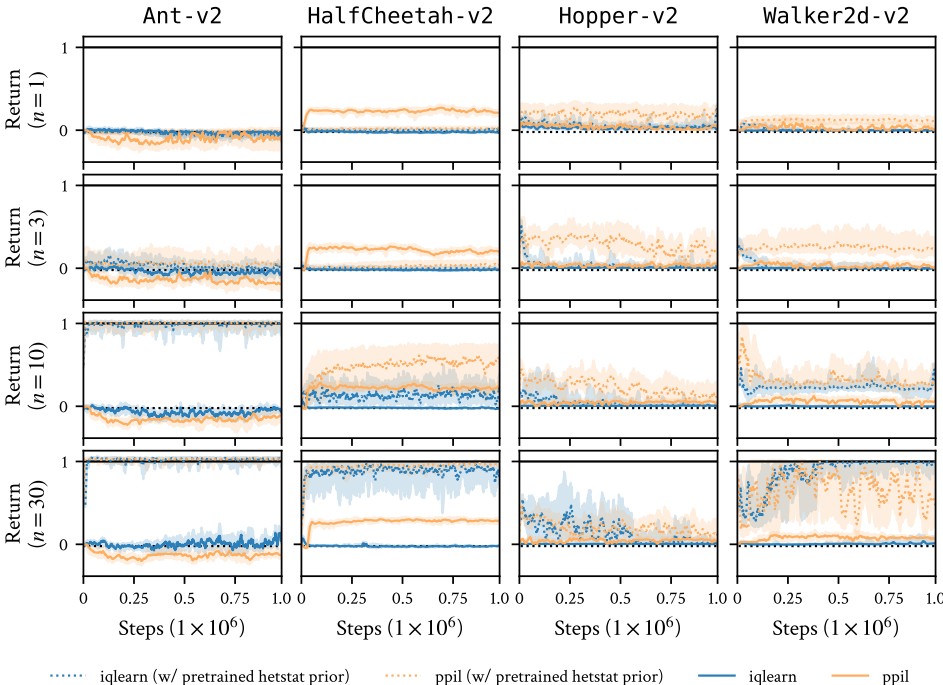

Figure 43: Normalized performance of IQLearn and PPIL for offline `Gym` tasks with pre-trained HETSTAT policies and KL-regularization. Uncertainty intervals depict quartiles over ten seeds. $n$ refers to demonstration trajectories.

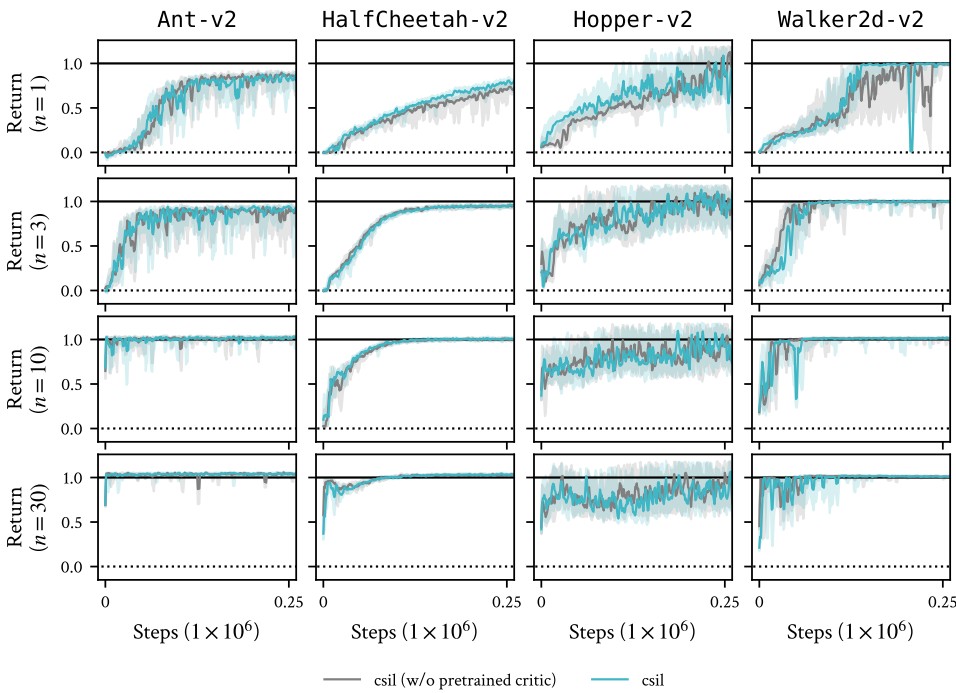

Figure 44: Normalized performance of CSIL for online Gym tasks a critic pre-training ablation. Uncertainty intervals depict quartiles over ten seeds. $n$ refers to demonstration trajectories.

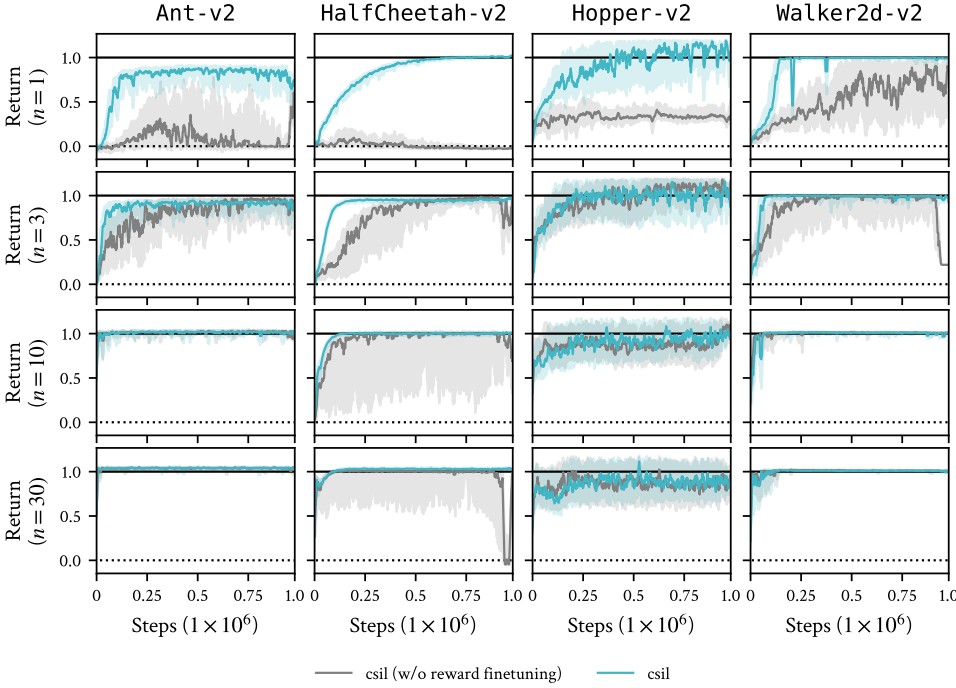

Figure 45: Normalized performance of CSIL for online Gym tasks with a reward finetuning ablation. Uncertainty intervals depict quartiles over ten seeds. Refinement is most important in the low demonstration setting where the initial BC policy and coherent reward is less defined, and therefore the initial policy is poor and the coherent reward is more susceptible to exploitation of the stationary approximation errors. $n$ refers to demonstration trajectories.

### N.5 Importance of critic pre-training

To ablate the importance of critic pre-training, Figure 44 shows the performance difference on online Gym tasks. Critic pre-training provides only a minor performance improvement.

### N.6 Importance of reward finetuning

To ablate the importance of reward finetuning, Figure 45. shows the performance difference on online Gym tasks. Reward finetuning is crucial in the setting of few demonstration trajectories, as the imitation reward is sparse and therefore CSIL is more susceptible to approximation errors in stationarity. It is also important for reducing performance variance and ensuring stable convergence.

### N.7 Importance of critic regularization

To ablate the importance of critic Jacobian regularization, Figure 46. shows the performance difference on online Gym tasks. The regularization has several benefits. One is faster convergence, as it encodes the coherency inductive bias into the critic. A second is stability, as it encodes first-order optimality w.r.t. the expert demonstrations, mitigating errors in policy evaluation. The inductive bias also appears to reduce performance variance. One observation is that the performance of Ant-v2 improves without the regularization. This difference could be because the regularization encourages imitation rather than apprenticeship-style learning where the agent improves on the expert.

### N.8 Stationary and non-stationary policies

One important ablation for CSIL is evaluating the performance with standard MLP policies like those used by the baselines. We found that this setting was not numerically stable enough to complete an ablation experiment. Due to the spurious behavior (e.g., as shown in Figure 3), the log-likelihood values used in the reward varied significantly in magnitude, destabilizing learning significantly, leading to policies that could no longer be numerically evaluated. This result means that stationarity is a critical component of CSIL.

### N.9 Effectiveness of constant rewards

Many imitation rewards typically encode an inductive bias of positivity or negativity, which encourage survival or 'minimum time' strategies respectively [40]. We ablate CSIL with this inductive bias for survival tasks (Hopper-v2 and Walker-v2) and minimum-time tasks (robomimic). Figure 47 shows that positive-only rewards are sufficient for Hopper-v2 but not for Walker-v2. Figure 48 shows that negative-only rewards is effective for the simpler tasks (Lift) but not the harder tasks (NutAssemblySquare).

### N.10 Importance of behavioral cloning pre-training

Figure 49 shows CSIL without any policy pre-training, so the reward is learned only though the finetuning objective. CSIL without pre-training does not appear to work for any of the tasks. This poor performance could be attributed to the lack of hyperparameter tuning, since the reward learning hyperparameters were chosen assuming a good initialization. It could also due to the maximum likelihood fitting of the coherent reward, since the refinement does not use the 'faithful' objective (Section D) that improves the regression fit.

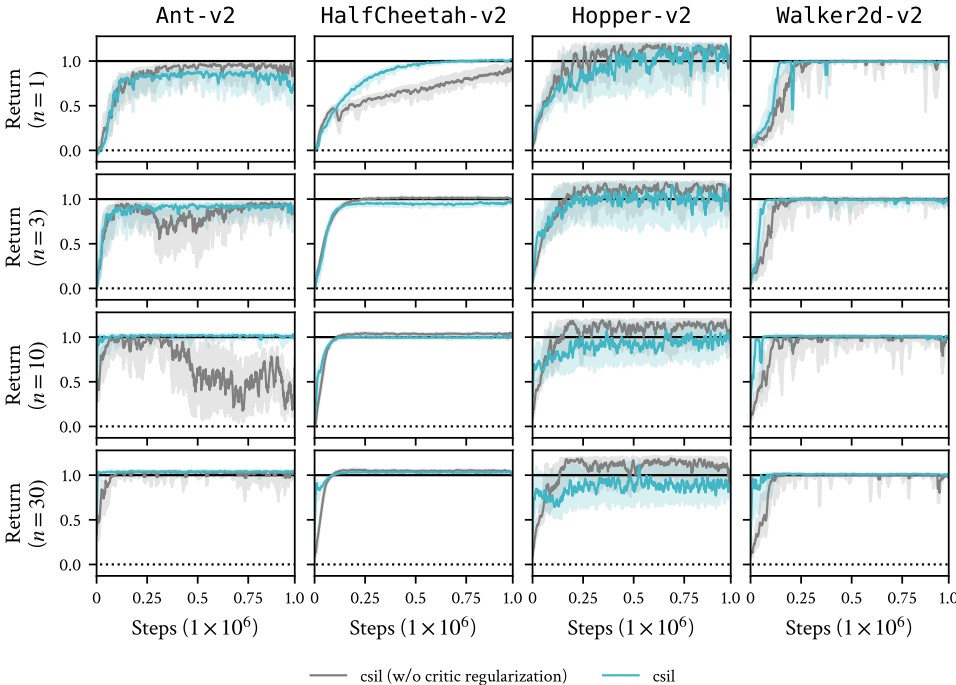

Figure 46: Normalized performance of CSIL for online MuJoCo Gym without regularizing the critic Jacobian. Uncertainty intervals depict quartiles over ten seeds. The benefit of this regularization is somewhat environment dependent, but clearly regularizes learning to the demonstrations effectively in many cases. Performance reduction is likely due to underfitting during policy evaluation due to the regularization. $n$ refers to demonstration trajectories.

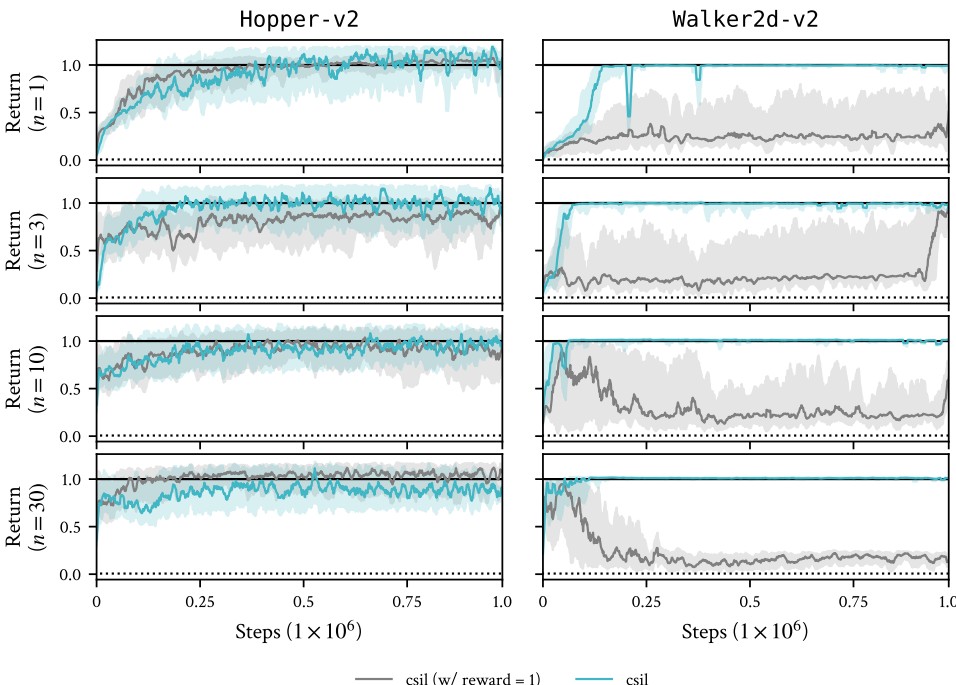

Figure 47: Online MuJoCo Gym tasks with absorbing states where CSIL is given a constant +1 reward function. Uncertainty intervals depict quartiles over ten seeds. While this is sometimes enough to solve the task (due to the absorbing state), the CSIL reward performs equal or better. $n$ refers to demonstration trajectories.

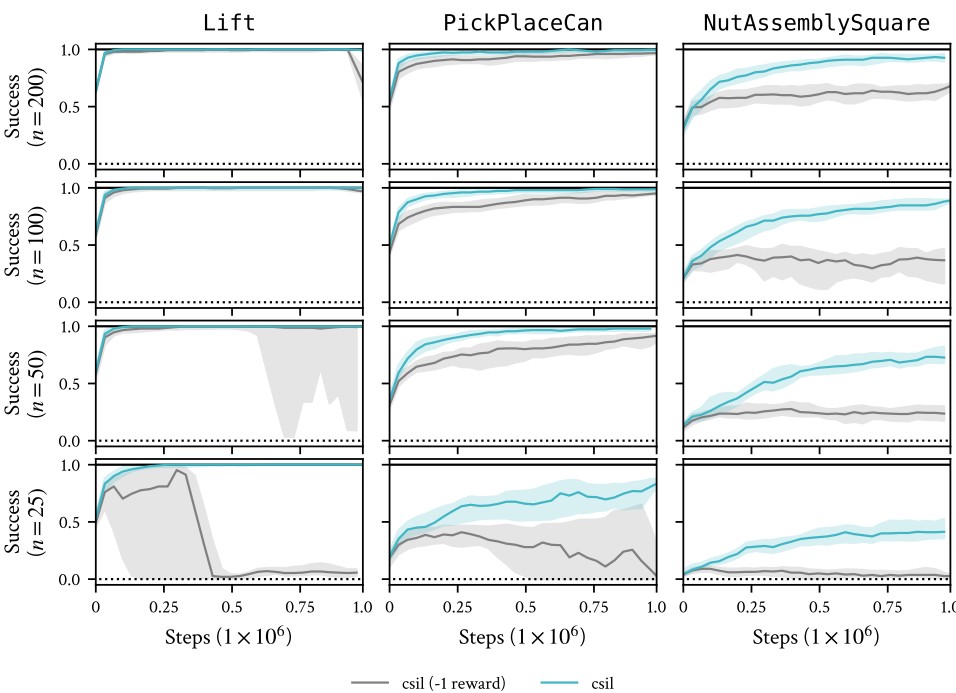

Figure 48: Online `robomimic` tasks where CSIL is given a constant -1 reward function. Uncertainty intervals depict quartiles over ten seeds. While a constant reward is sometimes enough to solve the task (due to the absorbing state), the CSIL reward performs roughly equal or better, with lower performance variance. $n$ refers to demonstration trajectories.

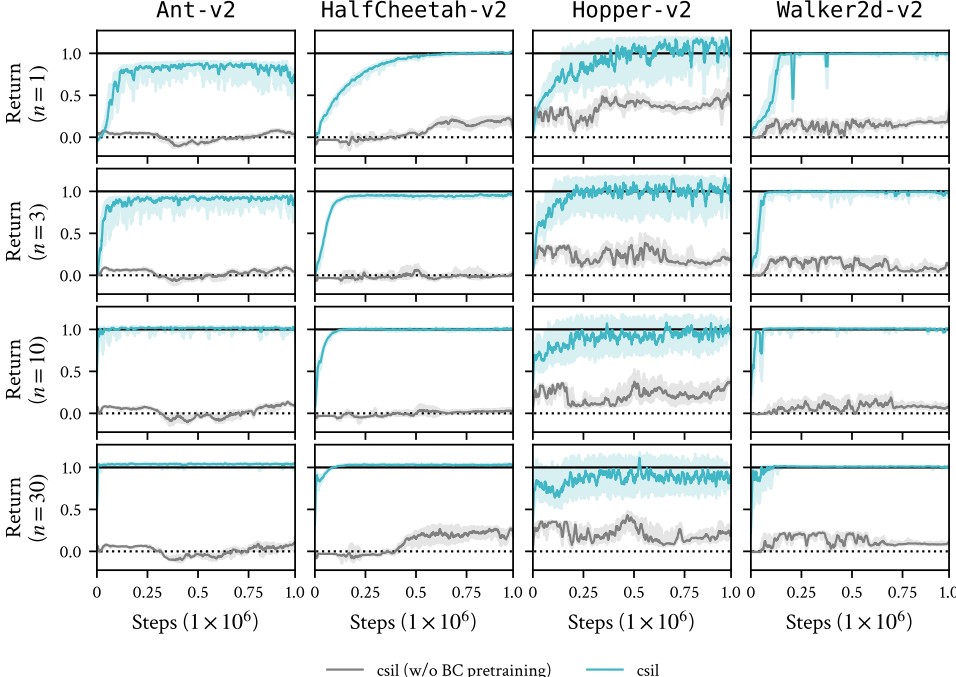

Figure 49: Online `MuJoCo` gym tasks where CSIL has no BC pre-training. Uncertainty intervals depict quartiles over 5 seeds. $n$ refers to demonstration trajectories.

