# OpenReview forum: "Coherent Soft Imitation Learning"
_NeurIPS.cc/2023/Conference — NeurIPS 2023 spotlight_

### Official Review · Reviewer_BA8W · 2023-06-26

**Soundness:** 3 good
**Presentation:** 2 fair
**Contribution:** 2 fair
**Rating:** 6
**Confidence:** 4

**Summary:**

This paper studies the imitation learning (IL) problem. There are two main classes of IL methods: behavioral cloning (BC) and inverse reinforcement learning (IRL), each with its own advantages. This paper proposes an IL method Coherent Soft Imitation Learning (CSIL) which combines BC and IRL. CSIL first learns a reward function named “coherent reward” under which the BC policy is optimal, and then performs soft policy iteration on the learned coherent reward. The authors evaluate CSIL on a wide range of settings and tasks including online and offline IL for high-dimensional and image-based continuous control tasks.

**Strengths:**

1. The proposed algorithm CSIL is novel according to my knowledge.
2. The authors evaluate CSIL on a wide range of settings and tasks including tabular tasks, online and offline IL from agent demonstrations, IL from human demonstrations from states and images.

**Weaknesses:**

1. The proposed method CSIL fails to combine the advantages of both BC and IRL, which is the main contribution claimed in this paper. In CSIL, the coherent reward function is learned under the principle that the BC policy is the (soft) optimal policy with that reward function. So the best policy we can expect to obtain from performing soft policy iteration on the learned coherent reward function is the BC policy. Besides, CSIL does not retain the main merit of IRL: addressing the compounding errors issue of BC. It is known that IRL leverages the state-action distribution matching principle to address the compounding errors issue in BC [1, 2, 3]. However, CSIL does not inherit this principle to learn the reward function. Therefore, the advantage of CSIL over BC and IRL is unclear.
2. The theory about the coherent reward (Theorem 1) is not novel. In [4], they have derived the feasible reward formula in the maximum entropy RL setting, which is exactly the case when the prior policy is a uniform policy in Theorem 1 (this choice of prior policy is also used in experiments in this paper). However, the authors do not discuss this existing result.
3. The writing in this paper needs improvement, and some parts are confusing. Even though I have read most of the IL papers in the reference, I still found it difficult to follow this paper. Please refer to the detailed comments in the Question section.
4. In experiments, CSIL performs worse than the existing method DAC on the online IL in MuJoCo tasks. DAC outperforms CSIL in HalfCheetah-v2 and Hopper-v2 while having a competitive performance with CSIL in Ant-v2 and Walker2d-v2. Besides, the offline IL setting considered in this paper is actually the offline IL with a supplementary dataset setting instead of the pure offline IL setting. However, the authors does include the methods in the offline IL with a supplementary dataset setting [5, 6] for comparison.

References:

[1] Seyed Kamyar Seyed Ghasemipour, Richard Zemel, and Shixiang Gu. A divergence minimization perspective on imitation learning methods. In Conference on Robot Learning, 2019.

[2] Nived Rajaraman, Lin Yang, Jiantao Jiao, and Kannan Ramchandran. Toward the fundamental limits of imitation learning. Advances in Neural Information Processing Systems, 2020.

[3] Tian Xu, Ziniu Li, and Yang Yu. Error bounds of imitating policies and environments. Advances in Neural Information Processing Systems, 2020.

[4] Haoyang Cao, Samuel Cohen, and Lukasz Szpruch. Identifiability in inverse reinforcement learning. In Advances in Neural Information Processing Systems, 2021.

[5] Geon-Hyeong Kim, Seokin Seo, Jongmin Lee, Wonseok Jeon, HyeongJoo Hwang, Hongseok Yang, and Kee-Eung Kim. DemoDICE: Offline imitation learning with supplementary imperfect demonstrations. In Proceedings of the 10th International Conference on Learning Representations, 2022.

[6] Haoran Xu, Xianyuan Zhan, Honglei Yin, and Huiling Qin. Discriminator-weighted offline imitation learning from suboptimal demonstrations. In Prooceedings of the 39th International Conference on Machine Learning, 2022.

**Questions:**

1. Was Figure 1 obtained from experimental results? If so, please provide a brief description of the experimental setup used to obtain the results shown in Figure 1.
2. Algorithm 1 does not provide a clear description of the proposed CSIL algorithm. First, it does not explain how to initialize the shaped critic and only shows a fixed point equation where both sides depend on the critic to be learned. Additionally, Algorithm 1 does not clearly demonstrate how CSIL utilizes online interactions, offline samples, or the dynamic model.

**Limitations:**

Please see the detailed reviews in the Weakness part.

---

> ### Author Rebuttal · Authors · 2023-08-09
>
> We would like to thank the reviewer for their comments and critical feedback.
>
> **Weakness 1.**  The reviewer highlights an important aspect of CSIL: If the demonstration data covers the entire state-action space and the BC fit is exact, then indeed CSIL will leverage and match this expert BC policy. But in this case, BC should be all you need, because there should be no compounding errors if the demonstration coverage is ideal.
> However, in the case of where the expert demonstrations do not cover the entire state-action space, then there will be regions where the BC policy is undefined and likely ineffective, so we perform RL to improve the policy in these gaps.
> This is where CSIL comes in, by using the BC policy to construct the coherent reward and then using this reward to improve using additional environment interactions.
> This property is in contrast to BC, which is a non-interactive algorithm and cannot interact with the environment to improve the policy.
>
> We demonstrate this empirically in the tabular inverse optimal control setting (Table 1), where there are no approximations. In the sparse task, the BC policy has high performance, but is brittle to the wind disturbance due to compounding errors when out-of-distribution. CSIL is able to learn out-of-distribution actions that resolve this weakness to the same level as the best IRL baselines. The value functions in Appendix K.1 show that CSIL and GAIL also learn visually similar value functions. This validates our statement above that CSIL improves BC outside of the demonstration distribution.
>
> We are also able to show this in the function approximation setting. For example, in Figures 32 CSIL is able to learn an effective high-dimensional manipulation policy on door-v0 from one demonstration, when its initial BC policy is non-performant. Figure 28 shows that DAC and SAC-from-demonstrations requires 3x the experience to reach similar performance on door-v0 with one demonstration.
> This is evidence that CSIL learns an effective shaped reward from the demonstration data.
>
> In summary:
>  * CSIL improves on BC by being able to use additional interactions with a learned reward.
> * CSIL is an inverse RL method as it inverts the RL policy update and learns a shaped reward.
> * These properties are validated in our experimental results across a range of settings.
>
> **Weakness 2.** We wish to thank the reviewer for highlighting Theorem 1 and Remark 3 of Cao et al. (2021). We will mention this work alongside our acknowledgement of Jacq et al. in the main text, who also derive a similar maximum entropy policy inversion in 2019 and inspired our approach. We will also replace ‘derive’ with ‘use’ in our first contribution summary (line 36) to emphasize that our contribution of combining the policy inversion result with BC for IRL, rather than the mathematical result, although our relative entropy inversion is also a slightly more general derivation than the maximum entropy one.
>
> **Weakness 3.** We have restructured Section 3 to be easier to follow (see our main rebuttal response). If you have specific feedback for which passages were confusing or hard to follow, that would be appreciated and we would be happy to incorporate that feedback.
>
> **Weakness 4.**  In the paper we tried to show the performance of CSIL in many diverse domains, even when baselines may be stronger.
> We believe that it is unrealistic to expect a new approach to beat all existing baselines on all tasks.
> While DAC is indeed better in some of the online MuJoCo locomotion tasks, CSIL still solves these tasks, whereas DAC performs less well on the more complex Adroit and Robomimic tasks that CSIL can solve. Moreover, our implementation of DAC was based on the extensive hyperparameter and regularizer investigation of Orsini et al. [11], which makes it a strong baseline.
>
> We agree with the reviewer's suggestion for explicit offline imitation baselines for the offline locomotion tasks.
> We have run DemoDICE [1] and SMODICE [2] in our offline setting.
> In the interest of time, we used the author’s original open-source implementations of these algorithms rather than reimplement them in Jax and Acme in our codebase.
> We discuss these algorithms and their performance in our main rebuttal comment and include the experimental results in the attached PDF.
>
> **Q1.** Figure 1 is a toy contextual bandit problem used for illustration purposes. The generating script is 'pull_figure.py' in the provided code in the supplementary material.
> The classifier uses the same spectral-normalized network used in Acme's DAC implementation. The CSIL reward was computed using a non-parametric Gaussian process with a stationary squared exponential kernel. The aim of this figure was to illustrate why our log ratio using regression with a stationary process might be preferable to a classification approach. We have added more context to the figure caption.
>
> **Q2.** Algorithm 1 was written as a general summary of both the tabular and deep implementation of CSIL, which is why it lacks specificity.
> In the tabular case, the critic is initialized with the log policy ratio, exactly like the coherent reward.
> In the continuous case, the critic is pre-trained using SARSA and the squared Bellman error.
> The soft policy iteration (SPI) loop at the end of the algorithm could be any oracle-based, model-free or model-based SPI routine that uses the coherent reward.
> For the deep learning implementation, we use SAC.
> We have updated Algorithm 1 to improve clarity and added a second algorithm to the Appendix that specifically describes the SAC-based online deep RL implementation.
>
> We hope this rebuttal addresses your concerns. Please let us know if you have remaining concerns or further questions.
>
> [1] DemoDICE: Offline Imitation Learning with Supplementary Imperfect Demonstrations, Kim et al., ICLR 2022
>
> [2] Versatile Offline Imitation from Observations and Examples via Regularized State-Occupancy Matching, Ma et al., ICML 2022

---

> > ### Comment · Reviewer_BA8W · 2023-08-13
> > **Thanks for authors' response.**
> >
> > Thanks for authors’ detailed response. However, my major concern in Weakness 1 is not well addressed. The authors argue that when the demonstrations do not cover the entire state-action space, CSIL performs RL to improve the BC policy by using the BC policy to construct the coherent reward and then using this reward to improve using additional interactions. However, there remains a key question that **whether performing RL with the coherent reward function can improve the BC policy or not**. I think the answer is no. This is because **the coherent reward function is learned under the principle that the BC policy is the optimal policy. Therefore, the best policy we can expect to obtain by performing RL with the coherent reward is the BC policy**. So I think that CSIL cannot improve on BC in theory. This also shows the key difference between CSIL and IRL methods in learning reward functions: CSIL learns a reward function such that the BC policy is the optimal policy while IRL learns a reward function under the distribution matching principle. So CSIL does not retain the merit of IRL.

---

> > > ### Author Response · Authors · 2023-08-16
> > > **BC improvement [1/2]**
> > >
> > > We thank the reviewer for engaging with the rebuttal and following up with their concerns.
> > >
> > > The reviewer asked
> > > 'How does performing RL with the coherent reward function improve the BC policy?'.
> > >
> > > On a high-level, the answer is
> > > * The KL-regularization temperature changes between the coherent reward definition and RL finetuning. This adjustment allows the policy to deviate slightly from the BC policy during RL finetuning and therefore allows improvement.
> > >  If the temperature does not change, the policy does not improve on the BC policy
> > > * The policy is optimized with a $Q$ function.
> > > The $Q$ function is not the same as the coherent reward, because it combines dynamic programming and the coherent reward.
> > > The reward enables overcoming the compounding error problem by using policy evaluation
> > >
> > > CSIL's ability to improve the policy out-of-distribution with the coherent reward is one of the more subtle aspects of the method, so we provide more in-depth explanations with some examples or derivations.
> > >
> > > **The shaping view.** The policy is not optimized with the coherent reward greedily, but rather using a $Q$ function fitted using Bellman’s equation. The definition of the coherent reward means that it should ideally be positive in-distribution $(s,a \in \mathcal{D})$, negative for incorrect actions ($s \in \mathcal{D}, a\notin\mathcal{D}$) and zero outside of the demonstration state distribution ($s \notin\mathcal{D}$), as mentioned in lines 169 - 172.
> > >
> > > This shaping means that the $Q$ function defined using the coherent reward assigns higher value to actions that keep or return the agent to the demonstration distribution.
> > > This credit assignment applies to states out-of-distribution as well as in-distribution.
> > >
> > > This $Q$ function enables the policy to learn to overcome the compounding error problem typical in BC.
> > >
> > > Also, remember that the temperature is reduced for RL ($\beta<\alpha$), so the KL-regularization lets the policy improve on the BC policy by regularizing against the prior less. If $\beta = \alpha$ then the policy won't improve on the BC policy, which happens because the KL divergence regularization cancels out the coherent reward in the soft Bellman equation.
> > >
> > > *Example.*
> > > Consider the tabular setting where the BC policy fits a deterministic expert perfectly for all states in the demonstration data, and matches the prior otherwise.
> > > The prior is uniform, where $p(a|s) = 1/|\mathcal{A}|\forall s\in\mathcal{S}$.
> > > This setting means that the coherent reward has the following values:
> > > * For $s,a \in \mathcal{D}$, $r(s,a) = \alpha(\log 1 - \log1/|\mathcal{A}|)=\alpha\log|\mathcal{A}|$
> > > * For $s \in \mathcal{D}, a\notin\mathcal{D}$, $r(s,a) =\alpha( \log 0 - \log1/|\mathcal{A}|)= -\infty$
> > > * For $s\notin \mathcal{D},\forall a\in\mathcal{A}$, $r(s,a) = \alpha( \log1/|\mathcal{A}|-\log1/|\mathcal{A}|)=0$
> > >
> > > This reward results in the following returns for discount factor $0 < \gamma \leq 1$
> > > * A trajectory that stays in-distribution has return $\frac{1}{1-\gamma}\alpha\log|\mathcal{A}|$
> > > * A trajectory that leaves the demonstration distribution (due to compounding errors) at time $t_1$ has return $\frac{1-\gamma^{t_1}}{1-\gamma}\alpha\log|\mathcal{A}|$
> > > * A trajectory that leaves the demonstration distribution (due to compounding errors) at time $t_1$, but manages to recover at time $t_2$ has return
> > >     $(\frac{1}{1-\gamma} - \frac{1 - \gamma^{t_2}}{1-\gamma} +  \frac{1-\gamma^{t_1}}{1-\gamma})\alpha\log|\mathcal{A}|$
> > >
> > > Therefore trajectories that stay in-distribution longer and recover faster have higher returns.
> > > These trajectory returns mean the agent is encouraged to learn to overcome compounding errors and stay in-distribution.
> > >
> > > (continued in the next comment)

---

> > > > ### Author Response · Authors · 2023-08-16
> > > > **BC improvement [2/2]**
> > > >
> > > > **The divergence view.**
> > > > We can derive a divergence-minimization perspective by looking at the BC step and RL step.
> > > > Combining these two objective can be viewed as a form of two-stage approximate distribution matching between the policy and the expert.
> > > >
> > > > The BC step maximizes the likelihood of the policy, so it can be written as minimizing the forward KL between the data distribution and the BC policy,
> > > >
> > > > $\text{argmin}_\theta \mathbb{E}\_{s,a \sim \mathcal{D}}[ -\log q\_\theta(a|s)] = \text{argmin}\_{\theta} \mathbb{E}\_{s \sim \mathcal{D}}[\text{KL}[p\_{\mathcal{D}}(a|s) \mid\mid q\_\theta(a|s)]]$
> > > >
> > > > There is also regularization to keep $q_\theta$ close to $p$ outside of the demonstration distribution.
> > > >
> > > > The KL-regularized RL step with the coherent reward
> > > > $r_\theta(s,a) = \alpha (\log q_\theta(a|s) - \log p(a|s))$, policy $\pi$ and discounted state distribution $\mu$ is
> > > >
> > > > $\max_{\pi} \mathbb{E}\_{a\sim \pi(\cdot|s),  s\sim \mu_\pi(\cdot) }\left[\alpha \log\frac{q\_\theta(a|s)}{p(a|s)} - \beta \log\frac{\pi(a|s)}{p(a|s)}\right].$
> > > >
> > > > Switching to the minimization problem, rearranging and dividing by $\alpha$ yields
> > > >
> > > > $\min_\pi \mathbb{E}\_{ a\sim \pi(\cdot|s), s\sim \mu_\pi(\cdot)}\left[\frac{\beta}{\alpha}\log \pi(a|s) - \log q_\theta(a|s) + (1-\beta/\alpha)\log p(a|s)\right].$
> > > >
> > > > This objective can be expressed using the conditional KL divergences
> > > > (note, since $\beta<\alpha$, $(1-\beta / \alpha)>0$),
> > > >
> > > > $\min_{\pi} \mathbb{E}\_{s \sim \mu_\pi(\cdot)}[\text{KL}[\pi(a|s) \mid\mid q_\theta(a|s)]] - (1-\beta / \alpha)\mathbb{E}\_{s \sim \mu_\pi(\cdot)}[\text{KL}[\pi(a|s) || p(a|s)].$
> > > >
> > > > This objective has the following properties (remembering $q_\theta$ is regularized to be close to $p$ out-of-distribution):
> > > > * The policy $\pi$ wants to match the BC policy (left) using the reverse KL
> > > > *  The policy maximizes the relative entropy between $\pi$ and $p$ (right), while also desiring a stationary distribution $\mu_\pi$ in the higher KL regions
> > > > * When $p$ is uniform, $\pi$ seeks to reduce its causal entropy and seek states where the causal entropy is low, which will be the demonstration states due to BC step. So $\mu_\pi$ should match the demonstration's state distribution.
> > > >
> > > > The KL maximization term on right means that $\pi$ will not have to perfectly match $q_\theta$, and allows 'slack' to improve upon the BC policy by seeking the low causal entropy states.
> > > > As mentioned earlier, if $\beta = \alpha$, $\pi$ will directly mimic the BC policy $q\_\theta$ and the KL maximization term disappears.
> > > >
> > > > What is interesting with this approach is that BC is usually done with the forward KL, as we cannot access the log probabilities of the data distribution. By using the BC policy as a model of the data distribution, in the RL step we can fit the reverse KL between the policy and the BC policy since we have access to the log probabilities. The reverse KL is more desirable as it requires samples from the policy (rather than the data) which means the policy is evaluated on the environment.
> > > >
> > > > If the reviewer finds this divergence perspective helpful, we can include it in the appendix.
> > > >
> > > > Please let us know if this addresses your question and if you have additional concerns.

---

> > > > > ### Comment · Reviewer_BA8W · 2023-08-18
> > > > >
> > > > > I greatly appreciate the authors’ detailed response. Here are my follow-up comments.
> > > > >
> > > > > First, I think that the ``shaping view'' cannot be used to support the claim that performing RL with the coherent reward can improve the BC policy. This is because, under the coherent reward function, the BC policy is exactly the optimal policy that is optimal with the $Q^\star$ function. Thus, optimizing with a Q-function does not necessarily yields a policy that is better than the BC policy.
> > > > >
> > > > > Second, the ``divergence view'' indeed conveys some useful intuition as mentioned in the authors' response. However, the divergence objective is so complicated such that it is hard to draw a rigorous argument from this perspective. In particular, there are two KL terms and the state distribution also depends on the policy variable. Therefore, it is difficult to give a formal characterization of the optimal solution to this divergence objective.
> > > > >
> > > > > Nevertheless, this divergence view indeed provides some justification for the proposed method, which partly addresses my concern. Thus I will raise my score from 4 to 5.

---

> > > > > > ### Author Response · Authors · 2023-08-19
> > > > > >
> > > > > > We wish to thank the reviewer for their follow-up comments.
> > > > > > We are glad to hear the explanations helped improve your understanding enough to increase their score.
> > > > > > We agree that the divergence view is unfortunately not as straightforward as prior works.
> > > > > >
> > > > > > To follow up on the reward shaping comment, perhaps the implication of the rescaled KL regularization on the reward was not made clear enough in our earlier explanation.
> > > > > > In soft policy iteration methods like SAC, the entropy regularization can be seen as augmenting the reward such that
> > > > > >
> > > > > > $r\_{\alpha, \pi}(s, a) = r(s,a) - \alpha (\log\pi(a\mid s) - \log p(a\mid s)).$
> > > > > >
> > > > > > The soft value function evaluates this augmented reward.
> > > > > >
> > > > > > This augmentation means that the BC policy $q$ is optimal for $r\_{\alpha, q}(s, a)$
> > > > > >
> > > > > > Our coherent reward shapes the $r(s,a)$ term using $\tilde{r}(s,a) = \alpha (\log q(a|s) - \log p(a|s))$.
> > > > > >
> > > > > > For the RL step of CSIL, we replace $r$ with $\tilde{r}$ and $\alpha$ with $\beta$ so
> > > > > >
> > > > > > $\tilde{r}_{\beta,\pi}(s,a) = \tilde{r}(s, a) - \beta(\log\pi(a\mid s) - \log p(a\mid s)).$
> > > > > >
> > > > > > When $\beta = \alpha$, the reviewer is correct that $\pi$ does not improve on the BC policy $q$ because $\tilde{r}\_{\alpha, q}(s,a) = 0 \forall s \in \mathcal{S}, \forall a \in \mathcal{A}$ when $\pi$ is initialized to $q$.
> > > > > >
> > > > > > Because of this, we use $\beta < \alpha$, so the reward is actually *not* the same as the one for which the BC policy is optimal.
> > > > > >
> > > > > > However, due to the coherent reward's shape and the reduction in the KL regularization, this slight change in reward is such that the policy
> > > > > > improves to overcome compounding errors via the $Q$ function, reduces causal entropy, and does not result in 'unlearning', as we discussed in the previous responses.
> > > > > >
> > > > > > Unfortunately, because $\tilde{r}\_{\beta, \pi}(s,a)$ depends on $\pi$ it's not possible to derive numerical values for our worked tabular example, but we hope the augmented reward equations make things sufficiently clear.
> > > > > >
> > > > > > We hope this explanation is useful. This discussion has also been very helpful for us to better understand and communicate these subtleties.

---

> > > > > > > ### Comment · Reviewer_BA8W · 2023-08-20
> > > > > > >
> > > > > > > Thanks a lot for the authors' further clarification. However, I find the subsequent argument somewhat challenging to grasp. The claim that "this slight change in reward is such that the policy improves to overcome compounding errors via the $Q$ function, reduces causal entropy." lacks clarity in its explanation. The key question here is **how** the $Q$ function helps the policy improve to overcome compounding errors. To answer this question, we need to analyze the Q-functions in the training process. Currently, it seems to be difficult to conduct such an analysis due to the complicated objective.

---

> > > > > > > > ### Author Response · Authors · 2023-08-20
> > > > > > > >
> > > > > > > > Thanks for following up.
> > > > > > > >
> > > > > > > > >The key question here is **how** the $Q$ function helps the policy improve to overcome compounding errors.
> > > > > > > >
> > > > > > > > We agree that analyzing the nature of CSIL's soft $Q$ function is an interesting question.
> > > > > > > >
> > > > > > > > We believe there are two ways to tackle this.
> > > > > > > >
> > > > > > > > 1] Set $\beta = 0$, so there is no entropy regulation during the RL stage. In this case, the reviewer can refer to the coherent reward returns we outlined in the tabular example in the 'BC improvement [1/2]' on the 16th August.
> > > > > > > >
> > > > > > > > These returns show that trajectories that
> > > > > > > > * stay in the demonstration distribution
> > > > > > > > * deviate and return back to the demonstration distribution
> > > > > > > >
> > > > > > > > have higher returns than those that deviate from the demonstration distribution and never return.
> > > > > > > >
> > > > > > > > Therefore, the $Q$ function will select actions that result in staying in, or returning to, the demonstration distribution.
> > > > > > > >
> > > > > > > > Since $\beta < \alpha$, the coherent reward term should dominate the KL regularization term, so the intuition from $\beta = 0$ should hold.
> > > > > > > > Moreover, if there is entropy regularization when $\beta = 0$, the causal entropy of the policy will decrease until the policy is deterministic.
> > > > > > > >
> > > > > > > > 2] Empirical results. In our tabular experiments in Appendix K.1, we visualize the tabular value functions of the algorithms.
> > > > > > > > For the sparse task, the demonstration data is only a single state-action trajectory in the state action space.
> > > > > > > > This state trajectory is shown in the 'nominal stationary distribution' of the expert in Figure 19.
> > > > > > > >
> > > > > > > > CSIL's soft value function, shown in Figure 26, shows that the learned values relate to the 'distance' from this demonstration trajectory.
> > > > > > > > Moreover, the 'Windy stationary distribution' subplot shows that CSIL's policy acts to return to the demonstration trajectory as fast as possible, which supports the intuition we provided in 1].
> > > > > > > >
> > > > > > > > This result is not surprising given our tabular example, where the reward is a positive constant for the demonstration states and actions and zero out-of-distribution, so using this reward with discounting will produce such a value function.
> > > > > > > > The classifier-based reward methods (e.g. GAIL, see Figures 21 and 22) also learn a very similar value function, since classifiers provide a similarly shaped reward in this setting.
> > > > > > > > We used CSIL's soft value function here to easily visualize the learned soft $Q$ function and verify that the $Q$ function was computing the value function we were expecting given the coherent reward shaping.
> > > > > > > >
> > > > > > > > Please let us know if these explanations have brought more clarity to the $Q$ function question.
> > > > > > > > If not, we would appreciate any suggestion of possible theory or experiments that would clarify the nature of the CSIL $Q$ function.

---

> > > > > > > > > ### Comment · Reviewer_BA8W · 2023-08-21
> > > > > > > > >
> > > > > > > > > I appreciate the authors' detailed response. The provided example and empirical results provide clear intuitions on the mechanism of CSIL. I strongly recommend that the authors could incorporate these discussions into the paper, which can help readers better understand the method. I have no further questions and have revised the score.

---

> > > > > > > > > > ### Author Response · Authors · 2023-08-22
> > > > > > > > > >
> > > > > > > > > > Thank you for your engagement and questions! We are pleased to hear we were able to address your concerns.
> > > > > > > > > >
> > > > > > > > > > We found the discussion very helpful and we will definitely incorporate the various discussion points into the paper.

---

### Official Review · Reviewer_nMWn · 2023-07-05

**Soundness:** 4 excellent
**Presentation:** 2 fair
**Contribution:** 3 good
**Rating:** 7
**Confidence:** 3

**Summary:**

This study seeks to leverage the advantages of Behavioral Cloning (BC) and Inverse Reinforcement Learning (IRL) to develop a sample-efficient imitation learning algorithm. However, the integration of these two approaches is not straightforward, as optimizing the policy using a dynamic reward diminishes the benefits of BC pre-training. To address this challenge, this work derives a shaped reward based on pre-trained BC according to the entropy-regularized policy update. This shaped reward enables policy refinement without compromising the advantages gained from BC, as it remains consistent with BC. Experimental results demonstrate that the proposed Coherent Soft Imitation Learning (CSIL) effectively addresses both online and offline imitation learning tasks in high-dimensional continuous control and image-based scenarios.

**Strengths:**

* The experiments are extensive and comprehensive.
* The proposed algorithm is able to solve complex control tasks using only a few demonstrations.
* The novelty is commendable.


**Weaknesses:**

* The performance is influenced a lot by the efficacy of BC.
* The algorithm requires some tricks to work well.
* The readability is not good.


**Questions:**

* How is p(a|s) in Eq 2 derived? It doesn’t seem to appear in [1].
* How is p(a|s) in Eq 4 derived? It doesn’t seem to appear in [2].
* Definition 1 appears to be somewhat redundant, given that the information is already provided in Eq 4 and KL-regularized RL objective in the Background section.
* What does "classifier" refer to in Fig 1? Does it correspond to the discriminator in GAIL?
* In Fig 1, it appears that the color intensity of Ours does not necessarily indicate a greater distance from the data. Does this imply that the agent may be unable to return to expert support in certain regions?
* Algorithm 1 is vague, for example:
    * Where is the fine tuning part using additional data(online/offline).
    * The shape coherent reward uses $\theta$ but there is no $\theta$ at the right hand side of $=$.
    * What’s the exact objective/equation for computing $\tilde{Q}_n$.
    * It would be better to include the tricks mentioned in Section 4 into Algorithm 1.
* The placement of “|” in Lemma 2 seems to be wrong: $p(Y, w| X)$ -> $p(Y | w, X)$
* What is the influence of the reference policy ($p(a | s)$ or $q_{\theta_1}(a | s)$)?
* What would happen if we only use Eq 10 or its improved version to train without BC pre-training?
* In Fig 27, BC’s performance is better than the initial performance of CSIL, which is different from Fig 31, what’s the reason for this?
* In Fig 31, there are some tasks where BC’s performance is better than CSIL, e.g HalfCheetah-v2(n=30) and Hopper-v2(n=3). This seems to contradict the feature of CSIL, what’s the reason for this?
* The scores of the experts are somewhat low at HalfCheetah-v2(8770) and Hopper-v2(2798), it would be better to train on better experts, e.g HalfCheetah-v2($\ge$12000) and Hopper-v2($\ge$3500).
* In Fig 3, why does CSIL exhibit a deteriorating performance in Hopper-v2 as the number of demonstrations increases?
* Some typos:
    * Fig 28, again -> against
    * Fig 36, s hows -> shows
    * Line 984, and and -> and

1: Brian D Ziebart. Modeling purposeful adaptive behavior with the principle of maximum causal entropy. Carnegie Mellon University, 2010.
2: Tuomas Haarnoja, Aurick Zhou, Pieter Abbeel, and Sergey Levine. Soft actor-critic: Off- policy maximum entropy deep reinforcement learning with a stochastic actor. In International Conference on Machine Learning, 2018.


**Limitations:**

As mentioned in the paper, the performance of CSIL is constrained by the capabilities of BC. If BC fails to successfully address the task with an adequate number of demonstrations, CSIL will encounter similar limitations.

---

> ### Author Rebuttal · Authors · 2023-08-09
>
> We thank the reviewer for their comprehensive review and feedback.
>
> We were able to address all the issues and address the specific questions below.
>
> **Policy prior $p(a|s)$.** Maximum entropy RL is a specific case of KL-regularized RL when $p(a|s)$ is a uniform distribution.  We mention this in the passage in lines 72 - 78.
>
> **Definition 1.** The inclusion of pseudo-posteriors is to emphasize the connection between KL-regularized optimization and Bayesian inference methods, which was crucial in this work due to the necessity of a stationary posterior policy given the uniform action prior. Repeating similar mathematical results is somewhat unfortunate, but is used to provide a formal definition while also showing how the notion of a pseudo-posterior is general to any KL-regularized optimization problem beyond the RL discussion in Section 2, such as regression task that BC performs.
>
> **Figure 1.** The goal of Figure 1 was to illustrate, on a low-dimensional contextual bandit problem with one state and one action, why the coherent reward with a stationary prior might be preferable to the classifier-based reward used by methods such as GAIL. Because it's a contextual bandit problem, there is no dynamical system.
> The agent will sample actions from the yellower regions.
> We will make this clearer in the caption.
> The coherent reward is constructed using a stationary nonparametric Gaussian process, so the color depends on the data points and the GP kernel.
> The main takeaway we wanted to communicate is that the coherent reward captures the shape of the true reward faithfully by using regression and the log ratio.
>
> **Reference policy.** This design decision is expanded in ‘Using the cloned policy as prior.’.
> If the prior $p(a|s)$ is the uniform distribution, then using it results in the maximum entropy soft Bellman equation.
> If instead the BC policy is used, like in the continuous setting, the soft Bellman equation and policy objective are now regularized explicitly against the initial BC policy.
> While this switch changes the policy update from which the coherent reward was derived, since the BC policy should match the prior outside of the demonstration distribution due to the stationary property, and we don't desire the policy to change significantly in-distribution, this switch does not change the objective too much and adds some very useful regularization.
>
> **BC vs CSIL performance.**
> The BC policy used by CSIL is trained in a simpler way than the BC baseline.
> For the BC baseline, we use a training protocol which learns for many iterations (e.g. 1M) and uses policy evaluation during training to pick the best performing policy, to mitigate the effects of under- and overfitting.
> For CSIL, we thought this protocol was too elaborate, so we instead use fewer iterations and early stopping to obtain a reasonable initial BC policy, which simplifies the algorithm and implementation.
> The BC baseline in Fig, 27 uses the training protocol described above, whereas the Fig. 31 ablation compares the initial CSIL BC policy.
>
> **Initial drop in CSIL performance.** We agree the initial performance below BC (CSIL) in Fig. 31 is unexpected.
> This happens rarely and the policy surpasses BC quickly.
> The best explanation we can give for the initial performance difference is that the initial $Q$ function may not be fully 'coherent' after SARSA pretraining, so the initial policy may 'unlearn' briefly during the initial updates.
> This is a consequence of having to use black-box function approximation for the critic.
>
> **Online Hopper-v2 performance.** Looking at Figure 27 and 31, it seems that CSIL on Hopper has the highest performance variance across the gym tasks, and is also quite high variance in the initial BC (CSIL) performance.
> This suggests that the Hopper data is harder to learn effective coherent rewards from; in other words, that it struggles to differentiate expert and non-expert actions and do good credit assignment.
> Also note that for Figure 3 we are pessimistic and use the best 25th percentile return, to show a best worst-case performance. In Figure 3, the selected Hopper-v2 performance is right at the end of training where both median performance and the interquartile range happen to drop. If you instead compare the step-based curves in Figure 27, the performance actually looks similar across the different dataset sizes but is quite high variance.
>
> **Expert performance.** We built on the datasets shared from peer-reviewed prior work (Orsini et al. [11]), who also used Acme and open-sourced their expert demonstrations for researchers to build off. We cannot comment on the performance of these demonstrations.
>
> **Lemma 2.** We believe we have written the result down correctly in Lemma 2. The DPI arises from manipulating a joint probability distribution and applying Jensen’s inequality, and in our case we use $p(y, w | x) = p(y|w, x)p(w)$ as our joint of interest.
>
> **Algorithm 1.** We wanted to keep Algorithm 1 high level so it captures a general description of CSIL within the available space.
> We have now added an additional algorithm in the Appendix that describes the SAC-based implementation and its additional tricks and highlighted it in the main text at the end of Section 4.
> To answer your questions:
> * The soft policy iteration loop at the end of the algorithm performs RL
> * We have fixed the math typo in the reward
> * $Q_n$ is updated using the soft Bellman equation in Equation 3. For SAC, learning uses target networks and the squared Bellman error.
>
> **CSIL without BC.** This is an interesting question. In theory (and the tabular case) it should work.
> We ran an ablation (BC iterations = 0) on the Gym tasks and added it to the rebuttal PDF.
> CSIL without BC it doesn't seem to work that well on Gym, probably because the reward learning is only tuned for refinement and the 'faithful' heteroscedastic regression fix is not done in the refinement loss.
>
> Please let us know if you have additional questions.

---

> > ### Comment · Reviewer_nMWn · 2023-08-16
> >
> > I would like to thank the authors for their replies. I was convinced. I have revised my score.

---

> > > ### Author Response · Authors · 2023-08-17
> > >
> > > Thanks. We are glad to hear your concerns have been resolved and you are happier with the work.

---

### Official Review · Reviewer_qELY · 2023-07-06

**Soundness:** 3 good
**Presentation:** 3 good
**Contribution:** 3 good
**Rating:** 7
**Confidence:** 3

**Summary:**

The authors propose a hybrid BC and IRL method, which uses a maximum entropy KL-regularized BC policy to define a shaped reward which can be optimized with IRL.  The authors derive a “coherent” reward which is defined as a reward for which the BC policy is optimal, by inverting the soft policy iteration update.  The full method first trains a BC policy, defines the coherent reward based on the policy, then uses soft policy iteration to optimize this reward.  The authors also provide a version of the method for continuous control which requires several additional components.  They compare against other IRL algorithms and BC in a tabular and continuous environments and in online and offline RL settings.  Overall, the method performs well in continuous environments given enough demonstrations and performs comparably with other methods elsewhere.

**Strengths:**

- The authors apply a novel perspective of coherence to formulating a hybrid BC and IRL method.
- The proposed method is well-derived with theoretical backing.
- Extensive connections to prior works throughout.
- Experiments in multiple different settings and control tasks.


**Weaknesses:**

- While the tabular form of the method is simple and elegant, the continuous version is quite complex and requires many additional components to work.  This includes training the critic with an additional auxiliary loss, using heteroscedastic regression to fit the expert data, and finetuning the coherent reward with a new objective.  In comparison, most other IL methods can be applied out-of-the-box for both discrete and continuous tasks.  Given its complexity, the proposed method could be difficult to reproduce and tune in new environments.  The authors do demonstrate the continuous method on multiple tasks in different domains and include multiple ablations.  However, many details and all ablations are not included in the main paper.  It would be helpful to at least summarize the results to address the method complexity.
- The derivation of the method can be difficult to follow and seems overly complicated in places.  For example, my understanding is that the initial policy is trained with KL-regularized BC but the authors use pseudo-likelihoods and stochastic process regression to derive it which seem like interesting connections but not necessary for the derivation.  Providing a simpler derivation or some intuition to guide readers could help make this part clearer.

Overall, these issues are minor, could be addressed with a little re-writing, and do not outweigh the main strengths of the paper.  The authors succeed in deriving and empirically verifying a novel hybrid BC and IRL method with the idea of coherence.  While additional analysis on the robustness of the method and a simplified derivation could help, these are not major concerns that detract from the paper's main contributions.


**Questions:**

- Is $r_\theta$ updated in each policy iteration with the new policy, $q_{\theta_i}$?  If so, does this change the coherence property?
- For continuous control policies, you state that you use the cloned policy as the prior in forming the reward.  Wouldn’t this lead to zero rewards everywhere because $r(s,a) = \alpha (\log q_\theta(s,a) - \log q_\theta(s,a))$?


**Limitations:**

Yes, the note that the BC policy is not always viable as a coherent reward, especially if the task is not solvable with BC even given many demonstrations.

---

> ### Author Rebuttal · Authors · 2023-08-09
>
> We wish to thank the reviewer for their comments and kind words about the submission.
> We have revised Section 3 to make the derivation easier to follow and emphasize the key technical points and motivations of the KL-regularized BC.
> We discuss more details of this text revision in our main rebuttal comment.
> We have also added more references to the relevant ablation studies in the main text.
>
> We now address specific concerns:
>
> **Implementation complexity.** In our experience, a deep imitation learning algorithm requires careful implementation details in practice. For example, GAIL / DAC implementations have additional regularization of the discriminator network using regularizers such as spectral norm or gradient penalties.
> IQLearn and PPIL used slightly different objectives (tuned per environment) to add additional regularization.
> We wanted to be explicit with the implementation details in the main paper for full transparency and to aid the ease of reproduction, as these implementation details are often hidden in the appendix or open-sourced implementation.
> We believe our extensive ablation experiments also help explain why each aspect is needed.
>
> **Updating reward parameters.** In the tabular setting, $r_{\theta}$ is fixed. In the function approximation, $r_\theta$ is refined due to non-stationary approximation error in the policy.
> This refinement is done to correct any stationary process approximation error out-of-distribution (i.e. where the BC policy doesn't match the prior), rather than alter the BC fit, so the coherency motivation is not violated.
> See Figure 9 in the Appendix to see a 1D example of how the refinement improves the stationary approximation of the policy.
>
> **Coherent reward prior.** We apologize that the ‘Using the cloned policy as prior.’ section is unclear.
> A uniform policy prior is used for the coherent reward in both the tabular and continuous implementations.
> The BC policy is not used as the prior in the coherent reward for the reason you correctly identify.
> For the deep imitation learning implementation, the BC policy is used as the prior in the KL-regularized Bellman equation (Equation 3) and policy objective to regularize the policy optimization. This is to encourage the policy to stay close to the BC policy, and has been used previously in online and offline RL to stabilize the policy updates.
> We have updated the text in this section so this passage is more clear.
> While this switch changes the policy update from which the coherent reward was derived, since the BC policy should match the prior outside of the demonstration distribution due to the stationary property, and we don't desire the policy to change significantly in-distribution, this switch does not change the objective substantially and adds beneficial policy regularization.
>
> We hope this answers your questions and addresses your concerns. Please let us know if you have additional issues.

---

> > ### Comment · Reviewer_qELY · 2023-08-16
> >
> > Thank you for your response and clarifications.  This addresses all my concerns and questions.

---

> > > ### Author Response · Authors · 2023-08-17
> > >
> > > Thank you for getting back to us. We are glad to hear your concerns have been addressed.

---

### Official Review · Reviewer_tsdg · 2023-07-08

**Soundness:** 3 good
**Presentation:** 3 good
**Contribution:** 3 good
**Rating:** 7
**Confidence:** 3

**Summary:**

The paper proposes an approach to inverse RL to fine-tune a behavior cloning policy using RL on online of offline data sources. Adopting the KL-regularized view to RL/IRL, CSIL expresses the reward in terms of the behavior cloned policy and connects the result with the well-known reward shaping results. This "coherent" reward is thereby used to improve the policy, resulting in a simple and performant inverse RL algorithm, compared to game-theoretic/adversarial approaches.

**Strengths:**

The paper provides a fairly comprehensive overview and background of inverse RL. The construction of the coherent reward using the entropy-regularized RL framework is interesting, and results in a simple algorithm with soft policy iteration. The experimental results are quite convincing and informative, and sufficiently compared against competitive methods.

**Weaknesses:**

In general, the presentation is a bit more jargon heavy than it needs to be, the writing can be greatly simplified to communicate key ideas more clearly, and make them more amenable for uptake.

The reward refinement seems to invoke a minimax optimization procedure. The paper claims to bypass adversarial IRL methods, but the reward refinement procedure seems to contradict that. Can you clarify the questions below?

It would be good to clarify where the uniform prior is used and where the reference policy is initialized to the BC cloned policy. More details for how the critic is initialized and trained would help address this question.

**Questions:**

How important is reward refinement? Do the final experiments use reward refinement? If so, can you report performance with and without reward refinement.

The paper mentions the issue of reward/critic causing the unlearning of the initialization. How does CSIL avoid this issue?

**Limitations:**

The paper provides a reasonable discussion of limitations, particularly how the success of CSIL can be sensitive to the success of the initial BC policy.

---

> ### Author Rebuttal · Authors · 2023-08-09
>
> Thank you for your review. We have revised Section 3 to improve clarity and minimize jargon, and revised the end of Section 4 to add more technical details regarding the reference policies and critic pretraining. To answer the questions here: In the coherent reward, the prior policy is always uniform. For deep imitation learning, in the Bellman equation and policy update objective, we replace the prior policy with the BC policy, for extra regularization. Regarding critic pretraining, in the tabular case we use the log policy ratio like the reward. For the deep imitation learning case, we train an MLP with SARSA and the squared Bellman error objective.
>
> We now answer the remaining questions:
>
> **Reward refinement.** Reward refinement is used in the deep imitation learning implementation. It is ablated in Appendix L.6, Figure 44 of the submission. The results show that it is mainly needed when there are few demonstrations (e.g. 1), and it also helps stabilize convergence in some cases. The reason why it is needed is to stop the RL agent from exploiting errors in the stationary approximation of the policy, which results in overestimation of the coherent reward being larger than desired. This issue is exacerbated in the single demonstration case since the initial BC is less defined and sub-optimal, and the coherent reward is defined in a smaller region of the state-action space, compared to when more demonstrations are available.
>
> Our claim that CSIL bypasses the minimax optimization of adversarial IRL is based on three factors:
> 1. The exact algorithm (e.g. in the tabular setting) does not require reward refinement.
> 2. The deep learning implementation can still work without reward refinement in some cases.
> 3. We use reward refinement to reduce approximation error rather than solve the IRL problem. With a better approximation of a stationary process (e.g. using a non-parametric implementation with a stationary kernel), the refinement step becomes less necessary.
>
> **Policy unlearning.** A randomly-initialized reward and critic network (used in baseline methods such as DAC and PPIL) can have arbitrary optimal actions for a given state. As a result, training a BC-initialized policy with this critic leads to ‘unlearning’ when the optimal actions of the BC policy differ from the arbitrary optimal actions of the randomly-initialized critic. In contrast, for our coherent reward, the optimal action is the maximum likelihood action of the BC policy, which after behavioral cloning should be equal or close to the expert action in the demonstration data. SARSA pretraining of the critic transfers these optimal actions from the reward to the critic. As a result, training the BC policy with this critic should not change the policy’s actions in-distribution since the BC policy, reward and critic are all ‘coherent’ w.r.t. the expert demonstrations.
>
> Outside of the expert data distribution, the optimal action is unknown and the coherent reward should be approximately zero. However, the learned Q function will encourage actions that return to the demonstration distribution since the coherent reward is higher in this region.
>
> We hope this addresses your questions and concerns. Please let us know if you have follow-up questions.

---

> > ### Comment · Reviewer_tsdg · 2023-08-12
> > **Thanks for the clarifications!**
> >
> > I have read the rebuttal and the response clarifies some of my concerns. I have revised my score accordingly. Great work!

---

> > > ### Author Response · Authors · 2023-08-17
> > >
> > > Thank you for replying to our rebuttal and the kind words. We are pleased to hear we have addressed your concerns.

---

### Author Rebuttal · Authors · 2023-08-09

We wish to thank all the reviewers for their comments and feedback. We believe we have been able to address all issues.

To summarize the four reviews:

**General Positives**
* CSIL is novel and interesting [tsdg, qely, nmwn, ba8w]
* The proposed method is sound theoretically [qely]
* Extensive connections to prior works [tsdg, qely]
* Comprehensive experiments, baselines, settings and / or ablations [tsdg, qely, nmwn, ba8w]

**General Weaknesses**
1. Clarity of Algorithm 1, Section 3 and the implementation details [tsdg, qely, nmwn, ba8w]
2. Missing an offline imitation learning baselines [ba8w]
3. Missing citation of Cao et al.’s similar policy inversion [ba8w]

**Our improvements**
1. Added missing technical details and improved clarity throughout the text, including Algorithm 1
    1. Restructured Section 3 to start with the policy inversion and then use the coherent reward to motivate pseudo-posteriors and regularized BC
    2. Added the coherent reward in an explicit and expanded definition block, and used an enumerated list to be more explicit about the regularized BC motivating points and details
    3. Added a second algorithm to the Appendix that describes the SAC-based implementation details and includes the additional implementation details from Section 4
2. We have run two additional offline imitation learning baselines: DemoDICE [1] and SmoDICE [2] and added them to Figure 5 and 29 (see PDF)
3. Added discussion of Cao et al. w.r.t. our Theorem 1

While we have restructured and rewritten parts of Section 3, we ensured the underlying topics and details covered content remains unchanged.

We are also preparing to open-source the code, so we can release our implementation if the submission is accepted.

The attached PDF contains the offline results with DemoDICE and SMODICE baselines, and an additional ablation study requested by nMWn that runs CSIL without BC pretraining.

We hope this addresses the reviewers concerns. Please let us know if there are remaining issues.

**A note on the new SMODICE and DemoDICE baselines.**
In the interest of time, we ran the author’s released code rather than implement the method in Jax and Acme like the other baselines.
The original papers used the D4RL datasets (expert and random) and hundreds (100-200) of demonstrations.
We ran the code for ten seeds with our offline setting (expert and full-replay datasets, with 1, 3, 10, 30 demonstrations).
The main difference is the use of the Orsini et al. expert datasets in our implementation and the use of the D4RL expert datasets in DemoDICE and SMODICE.
However, we have normalized to [0, 1] based on the expert performance in each algorithm’s corresponding expert dataset.

The interesting design decisions between SMODICE and DemoDICE are a discriminator-based reward, state-based value function and weighted BC-based policy update. These aspects separate them from the current offline baselines. The last two aspects are attractive for offline learning, as they mitigate the issue of estimating high value in unobserved actions.
SMODICE and DemoDICE are very similar methods.
The biggest difference between the methods is that DemoDICE learns a state-action reward, while SMODICE learns a state-based reward.

Our experiments show these are both two very strong baselines, especially for the Ant environment.
In particular, they also perform well in the one-demonstration setting.
However, CSIL maintains competitive performance on HalfCheetah-v2, Hopper-v2 and Walker-v2 for 10 and 30 demonstrations.

We would also like to note that some implementation details, such as weighted BC-based policy update, could be incorporated into CSIL, since it is agnostic to the SPI implementation (e.g. use MPO rather than SAC). The primary goal with our offline experiment was to compare CSIL against its similar baselines IQLearn and PPIL, since they share many implementation details.

[1] Demodice: Offline imitation learning with supplementary imperfect demonstrations, Kim et al., ICLR 2022

[2] Versatile offline imitation from observations and examples via regularized state-occupancy matching, Ma et al., ICML 2022

---

### Decision · Program_Chairs · 2023-09-21

**Decision:**

Accept (spotlight)

**Comment:**

Dear Authors,

After an exhaustive review and contemplation of the reviewers' feedback, the decision to accept the paper for NeurIPS 2023 has been reached.  We acknowledge the strengths that the work brings to the field, particularly in the domain of reward refinement and its implications in deep imitation learning. The clarification on the usage of reward refinement, especially when fewer demonstrations are present, is appreciated. The points related to CSIL bypassing the minimax optimization of adversarial IRL, through the three mentioned factors, have been comprehensively understood. Furthermore, the reviewers recognize the value of the paper’s exploration of the implementation complexity, and appreciate the transparency maintained by highlighting these often overlooked details in the primary content, which indeed aids in the reproduction of experiments.

After a comprehensive inspection, the conclusion has been made to accept the paper. We sincerely hope that this feedback serves as a useful guide and looks forward to witnessing the paper's contributions to the NeurIPS community.

Best regards,

Area Chair